# Exogenous Matching: Learning Good Proposals for Tractable Counterfactual Estimation

**Yikang Chen**[1], **Dehui Du**[1,*], **Lili Tian**[1]

[1]Shanghai Key Laboratory of Trustworthy Computing, East China Normal University

## Abstract

We propose an importance sampling method for tractable and efficient estimation of counterfactual expressions in general settings, named Exogenous Matching. By minimizing a common upper bound of counterfactual estimators, we transform the variance minimization problem into a conditional distribution learning problem, enabling its integration with existing conditional distribution modeling approaches. We validate the theoretical results through experiments under various types and settings of Structural Causal Models (SCMs) and demonstrate the outperformance on counterfactual estimation tasks compared to other existing importance sampling methods. We also explore the impact of injecting structural prior knowledge (counterfactual Markov boundaries) on the results. Finally, we apply this method to identifiable proxy SCMs and demonstrate the unbiasedness of the estimates, empirically illustrating the applicability of the method to practical scenarios.[1]

## 1 Introduction

Counterfactual reasoning, considered one of the advanced cognitive abilities of humans, aims to address questions about hypothetical worlds that have not been observed. Answering these questions, typically in the form of "what-if" or "why," is crucial for attribution and explanation, thus counterfactual reasoning is widely applied in generating explanations [48, 47, 94, 63], making decisions [71, 37, 95, 58], and evaluating fairness [56, 113, 105, 78, 117]. A class of generative models called Structural Causal Models (SCMs) provides a semantic characterization of counterfactuals, consisting of a set of endogenous mechanisms and an exogenous distribution, from which a collection of distributions described by specific formal languages is in-

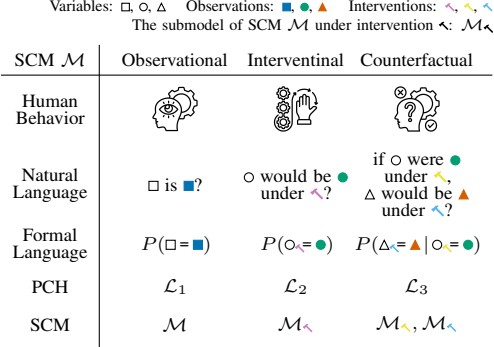

Figure 1: A brief illustration of SCM, PCH and counterfactual concepts.

duced. These languages are organized into a progressively refined hierarchical structure, and each is associated with human activities: language $\mathcal{L}_1$ is about seeing (observational), $\mathcal{L}_2$ is about doing (interventional), and $\mathcal{L}_3$ is about imagining (counterfactual). This hierarchy is referred to as the causal ladder [77] or the Pearl Causal Hierarchy (PCH) [5]. Fig. 1 provides a brief illustration of the relevant concepts. As the highest level of the PCH, counterfactuals entail the most subtle information

---

[*]Corresponding author (dhdu@sei.ecnu.edu.cn)
[1]Code is available at: https://github.com/cyisk/exom

38th Conference on Neural Information Processing Systems (NeurIPS 2024).

in the hierarchy, making the evaluation of counterfactual languages an appealing objective that has received widespread attention in recent years.

Counterfactual reasoning consists of two subtasks: counterfactual identification and counterfactual estimation. For an expression $\mathcal{Q}$ in $\mathcal{L}_3$, identification is to determine the uniqueness of the answer to $\mathcal{Q}$ [86, 17] or its bounds [114, 25, 66] under some assumptions, while estimation is to compute the specific value of $\mathcal{Q}$. A series of literature [86, 17] transforms $\mathcal{Q}$ into a combination of expressions in $\mathcal{L}_1$ and $\mathcal{L}_2$ through identification. However, estimating the transformed expression may be intractable. Another series of literature focuses on identifying and estimating $\mathcal{Q}$ in specific settings, such as back door [82, 38, 115, 102], front door [43, 109, 98], instrumental variables [14, 100, 35], representation decoupling [15, 97, 61], categorical exogenous [22], exponential family [85], domain counterfactuals [116]. There are also some literature applicable to relaxed cases, such as [28, 44, 45, 46, 57], but still less developed.

In recent years, a class of methods has emerged that simulate the structure of the original SCM using neural network architectures and learn causal mechanisms based on observed or intervened distributions, which we refer to as neural proxy SCMs. These methods include the use of VAEs [110, 55, 54, 92, 23], GANs [52, 108], normalizing flows [74, 50, 4, 69, 42], DDPMs [84, 12]. Here, we focus only on models that provide identifiability results. Using bijections, [69, 42] demonstrates the identifiability of learned causal mechanisms. However, constrained by bijections, they only provide tractable counterfactual estimation methods for fully observed endogenous variables. The closest method to performing tractable counterfactual estimation in general cases is NCM [107, 108], which employs an optimization algorithm to concurrently achieve tractable counterfactual estimation and identification (or partial identifiability [93]). However, the counterfactual estimation depends on rejection sampling, thereby lacking scalability.

It is commonly recognized that performing tractable counterfactual estimation in general cases is challenging, even when the SCM is fully specified. From the perspective of definition [5], one needs to integrate over the entire exogenous space, which involves determining whether the constraints of different hypothetical worlds are simultaneously satisfied. From the perspective of complexity, although [34] has demonstrated that counterfactual reasoning is tractable if observational and interventional reasoning are tractable, it relies on algorithms with exponential complexity. Furthermore, according to [112], performing marginal inference on SCMs, including parameterized SCMs such as NCM, has been shown to be NP-hard.

This paper presents a tractable and efficient counterfactual estimation method in general settings based on importance sampling. Here, by "general settings", we mean that the exogenous variables of the SCM can be discrete or continuous, the expressions to be estimated can involve an arbitrary finite number of hypothetical worlds, and the observations and interventions can also be arbitrary.

Specifically, the contributions of this paper are as follows:

1. We propose a tractable and efficient importance sampling method for counterfactual estimation in general settings, which is based on optimizing an upper bound on the variance of the estimators, and is formulated as a conditional distribution learning problem.
2. We explore the injection of structural prior knowledge (counterfactual Markov boundaries) into neural networks used for parameter optimization.
3. The experiments conducted on different SCM settings demonstrate that the proposed method outperforms existing importance sampling methods. Ablation studies empirically highlight the effectiveness of injecting structural prior knowledge. The experiments on counterfactual estimation tasks conducted on two identifiable proxy SCMs illustrate the feasibility of applying the method in conjunction with proxy SCMs to real-world problems.

## 2   Preliminaries

In this section, we provide the necessary background and definitions for our work, which are consistent with [5, 8, 114]. To maintain consistency in notation, we use the uppercase letter $X$ to denote random variables and the lowercase letter $x$ to denote their values. Bold uppercase letter $\mathbf{X}$ represents a set of variables and $\mathbf{x}$ the corresponding set of values. The domain of the variable $X$ is denoted as $\Omega_X$, and $\Omega_{\mathbf{X}}$ for $\mathbf{X} = \{X_1, \ldots, X_n\}$ represents the product of domains $\bigtimes_{i=1}^{n} \Omega_{X_i}$. We use $P_{\mathbf{X}}$ to denote a distribution over a set of random variables $\mathbf{X}$, and $P(\mathbf{X} \in \mathcal{X})$ (abbreviated as $P(\mathcal{X})$) to denote

probability when $\mathbf{X}$ take values $\mathcal{X} \subseteq \Omega_{\mathbf{X}}$. The lowercase $p(\mathbf{x})$ represents the probability density function of $P_{\mathbf{X}}$ if $\Omega_{\mathbf{X}}$ is continuous.

**Structural causal models** An SCM is a 4-tuple $\mathcal{M} = \langle \mathbf{U}, \mathbf{V}, \mathcal{F}, P_{\mathbf{U}} \rangle$, where $\mathbf{U}$ is a set of exogenous variables; $\mathbf{V}$ is a set of endogenous variables; $\mathcal{F}$, which describes the causal mechanisms of $\mathcal{M}$, is a collection of functions $\{f_{V_1}, f_{V_2}, \ldots, f_{V_n}\}$ such that each endogenous variable $V_i$ is determined by a function $f_{V_i}$. Each $f_{V_i} \in \mathcal{F}$ is a mapping from (the domains of) $\mathbf{U}_{V_i} \cup \mathbf{Pa}_{V_i}$ to $V_i$, where $\mathbf{U}_{V_i} \subseteq \mathbf{U}$ and $\mathbf{Pa}_{V_i} \subseteq \mathbf{V} \setminus V_i$. The entire set $\mathcal{F}$ forms a mapping from $\mathbf{U}$ to $\mathbf{V}$, while the uncertainty comes from a probability distribution $P_{\mathbf{U}}$ over exogenous variables $\mathbf{U}$.

An SCM yields a causal graph $\mathcal{G}$, where each $V_i \in \mathbf{V}$ is a vertex, and the edges come in two types: there is a directed edge $(V_i \to V_j)$ between each pair $V_i$ and $V_j \in \mathbf{Pa}_{V_i}$; and there is a bidirected edge $(V_i \leftrightarrow V_j)$ between each pair $V_i$ and $V_j$ if $\mathbf{U}_{V_i} \cup \mathbf{U}_{V_j} \neq \emptyset$. This resulting graph is also known as a directed mixed graph (DMG). Here, we assume that the SCM is recursive, which means that there exists an order in $\mathcal{F}$ such that for any pair $f_i, f_j \in \mathcal{F}$, if $f_i < f_j$, then $V_j \notin \mathbf{Pa}_{V_i}$. In particular, this implies that given values for exogenous variables (also known as a unit or an individual) $\mathbf{u} \in \Omega_{\mathbf{U}}$, all values of endogenous variables in $\mathbf{V}$ are also fixed (i.e., there exists a unique solution for endogenous variables w.r.t. $\mathbf{u}$), and the resulting causal graph $\mathcal{G}$ is an acyclic directed mixed graph (ADMG). We denote $\mathbf{Y}(\mathbf{u})$ as the solution for $\mathbf{Y} \subseteq \mathbf{V}$ given $\mathbf{u}$.

In the paradigm of SCM, an (perfect) intervention is described as replacing the mechanisms of some variables $\mathbf{X}$ with constants $\mathbf{x}$. The model after this intervention is termed a submodel, defined as

$$\mathcal{M}_{\mathbf{x}} = \langle \mathbf{U}, \mathbf{V}, \mathcal{F}_{\mathbf{x}}, P_{\mathbf{U}} \rangle, \quad \text{where } \mathcal{F}_{\mathbf{x}} = \{f_i : V_i \notin \mathbf{X}\} \cup \{\mathbf{X} \leftarrow \mathbf{x}\}. \tag{1}$$

In submodel $\mathcal{M}_{\mathbf{x}}$, the solution of $\mathbf{Y} \subseteq \mathbf{U}$ given $\mathbf{u}$, denoted as $\mathbf{Y}_{\mathcal{M}_{\mathbf{x}}}(\mathbf{u})$, is termed as the potential response, and is typically abbreviated as $\mathbf{Y}_{\mathbf{x}}(\mathbf{u})$.

**Counterfactual events and probabilities** The random variable $Y \in \mathbf{V}$ corresponding to the submodel $\mathcal{M}_{\mathbf{x}}$ is denoted as $Y_{\mathbf{x}}$, termed a counterfactual variable, with its domain denoted as $\Omega_{Y_{\mathbf{x}}}$. A set of counterfactual variables from the same submodel $\mathcal{M}_{\mathbf{x}}$ is denoted as $\mathbf{Y}_{\mathbf{x}}$, and the set of counterfactual variables from $k$ different submodels $\mathcal{M}_{\mathbf{x}_1}, \ldots, \mathcal{M}_{\mathbf{x}_k}$ is denoted as $\mathbf{Y}_* = \bigcup_{i=1}^{k} \mathbf{Y}_{i[\mathbf{x}_i]}$, with the corresponding domain $\Omega_{\mathbf{Y}_*} = \times_{i=1}^{k} \Omega_{\mathbf{Y}_{i[\mathbf{x}]}}$. Let $\sigma$-algebra $\Sigma_{\mathbf{Y}_*} \subseteq 2^{\Omega_{\mathbf{Y}_*}}$, where $2^{\Omega_{\mathbf{Y}_*}}$ is the power set of $\Omega_{\mathbf{Y}_*}$. The counterfactual probability space is then a triplet $\langle \Omega_{\mathbf{Y}_*}, \Sigma_{\mathbf{Y}_*}, P_{\mathbf{Y}_*} \rangle$, where $P_{\mathbf{Y}_*}$ is a probability measure $P_{\mathbf{Y}_*} : \Sigma_{\mathbf{Y}_*} \to [0, 1]$, and for any measurable set $\mathcal{Y}_* \in \Sigma_{\mathbf{Y}_*}$, its value equals the Lebesgue integral,

$$P_{\mathbf{Y}_*}(\mathcal{Y}_*) = \int_{\Omega_{\mathbf{U}}} \mathbb{1}_{\Omega_{\mathbf{U}}(\mathcal{Y}_*)}(\mathbf{u}) \, dP_{\mathbf{U}}, \tag{2}$$

where $\mathbb{1}_{\Omega}(\mathbf{u})$ is an indicator function, equal to 1 if $\mathbf{u} \in \Omega$ and 0 otherwise. The set $\Omega_{\mathbf{U}}(\mathcal{Y}_*) = \{\mathbf{u} \mid \mathbf{Y}_*(\mathbf{u}) \in \mathcal{Y}_*\}$, where $\mathbf{Y}_*(\mathbf{u})$ denotes the potential responses $\bigcup_{i=1}^{k} \mathbf{Y}_{i[\mathbf{x}_i]}(\mathbf{u})$ of counterfactual variables in $\mathbf{Y}_*$ w.r.t. $\mathbf{u}$. We refer to the random event $\mathbf{Y}_* \in \mathcal{Y}_*$ as a counterfactual event, and its probability $P(\mathcal{Y}_*) = P_{\mathbf{Y}_*}(\mathcal{Y}_*)$ is termed the counterfactual probability.

Counterfactual probability is the cornerstone of the language $\mathcal{L}_3$ in PCH. Specifically, all expressions in $\mathcal{L}_3$ consist of inequalities between polynomials (and their Boolean combinations) over terms of the form $P(\mathcal{Y}_*)$. Here, we assume that the $\mathcal{L}_3$ expressions to be estimated can be represented by a finite number of $P(\mathcal{Y}_*)$ terms (including those approximated by Monte Carlo methods). Therefore, the estimation of $P(\mathcal{Y}_*)$ will become a focal point of the subsequent discussion.

**Normalizing flows** The normalizing flow $\mathcal{T}_\theta : \mathbb{R}^d \to \mathbb{R}^d$ is a transformation parameterized by $\theta$ that maps observable samples $\mathbf{x}$ with $d$ features to latent samples $\mathbf{z}$ distributed according to a simple distribution $P_{\mathbf{Z}}$ with probability density function $p(\mathbf{z})$. The mapping $\mathcal{T}_\theta$ is a diffeomorphism, which allows us to compute the density $p(\mathbf{x})$ through the change-of-variables formula:

$$\log p(\mathbf{x}) = \log p(\mathcal{T}_\theta(\mathbf{x})) + \log |\det(\nabla_{\mathbf{x}} \mathcal{T}_\theta(\mathbf{x}))|. \tag{3}$$

When the Jacobian determinant of $\mathcal{T}_\theta(\mathbf{x})$ is tractable to compute, we can use samples from the observable distribution and maximum likelihood estimation to train the flow $\mathcal{T}_\theta$. Additionally, we can easily generate i.i.d. samples conforming to the learned observable distribution by first sampling $\mathbf{z}^{(i)}$ from the base distribution $P_{\mathbf{Z}}$, and then evaluating the inverse transformation $\mathcal{T}_\theta^{-1}(\mathbf{z}^{(i)})$.

## 3 Exogenous Matching

In this section, we present only the final theoretical conclusions. Detailed derivations of all theories and equations can be found in App. A.

**Assumptions** In addition to assuming that the SCM $\mathcal{M} = \langle \mathbf{U}, \mathbf{V}, \mathcal{F}, P_{\mathbf{U}} \rangle$ is recursive, the proposed method also assumes: i) for any $\mathbf{X} \subseteq \mathbf{V}, \mathbf{x} \in \Omega_{\mathbf{X}}, \mathbf{u} \in \Omega_{\mathbf{U}}, \mathcal{F}_{\mathbf{x}}(\mathbf{u})$ is computable; ii) $P_{\mathbf{U}}$ is sampleable and computable. These assumptions imply the feasibility of counterfactual generation, meaning we can draw arbitrary counterfactual samples, with $\mathcal{F}_{\mathbf{x}}$ and $P_{\mathbf{U}}$ both treated as black boxes. For example, in the context of neural proxy SCMs, $P_{\mathbf{U}}$, as the latent distribution of the generative model, is typically modeled as a simple distribution, while $\mathcal{F}_{\mathbf{x}}$ is a neural network.

**Importance sampling for Monte Carlo integration** A widely used method for estimating the Lebesgue integral is Monte Carlo integration. However, the indicator function $\mathbb{1}_{\Omega_{\mathbf{U}}(\mathcal{Y}_*)}(\mathbf{u})$ in Eq. 2 limits the efficiency. We consider $\mathbb{1}_{\Omega_{\mathbf{U}}(\mathcal{Y}_*)}(\mathbf{u}) = 1$ as a rare event. Since importance sampling is commonly used to handle Monte Carlo estimations involving rare events, we adopt the method of importance sampling on a parameterized proposal distribution $Q_{\mathbf{U}}$ for tractable Monte Carlo integration, and the estimator can be derived as:

$$P(\mathcal{Y}_*) = \mathbb{E}_{\mathbf{u} \sim Q_{\mathbf{U}}} [\sigma_{\mathcal{Y}_*}(\mathbf{u})] \approx \frac{1}{n} \sum_{i=1}^{n} \sigma_{\mathcal{Y}_*}(\mathbf{u}^{(i)}), \quad \text{let } \sigma_{\mathcal{Y}_*}(\mathbf{u}) = \frac{p(\mathbf{u})}{q(\mathbf{u})} \mathbb{1}_{\Omega_{\mathbf{U}}(\mathcal{Y}_*)}(\mathbf{u}), \tag{4}$$

where for each sample $\mathbf{u}^{(i)} \sim Q_{\mathbf{U}}$, it is required that if $p(\mathbf{u}^{(i)}) > 0$, then $q(\mathbf{u}^{(i)}) > 0$. The density ratio $p(\mathbf{u})/q(\mathbf{u})$ is also termed as the importance weight and is usually assumed to be bounded. In this section, we assume that our derivations are based on an exogenous distribution defined over a continuous space. Naturally, similar conclusions hold for discrete exogenous distributions.

### 3.1 An Optimizable Variance Upper Bound

In importance sampling, a proposal distribution with low variance regarding $\sigma_{\mathcal{Y}_*}$ is crucial, as it implies that fewer samples are required to achieve the same effect, leading to higher sampling efficiency. Previous works have attempted to constrain variance, such as providing upper bounds [18, 49], introducing multiple proposals [16, 27], and learning from sampled data [72, 83, 11, 65]. Given that the samples are i.i.d., with the same sample size $n$, a lower variance of $\sigma_{\mathcal{Y}_*}$ implies a lower estimator variance:

$$\mathbb{V}_{\mathbf{u} \sim Q_{\mathbf{U}}} [\sigma_{\mathcal{Y}_*}(\mathbf{u})] = n \cdot \mathbb{V} \left[ \frac{1}{n} \sum_{i=1}^{n} \sigma_{\mathcal{Y}_*}(\mathbf{u}^{(i)}) \right] = \mathbb{E}_{\mathbf{u} \sim P_{\mathbf{U}}} [\sigma_{\mathcal{Y}_*}(\mathbf{u})] - P^2(\mathcal{Y}_*), \tag{5}$$

It is evident that this formulation can be directly optimized, and there are several methods employing similar approaches (e.g., transforming it into cross-entropy or $\chi^2$ divergence), as shown in [32, 68]. This process is called proposal learning, which ultimately yields a proposal distribution $Q_{\mathbf{u}}$ that is suitable for the estimator.

The learned $Q_{\mathbf{U}}$ is typically "one-off", rendering it unsuitable for other estimators. This necessitates multiple rounds of proposal learning to estimate an $\mathcal{L}_3$ expression that involves various counterfactual events, which remains inefficient. To address this issue, we propose replacing the proposal distribution $Q_{\mathbf{U}}$ with a conditional proposal distribution $Q_{\mathbf{U}|\mathbf{Y}_*}$. Intuitively, the proposal distribution $Q_{\mathbf{U}|\mathbf{y}_*}$ corresponding to $\mathbf{y}_* \in \Omega_{\mathbf{Y}_*}$ should be concentrated on the support $\Omega_{\mathbf{U}}(\delta(\mathbf{y}_*))$, thus allowing for reuse across different estimators.

Building on this idea, we extend the methods from [32, 68], as illustrated in App. B.1. Additionally, we find that when certain conditions are met in $\mathcal{Y}_*$, for any $\mathbf{y}_* \in \mathcal{Y}_*$, the proposal distribution $Q_{\mathbf{U}|\mathbf{y}_*}$ shares a common variance upper bound:

**Theorem 1** (Variance Upper Bound). Let $\sigma_{\mathcal{Y}_*}(\mathbf{u}) = (p(\mathbf{u})/q(\mathbf{u}|\mathbf{y}_*)) \mathbb{1}_{\Omega_{\mathbf{U}}(\mathcal{Y}_*)}(\mathbf{u})$, where $q(\mathbf{u}|\mathbf{y}_*)$ denote the density of the proposal distribution $Q_{\mathbf{U}|\mathbf{y}_*}$, and let $\mathbf{Y}_*(\mathbf{u})$ be the potential response w.r.t. $\mathbf{u}$. If for any $\mathbf{y}_* \in \mathcal{Y}_*$, there exists $\kappa \geq 1$ such that $1/\kappa \leq p(\mathbf{u})/q(\mathbf{u}|\mathbf{y}_*) \leq \kappa$ holds almost surely on the support $\Omega_{\mathbf{U}}(\mathcal{Y}_*)$, then for any $Q_{\mathbf{U}|\mathbf{y}_*}$ where $\mathbf{y}_* \in \mathcal{Y}_*$:

$$\mathbb{V}_{\mathbf{u} \sim Q_{\mathbf{U}|\mathbf{y}_*}} [\sigma_{\mathcal{Y}_*}(\mathbf{u})] \leq -\mathbb{E}_{\mathbf{u} \sim P_{\mathbf{U}}} [\log q(\mathbf{u}|\mathbf{Y}_*(\mathbf{u}))] + c, \tag{6}$$

where the constant $c$ is solely dependent on $\kappa$ and $P_{\mathbf{U}}$.

It is noted that the expected term in Eq. 6 bears resemblance to cross-entropy, albeit not strictly. Intuitively, the original indicator function is now implied as potential responses within the condition, and optimizing this expected term can encourage the density of each proposal distribution at specific locations to match that of the exogenous distribution as closely as possible. Therefore, this approach is named **Exogenous Matching (EXOM)**.

The boundedness condition on the importance weights in Thm. 1 is likely to be violated for $\mathcal{Y}_*$ with an infinite support set. We further discuss this issue in App. A.2, where we: i) provide two relaxed versions of Thm. 1 based on concentration inequalities; ii) demonstrate how to construct a guard proposal distribution that necessarily satisfies the boundedness condition.

## 3.2 Learning and Inference

**Generalize Thm. 1** Since any $\mathcal{L}_3$ expression $\mathcal{Q}$ is composed of terms $P(\mathcal{Y}_*)$, and assuming that it is expressible by a finite set of $P(\mathcal{Y}_*)$, we can consider all $\mathcal{Y}_*$ involved in $\mathcal{Q}$ as outcomes of a stochastic process. To allow different forms of $\mathcal{Y}_*$, we extend Thm. 1 to general cases, making it potentially applicable to the estimation of any counterfactual probability term $P(\mathcal{Y}_*)$ in an $\mathcal{L}_3$ expression $\mathcal{Q}$, with only one proposal learning process is required.

**Definition 1** (Stochastic Counterfactual Process). The collection of counterfactual variable sets $\mathfrak{Y}_* = \{\mathbf{Y}_*^{(s)} \mid s \in \mathcal{S}\}$ is referred to as a stochastic counterfactual process w.r.t. the state space $\mathcal{S}$. A state $s \in \mathcal{S}$ is a set composed of triplets $\langle \mathbf{Y}, \mathbf{X}, \mathbf{x} \rangle$, where $\mathbf{X}, \mathbf{Y} \subseteq \mathbf{V}$ represent the intervened and observed variables, respectively, and $\mathbf{x}$ denotes the intervened values. Each state $s$ corresponds to a set of counterfactual variables $\mathbf{Y}_*^{(s)} = \bigcup_{\langle \mathbf{Y}, \mathbf{X}, \mathbf{x} \rangle \in s} \mathbf{Y}_{\mathbf{x}}$.

**Corollary 1** (Expected Variance Upper Bound). Let $Q_{\mathcal{S}}$ denote an arbitrary distribution defined over the state space $\mathcal{S}$ of a stochastic counterfactual process, and let $P_{\mathcal{Y}_*}^{(s)}$ denote an arbitrary distribution defined over the $\sigma$-algebra $\Sigma_{\mathbf{Y}_*}^{(s)}$ corresponding to a set of counterfactual variables $\mathbf{Y}_*^{(s)}$ given a state $s \in \mathcal{S}$. $P_{\mathcal{Y}_*}$ is the joint distribution induced by $P_{\mathcal{Y}_*}^{(s)}$ and $Q_{\mathcal{S}}$. If the conditions in Thm. 1 are met for any $\mathcal{Y}_*^{(s)}$ and any $s \in \mathcal{S}$, then for any $Q_{\mathbf{U}|\mathbf{y}_*}$ where $\mathbf{y}_* \in \mathcal{Y}_*^{(s)}$ and any $s \in \mathcal{S}$:

$$\mathbb{E}_{\mathcal{Y}_*^{(s)} \sim P_{\mathcal{Y}_*}} \left[ \mathbb{V}_{\mathbf{u} \sim Q_{\mathbf{U}|\mathbf{y}_*}} \left[ \sigma_{\mathcal{Y}_*}(\mathbf{u}) \right] \right] \leq -\mathbb{E}_{s \sim Q_{\mathcal{S}}} \left[ \mathbb{E}_{\mathbf{u} \sim P_{\mathbf{U}}} \left[ \log q(\mathbf{u} \mid \mathbf{Y}_*^{(s)}(\mathbf{u})) \right] \right] + c, \qquad (7)$$

where the constant $c$ is the same as in Thm. 1.

**Sampling and optimization** Disregarding the constant terms of the expected variance upper bound (Eq. 7) in Cor. 1, we obtain the optimization objective as follows:

$$\arg\min_q -\mathbb{E}_{s \sim Q_{\mathcal{S}}} \left[ \mathbb{E}_{\mathbf{u} \sim P_{\mathbf{U}}} \left[ \log q(\mathbf{u} \mid \mathbf{Y}_*^{(s)}(\mathbf{u})) \right] \right]. \qquad (8)$$

Based on our assumptions, $P_{\mathbf{U}}$ and $\mathbf{Y}_*^{(s)}(\mathbf{u})$ are known, while the prior distribution of states $Q_{\mathcal{S}}$ needs to be specifically designed for the particular expression in $\mathcal{L}_3$. The problem is now reformulated as a conditional distribution learning problem, and we can model this conditional distribution using neural networks. The complete learning algorithm is presented in Fig. 5 in App. A.3.

**Inference** Given the conditional proposal $Q_{\mathbf{U}|\mathbf{Y}_*}$, we can take the expectation of the importance sampling results over $\mathcal{Y}_*$ to intuitively further enhance robustness. This leads to another unbiased estimator in the form of multiple importance sampling [16, 27]:

$$P(\mathcal{Y}_*) = \mathbb{E}_{\mathbf{y}_* \sim Q_{\mathbf{Y}_*}} \left[ \mathbb{E}_{\mathbf{u} \sim Q_{\mathbf{U}|\mathbf{y}_*}} \left[ \sigma_{\mathcal{Y}_*[\mathbf{y}_*]}(\mathbf{u}) \right] \right] \approx \frac{1}{n} \sum_{i=1}^{n} \sigma_{\mathcal{Y}_*[\mathbf{y}_*^{(i)}]}(\mathbf{u}^{(i)}), \qquad (9)$$

where $\sigma_{\mathcal{Y}_*[\mathbf{y}_*]}(\mathbf{u}) = (p(\mathbf{u})/q(\mathbf{u} \mid \mathbf{y}_*)) \mathbb{1}_{\Omega_{\mathbf{U}}(\mathcal{Y}_*)}(\mathbf{u})$, and $\mathbf{y}_*^{(i)} \sim Q_{\mathbf{Y}_*}$, $\mathbf{u}^{(i)} \sim Q_{\mathbf{U}|\mathbf{y}_*^{(i)}}$. The distribution $Q_{\mathbf{Y}_*}$ could be any distribution with support covering $\mathcal{Y}_*$.

**Conditioning** We attempt to find a function that maps $\mathbf{y}_*$ to the vectorized parameters $\theta_{\mathbf{y}_*}$ required for the proposal distribution $Q_{\mathbf{U}|\mathbf{y}_*}$. The information provided by $\mathbf{y}_*$ (i.e., the counterfactual event $\mathbf{Y}_* \in \{\mathbf{y}_*\}$) can be interpreted as a set of 4-tuples, where each 4-tuple $\langle \mathbf{Y}, \mathbf{y}, \mathbf{X}, \mathbf{x} \rangle$ expresses

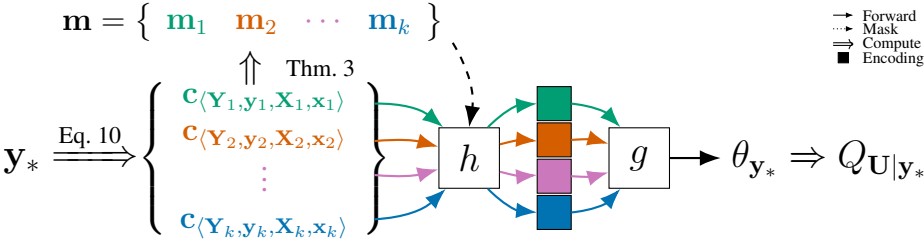

Figure 2: Overview of the conditioning and masking process. $\mathbf{y}_*$ serves as the input to the entire process, $\mathbf{m}$ represents the inferred mask, and the vectorized parameters $\theta_{\mathbf{y}_*}$ of the proposal distribution $Q_{\mathbf{U}|\mathbf{y}_*}$ are the output. Different colors represent information from different submodels. Both $h$ and $g$ represent neural networks.

counterfactual information within the same submodel $\mathcal{M}_{\mathbf{x}}$, including the observed variables $\mathbf{Y}$ and their values $\mathbf{y}$, and the intervened variables $\mathbf{X}$ and their values $\mathbf{x}$. We represent it as a vector:

$$\mathbf{c}_{\langle \mathbf{Y}, \mathbf{y}, \mathbf{X}, \mathbf{x} \rangle} = \pi(\mathbf{y} \cup \mathbf{x}; \mathbf{Y} \cup \mathbf{X}) \oplus \omega(\mathbf{Y}) \oplus \omega(\mathbf{X}), \tag{10}$$

where $\oplus$ denotes vector concatenation. The functions $\pi$ and $\omega$ map inputs to vectors, with each component of the vector corresponding to an exogenous variable $X_i \in \mathbf{V}$, defined as follows:

$$\pi_i(\mathbf{x}; \mathbf{X}) = \begin{cases} x_i, & \text{if } X_i \in \mathbf{X} \\ 0, & \text{otherwise} \end{cases} \quad \text{and} \quad \omega_i(\mathbf{X}) = \begin{cases} 1, & \text{if } X_i \in \mathbf{X} \\ 0, & \text{otherwise} \end{cases}. \tag{11}$$

To encode the entire set $\{\langle \mathbf{Y}_1, \mathbf{y}_1, \mathbf{X}_1, \mathbf{x}_1 \rangle, \langle \mathbf{Y}_2, \mathbf{y}_2, \mathbf{X}_2, \mathbf{x}_2 \rangle, \dots \}$, we introduce two functions, $h$ and $g$. Here, $h$ serves as an encoder, mapping the vector of a single 4-tuple $\mathbf{c}_{\langle \mathbf{Y}, \mathbf{y}, \mathbf{X}, \mathbf{x} \rangle}$ to a latent encoding, while $g$ acts as an aggregator, capturing the permutation invariance of the set. The vectorized parameters $\theta_{\mathbf{y}_*}$ are inferred through the interplay of these two functions:

$$\theta_{\mathbf{y}_*} = g\left(\left\{ h(\mathbf{c}_{\langle \mathbf{Y}_i, \mathbf{y}_i, \mathbf{X}_i, \mathbf{x}_i \rangle}) \mid \mathbf{Y}_{i[\mathbf{x}_i]} \in \mathbf{Y}_* \right\}\right). \tag{12}$$

We choose to model $h$ using a multilayer perceptron (MLP), and model $g$ as a function that ensures permutation invariance for encoding aggregation, such as summation or weighted summation, where we opt for attention [96] to enhance expressiveness.

### 3.3 Injecting Markov Boundaries

This section will explore how to inject available structural prior knowledge (specifically, counterfactual Markov boundaries) into neural networks used for conditional encoding, intuitively improving the quality and efficiency of learning.

**Counterfactual Markov boundary** Markov boundaries [75] have been used in feature selection [111, 64, 104] and causal discovery [3, 87, 103], since they reveal the local causal structure of variables, where all elements are strongly correlated with the variable [2]. Just as in feature selection, intuitively we can enhance the performance of distribution learning by masking redundant information. For modeling distributions defined over exogenous variables and conditioned on counterfactual variables, the structural prior information used here is named counterfactual Markov boundaries:

**Definition 2** (Counterfactual Markov Boundary). For an exogenous variable $U_j \in \mathbf{U}$ and a set of counterfactual variables $\mathbf{Y}$, along with their joint distribution $P$. If $U_j$ is independent of $\mathbf{Y} \setminus \mathbf{X}$ given $\mathbf{X}$ under $P$, i.e. $U_j \perp\!\!\!\perp_P (\mathbf{Y} \setminus \mathbf{X}) \mid \mathbf{X}$, then $\mathbf{X}$ is termed a Markov blanket of $U_j$ on $\mathbf{Y}$. The collection of all Markov blankets of $U_j$ on $\mathbf{Y}$ is denoted as $\mathfrak{B}_j(\mathbf{Y})$. If $\mathbf{X} \in \mathfrak{B}_j(\mathbf{Y})$ is a Markov blanket of $U_j$ on $\mathbf{Y}$, and there exists no $\mathbf{X}' \subsetneq \mathbf{X}$ such that $\mathbf{X}' \in \mathfrak{B}_j(\mathbf{Y})$, then $\mathbf{X}$ is termed a (counterfactual) Markov boundary of $U_j$ on $\mathbf{Y}$, denoted as $\mathbf{B}_j(\mathbf{Y})$.

Methods for learning Markov boundaries based on independence in data distribution (e.g. [3, 87, 103]) cannot be directly applied to learn counterfactual Markov boundaries unless exogenous variables are observable. Moreover, these methods may have low efficiency due to their reliance on independence tests. To obtain the counterfactual Markov boundary, in this work, we assume that

another representation of SCM graphs called augmented graphs is known. Compared to causal graphs, augmented graphs incorporate exogenous variables $\mathbf{U}$ into the nodes, then remove bidirectional edges and add unidirectional edges $(U_j \rightarrow V_i)$ for all $V_i \in \mathbf{U}$ and $U_j \in \mathbf{U}_{V_i}$.

When the SCM is recursive, the augmented graph of any submodel is necessarily acyclic due to the closure of recursiveness under (perfect) intervention [8], which implies that d-separation [75] always holds. We additionally assume faithfulness [2], which means that only the variables that are d-separated in the graph are independent in the distribution, allowing us to directly compute counterfactual Markov boundary for any exogenous variable by graph algorithm.

Under the above assumptions, the following theorem indicates that the counterfactual Markov boundary of $U_j$ is independent across different submodels. This not only reduces the search space of the Markov boundary, but also aligns with Eq. 12, where we encode counterfactual information according to each submodel.

**Theorem 2** (Counterfactual Markov Boundary Independence)**.** If $\mathbf{Y}_* = \bigcup_{i=1}^{k} \mathbf{Y}_{i[\mathbf{x}_i]}$ and each $\mathbf{Y}_{i[\mathbf{x}_i]}$ corresponds to a different submodel $\mathcal{M}_{\mathbf{x}_i}$, then for each $U_j \in \mathbf{U}$, there exists a Markov boundary $\mathbf{B}_j(\mathbf{Y}_*) = \bigcup_{i=1}^{k} \mathbf{B}_j(\mathbf{Y}_{i[\mathbf{x}_i]})$ on $\mathbf{Y}_*$, where $\mathbf{B}_j(\mathbf{Y}_{i[\mathbf{x}_i]})$ is a Markov boundary on $\mathcal{M}_{\mathbf{x}_i}$.

Another theorem demonstrates how to obtain counterfactual Markov boundaries via d-separation, and indirectly proves their uniqueness:

**Theorem 3** (Counterfactual Markov Boundary on Graph)**.** For an exogenous variable $U_j \in \mathbf{U}$ and a counterfactual variable set $\mathbf{Y}_\mathbf{x}$ from the submodel $\mathcal{M}_\mathbf{x}$, the counterfactual variable $Y_\mathbf{x} \in \mathbf{B}_j(\mathbf{Y}_\mathbf{x})$ if and only if $Y_\mathbf{x} \not\perp\!\!\!\perp_{\mathcal{G}_\mathbf{x}^a} U_j \mid \mathbf{Y}_\mathbf{x} \setminus \{Y_\mathbf{x}\}$, i.e., when given $\mathbf{Y}_\mathbf{x} \setminus \{Y_\mathbf{x}\}$, $Y_\mathbf{x}$ and $U_j$ are not d-separated on $\mathcal{G}_\mathbf{x}^a$, where $\mathcal{G}_\mathbf{x}^a$ is the augmented graph induced from the submodel $\mathcal{M}_\mathbf{x}$.

The graph algorithm to determine counterfactual Markov boundaries $\mathbf{B}_j(\mathbf{Y}_*)$ for $U_j \in \mathbf{U}$ is derived simply by combining Thms. 2 and 3 with the d-separation criterion as a subroutine, which can be found in Fig. 7 in App. A.4.

**Masking**   We then inject the counterfactual Markov boundary into the neural network used for conditioning by vectorizing it as a weight mask, as briefly illustrated in Fig. 2 depicting the entire conditioning and masking process. Specifically, let $\theta_{\mathbf{y}_*,j}$ be any parameter corresponding to exogenous variable $U_j \in \mathbf{U}$ in $Q_{\mathbf{U}|\mathbf{y}_*}$. According to the description in Thm. 2, the Markov boundary on $\mathbf{Y}_* = \bigcup_{i=1}^{k} \mathbf{Y}_{i[\mathbf{x}_i]}$ can be precisely decomposed into the union of the Markov boundaries on each $\mathbf{Y}_{i[\mathbf{x}_i]}$. This corresponds exactly to encoding the information for counterfactual events on each submodel $\mathcal{M}_{\mathbf{x}_i}$ as depicted in Eq. 10. Therefore, we can modify Eq. 12 as follows:

$$\theta_{y_*,j} = g\left(\left\{ h_j(\mathbf{c}_{\langle \mathbf{Y}_i, \mathbf{y}_i, \mathbf{X}_i, \mathbf{x}_i \rangle}, \mathbf{m}_{ij}) \mid \mathbf{Y}_{i[\mathbf{x}_i]} \in \mathbf{Y}_* \right\}\right). \tag{13}$$

In particular, $\mathbf{m}_{ij} = \omega(\mathbf{B}_i(\mathbf{Y}_j))$, and $\mathbf{B}_i(\mathbf{Y}_j)$ are obtained through algorithm derived by Thm. 3. The correspondence between the parameters $\theta_{\mathbf{y}_*,j}$ and $U_j$ exists in the specific design of the model, such as the mean and covariance of each component in GMMs or the element-wise transformations in normalizing flows. As used in [31, 81, 13], $h$ is an MLP that allows weight masking, such that for all $j$, the $i$-th component of $\nabla_\mathbf{x} h_j(\mathbf{x}, \mathbf{m}_{ij}) \neq 0$ if and only if $\mathbf{m}_{ij} = 1$. This ensures that $\theta_{\mathbf{y}_*,j}$ is only related to the Markov boundary of $U_j$ on $\mathbf{y}_*$.

## 4   Related Works

**Importance sampling with normalizing flows**   In recent years, the combination of normalizing flows and importance sampling has been widely employed in tasks related to Monte Carlo integration. For instance, a plethora of literature [70, 99, 101, 53, 60, 10, 51, 67] utilizes this combination to efficiently compute the partition function of energy. Some studies [68, 30, 36] also employ this method to solve integration problems of complex functions in high-dimensional spaces. When the integrand involves rejection sampling [6, 89], particle events [29, 9, 88], or failure events [33, 20], this method can improve the efficiency of sampling rare events. This combination is also applied for posterior inference [19, 1, 79, 21] as an alternative to integrating over the intractable distribution.

**Identifiable neural proxy SCMs**   Our approach is applicable to pre-trained neural proxy SCMs, where causal mechanisms are modeled as neural networks. Increasing attention has been paid to the

identifiability of these models. BGM [69] combines bijections to demonstrate the identifiability of causal mechanisms in several special cases. CausalNF [42] builds upon the conclusions of [106] to prove the identifiability of its causal mechanisms up to invertible functions. NCM [108] extends the findings of their previous work [107] and proposes a sound and complete algorithm to identify counterfactual queries.

**Arbitrary conditioning**   In dealing with exponentially many conditional distributions, our work establishes a connection with another unsupervised learning task, known as arbitrary conditioning. The task of this paper can be viewed as extending arbitrary conditioning from a single observational distribution to multiple counterfactual distributions. Relevant works include VAE-AC [40] based on VAE, NC [7] based on GAN, ACE [90] based on energy models, and ACFlow [59] based on normalizing flows. The most similar to our work is [91], which matches on the posterior distribution of VAE, akin to our matching on the exogenous distribution.

## 5   Experiments

**Experiment settings and metrics**   We first define the prior distribution $Q_{\mathcal{S}}$ over the state space of a stochastic counterfactual process $\mathfrak{Y}_*$ and train according to Eq. 8. Subsequently, based on $Q_{\mathcal{S}}$, we construct a distribution $P_{\mathcal{Y}_*}$ concerning counterfactual events for the evaluation of the estimator (Eq. 4). Inspired by the metrics for rare event sampling effectiveness, we utilize Effective Sample Proportion (ESP) and Failure Rate (FR) with threshold $m$ to measure the performance of the importance sampling, defined as:

$$\text{ESP} = \mathbb{E}_{\mathcal{Y}_* \sim P_{\mathcal{Y}_*}} [\eta(\mathcal{Y}_*)] \quad \text{and} \quad \text{FR} = \mathbb{E}_{\mathcal{Y}_* \sim P_{\mathcal{Y}_*}} \left[ \begin{cases} 1, & \eta(\mathcal{Y}_*) \leq m \\ 0, & \text{otherwise} \end{cases} \right], \quad (14)$$

where $\eta(\mathcal{Y}_*) = \left( \sum_{i=1}^{n} \mathbb{1}_{\Omega_{\mathbf{U}}(\mathcal{Y}_*)}(\mathbf{u}^{(i)}) \right) / n$ is the proportion of effective samples (nonzero indicator) to estimate the probability $P(\mathcal{Y}_*)$, which indirectly reflects the sampling efficiency of a single estimate; this is equivalent to the success rate in rare event sampling. Therefore, ESP reflects the overall efficiency of the sampling method in supporting Thm. 1, while FR reflects its overall effectiveness in the state space to support Cor. 1 and Eq. 8. For further discussion on metrics, see App. C.1.

Specifically, the following two types of stochastic counterfactual processes are involved in subsequent experiments: i) $\mathfrak{Y}_*^{\mathcal{B}}$ with state space $\mathcal{S}^{\mathcal{B}} = \{s \mid |s| = k\}$ (that is, with $k$ submodels) and prior distribution $Q_{\mathcal{S}}^{\mathcal{B}}$, where the indicators for intervention and observation follow Bernoulli distributions, and their values follow endogenous distributions; ii) $\mathfrak{Y}_*^{\mathcal{Q}}$, where $\mathcal{Q} \in \{\text{ATE}, \text{ETT}, \text{NDE}, \text{CtfDE}\}$, the design of state space $\mathcal{S}^{\mathcal{Q}}$ and prior distribution $Q^{\mathcal{Q}}$ are detailed in App. C.2.

During evaluation, the distribution of counterfactual events $P_{\mathcal{Y}_*}$ is contingent upon the type of counterfactual variables: for discrete, we focus on counterfactual event $\mathcal{Y}_* \in \{\mathbf{y}_*\}$; for continuous, we focus on counterfactual event $\mathcal{Y}_* \in \delta_l(\mathbf{y}_*)$, where $\delta_l(\mathbf{y}_*)$ is a cube with side length $l = 0.02$ centered at $\mathbf{y}_*$. The latter can be further used to estimate the counterfactual density in continuous space, as discussed in App. D.2.

Our experiments involve the following fully specified SCMs, which fall into three categories: i) Markovian diffeomorphic [42], where there is no hidden confounder and causal mechanisms are diffeomorphic, including SIMPSON-NLIN, TRIANGLE-NLIN and LARGEBD-NLIN; ii) Semi-Markovian continuous [107], where hidden confounders may exist and endogenous variables are continuous, including M and NAPKIN; iii) Regional canonical [108], where hidden confounders may exist and endogenous variables are discrete and finite, including FAIRNESS and FAIRNESS-XW. The consistent performance observed in experiments across these different types of SCMs will serve as a reference for evaluating the robustness of the proposed method.

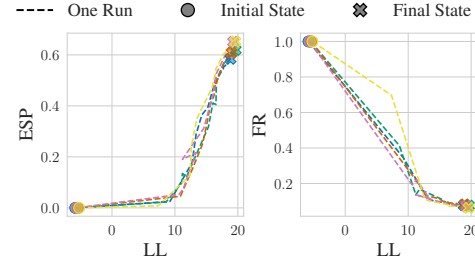

Figure 3: LL (negative Eq. 8) and ESP, FR on SIMPSON-NLIN. As LL increases, ESP increases while FR decreases, until convergence.

Table 1: Comparison of RS, CEIS, NIS, and EXOM (ours) across 3 different SCMs, with $|s| = 1, 3, 5$. Among the three SCMs, SIMPSON-NLIN is Markovian diffeomorphic, NAPKIN is Semi-Markovian continuous, and FAIRNESS-XW is Regional canonical. The results are averaged over 5 runs. Higher ESP and lower FR indicate better performance.

| $|s|$ | Model | SIMPSON-NLIN | | NAPKIN | | FAIRNESS-XW | |
|---|---|---|---|---|---|---|---|
| | | ESP ↑ | FR ↓ | ESP ↑ | FR ↓ | ESP ↑ | FR ↓ |
| 1 | RS | 0.008 | 0.971 | 0.006 | 0.967 | 0.156 | 0.065 |
| | CEIS | 0.037 | 0.809 | 0.017 | 0.911 | 0.255 | 0.532 |
| | NIS | 0.008 | 0.961 | 0.007 | 0.965 | 0.193 | 0.527 |
| | EXOM[GMM] | 0.117 | 0.245 | 0.039 | 0.476 | **0.358** | 0.009 |
| | EXOM[MAF] | **0.581** | **0.005** | **0.306** | **0.066** | 0.339 | **0.007** |
| 3 | RS | 0.000 | 1.000 | 0.000 | 1.000 | 0.038 | 0.281 |
| | CEIS | 0.000 | 1.000 | 0.000 | 1.000 | 0.122 | 0.704 |
| | NIS | 0.000 | 1.000 | 0.000 | 1.000 | 0.116 | 0.686 |
| | EXOM[GMM] | 0.024 | 0.500 | 0.011 | 0.613 | 0.247 | 0.056 |
| | EXOM[MAF] | **0.606** | **0.075** | **0.397** | **0.183** | **0.261** | **0.043** |
| 5 | RS | 0.000 | 1.000 | 0.000 | 1.000 | 0.030 | 0.368 |
| | CEIS | 0.000 | 1.000 | 0.000 | 1.000 | 0.106 | 0.727 |
| | NIS | 0.000 | 1.000 | 0.000 | 1.000 | 0.094 | 0.724 |
| | EXOM[GMM] | 0.020 | 0.497 | 0.009 | 0.644 | 0.231 | 0.084 |
| | EXOM[MAF] | **0.698** | **0.031** | **0.482** | **0.094** | **0.237** | **0.070** |

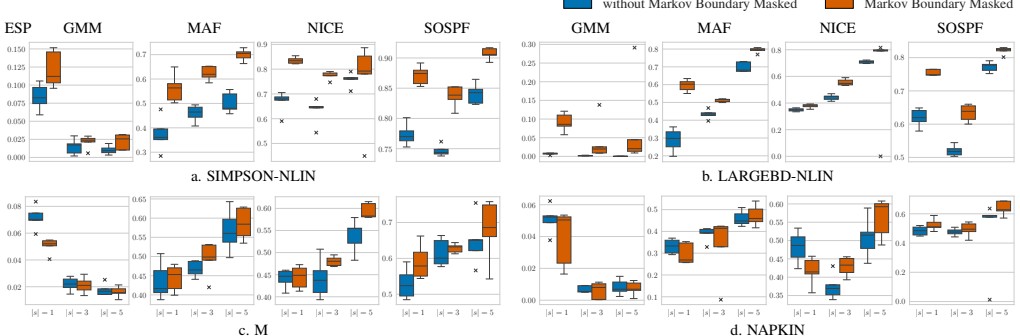

Figure 4: Ablation study for Markov boundaries on 4 different settings of SCMs: (a) SIMPSON-NLIN, (b) LARGEBD-NLIN, (c) M, (d) NAPKIN. A higher ESP signifies greater sampling efficiency. In most cases, EXOM with Markov boundaries masked (orange bar) exhibits superior performance compared to when the Markov boundaries are not masked (blue bar).

**Convergence** We integrated Exogenous Matching with various density estimation models (GMM, MAF [73], NICE [24], SOSPF [41]) to conduct experiments on the stochastic counterfactual process $\mathfrak{Y}_*^{\mathcal{B}}$ (with $|s| = 3$) over different SCM settings. We present the results of ESP and FR during the training process. As shown in Fig. 3 (with all results detailed in App. C.5), both the ESP and FR metrics change as expected until convergence, along with the decrease in training loss Eq. 8 (equivalent to the increase in LL in Fig. 3). This validates the effectiveness of Thm. 1 and Cor. 1, and empirically suggests that the conditions we assumed are generally not violated under our settings.

**Comparison** We compare our proposed method with three different but related sampling methods: i) Rejection Sampling (RS); ii) Cross-Entropy based Importance Sampling (CEIS, [32]); iii) Neural Importance Sampling (NIS, [68]). To apply the latter two methods under the same experimental setup, we extended them as detailed in App. B.1. As shown in Tab. 1, our method outperforms other approaches in both continuous and discrete cases. Specifically, since MAF exhibits stronger representational capabilities than GMM, it is expected that EXOM using MAF as the conditional distribution model generally outperforms EXOM using GMM.

Table 2: Estimation of counterfactual densities on CausalNF and counterfactual effects on NCM. Here, "O" represents the original SCM, and "P" represents the proxy SCM. For SIMPSON-NLIN, the proxy SCM is CausalNF, and the metric used is FR ("-" indicates FR equals 1); whereas for FAIRNESS, the proxy SCM is NCM, and the metric used is the average bias w.r.t. the ground truth. The subscript denotes the 95% CI error bound over 5 trials. For more details, see App. C.9

| Method | SCM | SIMPSON-NLIN | | | FAIRNESS | | | |
|---|---|---|---|---|---|---|---|---|
| | | $|s|=1$ | $|s|=3$ | $|s|=5$ | ATE | ETT | NDE | CtfDE |
| RS | O | $0.89_{0.014}$ | - | - | $0.01_{0.013}$ | $0.01_{0.018}$ | $0.01_{0.015}$ | $0.01_{0.020}$ |
| | P | $0.90_{0.012}$ | - | - | $0.01_{0.013}$ | $0.01_{0.021}$ | $0.01_{0.014}$ | $0.01_{0.023}$ |
| EXOM[MAF] | O | $0.00_{0.005}$ | $0.02_{0.015}$ | $0.01_{0.021}$ | $0.04_{0.095}$ | $0.06_{0.168}$ | $0.04_{0.131}$ | $0.07_{0.236}$ |
| | P | $0.01_{0.005}$ | $0.40_{0.012}$ | $0.61_{0.016}$ | $0.01_{0.013}$ | $0.01_{0.024}$ | $0.01_{0.013}$ | $0.01_{0.044}$ |
| EXOM[NICE] | O | $0.00_{0.006}$ | $0.02_{0.016}$ | $0.01_{0.046}$ | $0.01_{0.018}$ | $0.01_{0.030}$ | $0.01_{0.020}$ | $0.01_{0.039}$ |
| | P | $0.01_{0.005}$ | $0.40_{0.013}$ | $0.61_{0.018}$ | $0.01_{0.017}$ | $0.01_{0.021}$ | $0.01_{0.012}$ | $0.01_{0.022}$ |

**Ablation**   To investigate the impact of injecting Markov boundaries, we present the improvement of ESP after injecting Markov boundaries during the training process compared to the default scenario. We conducted ablation experiments on 4 continuous SCMs and various density estimators. As shown in Fig. 4, the use of Markov boundaries as masks significantly improves the performance of EXOM under various settings. We also observed that when masks are used, the width of the hidden layers of neural networks in conditioning affects practical performance, which will be discussed in App. C.8.

**Counterfactual Estimation on Proxy SCMs**   We combine identifiable proxy SCMs with EXOM for counterfactual estimation. We employ CausalNF [42] for SIMPSON-NLIN (based on the stochastic process $\mathfrak{Y}_*^{\mathcal{B}}$) and NCM [108] for FAIRNESS (based on the stochastic process $\mathfrak{Y}_*^{\mathcal{Q}}$). For the combination of CausalNF and SIMPSON-NLIN, we estimate the counterfactual probability on cubes $\delta_l(\mathbf{y}_*)$ with side length $l = 0.02$ centered around 1024 randomly sampled $\mathbf{y}_*$ from $P_{\mathcal{Y}_*}$ (this can be used further to estimate counterfactual density), measure the sampling FR, and employ a dimension-regularized 95% CI error bound. For the combination of NCM and FAIRNESS, we estimate 4 different counterfactual queries, measure the average bias of the query results relative to the ground truth, and use a 95% CI error bound to demonstrate the unbiasedness of the estimates.

The results in Tab. 2 empirically demonstrate the effectiveness of EXOM in counterfactual density and effect estimation tasks compared to RS. Specifically, when estimating the cube $\delta_l(\mathbf{y}_*)$ (equivalent to counterfactual density), RS almost fails in high-dimensional settings, while EXOM exhibits good sampling efficiency and lower error compared to RS. When estimating counterfactual queries, the bias and error of EXOM are similar to those of RS (partly due to the lower sampling difficulty in discrete cases). Further experimental details and analysis of the conclusions can be found in App. C.9.

# 6   Conclusion

We propose Exogenous Matching for tractable estimation of expressions in $\mathcal{L}_3$ in general settings. Specifically, leveraging importance sampling, we find a variance upper bound for estimators potentially applicable to any relevant counterfactual probability (Cor. 1), and then introduce an optimization objective (Eq. 8) amenable to sampling, transforming the problem into a conditional distribution learning problem. Theoretical and empirical results demonstrate that this approach can be combined with identifiable proxy SCMs to address practical problems. We also explore injecting Markov boundaries as prior knowledge and empirically validate effectiveness in several scenarios.

**Limitations**   i) The proposed method does not directly estimate counterfactuals through the available distribution but requires a partially specified SCM. This work explores its effectiveness in combination with identifiable proxy SCMs, which are still an active area of research; ii) It is important to emphasize that the "general case" mentioned in this paper does not cover all scenarios. For instance, our estimand, an $\mathcal{L}_3$ expression $\mathcal{Q}$, requires the assumptions of finite submodels and approximability by a finite number of counterfactual probabilities; iii) Furthermore, the specific structures and tricks demonstrated in this paper (such as the injection of Markov boundaries) may not be applicable to SCMs with general parameter settings, which warrants further investigation.

## Acknowledgments and Disclosure of Funding

This work was supported in part by National Key R&D Program of China (No. 2022ZD0120302).

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

# Appendix

## Table of Contents

# A  Proofs

## A.1  Counterfactuals

In this section, we demonstrate the validity of counterfactual probability-related concepts, providing intuitions based on derivations in discrete spaces, and proving lemmas to streamline subsequent derivations. Firstly, we establish the validity of counterfactual probabilities as defined in Eq. 2.

**Proposition 1.** The probability measure defined by Eq. 2 (on recursive SCMs) adheres to the Kolmogorov axioms.

*Proof.* (i) For any $\mathcal{Y}_* \in \Sigma_{\mathbf{Y}_*}$, $0 \leq P_{\mathbf{Y}_*}(\mathcal{Y}_*) \leq 1$. Since $0 \leq \mathbb{1}_{\Omega_{\mathbf{U}}(\mathcal{Y}_*)}(\mathbf{u}) \leq 1$, the proof follows from the Monotonicity of Lebesgue integration:

$$0 = \int_{\Omega_{\mathbf{U}}} 0 \, \mathrm{d}P_{\mathbf{U}} \leq \int_{\Omega_{\mathbf{U}}} \mathbb{1}_{\Omega_{\mathbf{U}}(\mathcal{Y}_*)}(\mathbf{u}) \, \mathrm{d}P_{\mathbf{U}} \leq \int_{\Omega_{\mathbf{U}}} 1 \, \mathrm{d}P_{\mathbf{U}} = 1, \tag{15}$$

where $P_{\mathbf{Y}_*}(\mathcal{Y}_*) = \int_{\Omega_{\mathbf{U}}} \mathbb{1}_{\Omega_{\mathbf{U}}(\mathcal{Y}_*)}(\mathbf{u}) \, \mathrm{d}P_{\mathbf{U}}$, thus $0 \leq P_{\mathbf{Y}_*}(\mathcal{Y}_*) \leq 1$.

(ii) $P_{\mathbf{Y}_*}(\Omega_{\mathbf{Y}_*}) = 1$. Since SCM is recursive, given any $\mathbf{u} \in \Omega_{\mathbf{U}}$, the values of exogenous variables are determined, and hence all values in $\mathbf{Y}_*$ are determined, that is, the potential responses $\mathbf{Y}_*(\mathbf{u})$. Since $\mathbf{Y}_*(\mathbf{u}) \in \Omega_{\mathbf{Y}_*}$, according to the definition of $\mathbb{1}_{\Omega_{\mathbf{U}}(\Omega_{\mathbf{Y}_*})}(\mathbf{u})$, $\mathbb{1}_{\Omega_{\mathbf{U}}(\Omega_{\mathbf{Y}_*})}(\mathbf{u}) = 1$, thus:

$$P_{\mathbf{Y}_*}(\Omega_{\mathbf{Y}_*}) = \int_{\Omega_{\mathbf{U}}} \mathbb{1}_{\Omega_{\mathbf{U}}(\Omega_{\mathbf{Y}_*})}(\mathbf{u}) \, \mathrm{d}P_{\mathbf{U}} = \int_{\Omega_{\mathbf{U}}} 1 \, \mathrm{d}P_{\mathbf{U}} = 1. \tag{16}$$

(iii) $P_{\mathbf{Y}_*}(\bigcup_{i \geq 1} \mathcal{Y}_*^{(i)}) = \sum_{i \geq 1} P_{\mathbf{Y}_*}(\mathcal{Y}_*^{(i)})$, where $\{\mathcal{Y}_*^{(1)}, \mathcal{Y}_*^{(2)}, \dots\}$ is a countable (possibly countably infinite) set of pairwise disjoint counterfactual events. For $\mathbf{u} \in \Omega_{\mathbf{U}}$, if $\mathbf{Y}_*(\mathbf{u}) \in \mathcal{Y}_*^{(i)}$, since for any $j \neq i$, $\mathcal{Y}_*^{(i)} \cap \mathcal{Y}_*^{(j)} = \emptyset$, it follows that $\mathbf{Y}_*(\mathbf{u}) \notin \mathcal{Y}_*^{(j)}$, thus $\mathbb{1}_{\Omega_{\mathbf{U}}(\mathcal{Y}_*^{(i)})}(\mathbf{u}) = \mathbb{1}_{\Omega_{\mathbf{U}}(\bigcup_{i \geq 1} \mathcal{Y}_*^{(i)})}(\mathbf{u}) = 1$ and $\sum_{j \neq i} \mathbb{1}_{\Omega_{\mathbf{U}}(\mathcal{Y}_*^{(i)})}(\mathbf{u}) = 0$, hence.

$$P_{\mathbf{Y}_*}(\bigcup_{i \geq 1} \mathcal{Y}_*^{(i)}) = \int_{\Omega_{\mathbf{U}}} \mathbb{1}_{\Omega_{\mathbf{U}}(\bigcup_{i \geq 1} \mathcal{Y}_*^{(i)})}(\mathbf{u}) \, \mathrm{d}P_{\mathbf{U}} \qquad \text{Definition} \tag{17}$$

$$= \int_{\Omega_{\mathbf{U}}} \left( \mathbb{1}_{\Omega_{\mathbf{U}}(\mathcal{Y}_*^{(i)})}(\mathbf{u}) + \sum_{j \neq i} \mathbb{1}_{\Omega_{\mathbf{U}}(\mathcal{Y}_*^{(j)})}(\mathbf{u}) \right) \mathrm{d}P_{\mathbf{U}} \tag{18}$$

$$= \int_{\Omega_{\mathbf{U}}} \mathbb{1}_{\Omega_{\mathbf{U}}(\mathcal{Y}_*^{(i)})}(\mathbf{u}) \, \mathrm{d}P_{\mathbf{U}} + \int_{\Omega_{\mathbf{U}}} \sum_{j \neq i} \mathbb{1}_{\Omega_{\mathbf{U}}(\mathcal{Y}_*^{(j)})}(\mathbf{u}) \, \mathrm{d}P_{\mathbf{U}} \qquad \text{Linearity} \tag{19}$$

$$= \int_{\Omega_{\mathbf{U}}} \mathbb{1}_{\Omega_{\mathbf{U}}(\mathcal{Y}_*^{(i)})}(\mathbf{u}) \, \mathrm{d}P_{\mathbf{U}} + 0 \qquad \text{Zero measure} \tag{20}$$

$$= \int_{\Omega_{\mathbf{U}}} \mathbb{1}_{\Omega_{\mathbf{U}}(\mathcal{Y}_*^{(i)})}(\mathbf{u}) \, \mathrm{d}P_{\mathbf{U}} + \sum_{j \neq i} \int_{\Omega_{\mathbf{U}}} \mathbb{1}_{\Omega_{\mathbf{U}}(\mathcal{Y}_*^{(j)})}(\mathbf{u}) \, \mathrm{d}P_{\mathbf{U}} \tag{21}$$

$$= \sum_{i \geq 1} \int_{\Omega_{\mathbf{U}}} \mathbb{1}_{\Omega_{\mathbf{U}}(\mathcal{Y}_*^{(i)})}(\mathbf{u}) \, \mathrm{d}P_{\mathbf{U}} \tag{22}$$

$$= \sum_{i \geq 1} P_{\mathbf{Y}_*}(\mathcal{Y}_*^{(i)}) \tag{23}$$

If for all $i$, $\mathbf{Y}_*(\mathbf{u}) \notin \mathcal{Y}_*^{(i)}$, then $\sum_{i \geq 1} \mathbb{1}_{\Omega_{\mathbf{U}}(\mathcal{Y}_*^{(i)})}(\mathbf{u}) = \mathbb{1}_{\Omega_{\mathbf{U}}(\bigcup_{i \geq 1} \mathcal{Y}_*^{(i)})}(\mathbf{u}) = 0$. Therefore, both sides of the equation are zero, so it holds. Hence, we have proved $P_{\mathbf{Y}_*}(\bigcup_{i \geq 1} \mathcal{Y}_*^{(i)}) = \sum_{i \geq 1} P_{\mathbf{Y}_*}(\mathcal{Y}_*^{(i)})$.

Given that axioms (i), (ii) and (iii) hold, which are the Kolmogorov axioms, it follows that Eq. 2 satisfies the definition of a probability measure. $\square$

From a measure-theoretic perspective, the indicator function in Eq. 2 can be transformed into the Dirac measure, and vice versa:

**Lemma 1.** For any $\mathbf{Y}_* \in \Sigma_{\mathbf{Y}_*}$ and any $\mathbf{u} \in \Omega_{\mathbf{U}}$,

$$\mathbb{1}_{\Omega_{\mathbf{U}}(\mathcal{Y}_*)}(\mathbf{u}) = \delta_{\mathbf{Y}_*(\mathbf{u})}(\mathcal{Y}_*) = \int_{\mathcal{Y}_*} \mathrm{d}\delta_{\mathbf{Y}_*(\mathbf{u})}. \tag{24}$$

*Proof.* Given $\mathbf{Y}_* \in \Sigma_{\mathbf{Y}_*}$ and $\mathbf{u} \in \Omega_{\mathbf{U}}$, we can consider $\mathbf{Y}_*$ as $\mathbf{u}$ being constant, thus

$$
\begin{align}
\mathbb{1}_{\Omega_{\mathbf{U}}(\mathcal{Y}_*)}(\mathbf{u}) &= \mathbb{1}_{\{\mathbf{u}|\mathbf{Y}_*(\mathbf{u}) \in \mathcal{Y}_*\}}(\mathbf{u}) && \text{Definition of } \Omega_{\mathbf{U}}(\mathcal{Y}_*) \tag{25}\\
&= \mathbb{1}_{\mathcal{Y}_*}(\mathbf{Y}_*(\mathbf{u})) && \text{Let } \mathbf{y}_* = \mathbf{Y}_*(\mathbf{u}) \tag{26}\\
&= \delta_{\mathbf{Y}_*(\mathbf{u})}(\mathcal{Y}_*) && \text{Definition of } \delta_{\mathbf{Y}_*(\mathbf{u})} \tag{27}\\
&= \int_{\Omega_{\mathbf{Y}_*}} \mathbb{1}_{\mathcal{Y}_*}(\mathbf{y}_*)\, \mathrm{d}\delta_{\mathbf{Y}_*(\mathbf{u})} && \text{Integral w.r.t. Dirac measure} \tag{28}\\
&= \int_{\mathcal{Y}_*} \mathrm{d}\delta_{\mathbf{Y}_*(\mathbf{u})}, \tag{29}
\end{align}
$$

where the integral $\int_{\mathcal{Y}_*} \mathrm{d}\delta_{\mathbf{Y}_*(\mathbf{u})}$ remains a function of $\mathbf{u}$. $\qquad\square$

Then, substituting Lem. 1 into Eq. 2, we have:

$$
\begin{align}
P(\mathcal{Y}_*) &= \int_{\Omega_{\mathbf{U}}} \mathbb{1}_{\Omega_{\mathbf{U}}(\mathcal{Y}_*)}(\mathbf{u})\, \mathrm{d}P_{\mathbf{U}} \tag{2}\\
&= \int_{\Omega_{\mathbf{U}}} \int_{\mathcal{Y}_*} \mathrm{d}\delta_{\mathbf{Y}_*(\mathbf{u})}\, \mathrm{d}P_{\mathbf{U}} && \text{Lem. 1} \tag{30}\\
&= \int_{\Omega_{\mathbf{U}}} \int_{\mathcal{Y}_*} \mathrm{d}P_{\mathbf{Y}_*,\mathbf{U}}, \tag{31}
\end{align}
$$

where $P_{\mathbf{Y}_*,\mathbf{U}}$ satisfying $\mathrm{d}P_{\mathbf{Y}_*,\mathbf{U}} = \mathrm{d}\delta_{\mathbf{Y}_*(\mathbf{u})}\, \mathrm{d}P_{\mathbf{U}}$ is called the joint distribution of counterfactual variables and exogenous variables, which we abbreviate as the joint distribution.

In words, the joint distribution is determined by the exogenous distribution and the Dirac distribution, while the counterfactual probability is the integral over the endogenous space after marginalizing out the exogenous variables from the joint distribution.

### A.2 Importance Sampling

**Importance sampling estimator for counterfactual estimation**   Eq. 4 only illustrates the case where exogenous variables are continuous. Here, we will provide a complete derivation. For discrete cases, the Lebesgue integral can be expressed as a summation, thus,

$$
\begin{align}
P(\mathcal{Y}_*) &= \int_{\Omega_{\mathbf{U}}} \mathbb{1}_{\Omega_{\mathbf{U}}(\mathcal{Y}_*)}(\mathbf{u})\, \mathrm{d}P_{\mathbf{U}} \tag{32}\\
&= \sum_{\mathbf{u} \in \Omega_{\mathbf{U}}} P(\mathbf{u})\, \mathbb{1}_{\Omega_{\mathbf{U}}(\mathcal{Y}_*)}(\mathbf{u}) \tag{33}\\
&= \sum_{\mathbf{u} \in \Omega_{\mathbf{U}}} Q(\mathbf{u}) \frac{P(\mathbf{u})}{Q(\mathbf{u})} \mathbb{1}_{\Omega_{\mathbf{U}}(\mathcal{Y}_*)}(\mathbf{u}) \tag{34}\\
&= \mathbb{E}_{\mathbf{u} \sim Q_{\mathbf{U}}} \left[ \frac{P(\mathbf{u})}{Q(\mathbf{u})} \mathbb{1}_{\Omega_{\mathbf{U}}(\mathcal{Y}_*)}(\mathbf{u}) \right]. \tag{35}
\end{align}
$$

For continuous cases, the Radon–Nikodym theorem is required. Suppose that the probability density functions of $P_{\mathbf{U}}$ and $Q_{\mathbf{U}}$ exist, denoted by $p(\mathbf{u})$ and $q(\mathbf{u})$, respectively. According to the definition of probability density functions, $p(\mathbf{u}) = \mathrm{d}P_{\mathbf{U}}/\mathrm{d}\mathbf{u}$ and $q(\mathbf{u}) = \mathrm{d}Q_{\mathbf{U}}/\mathrm{d}\mathbf{u}$. Then

$$
\begin{align}
P(\mathcal{Y}_*) &= \int_{\Omega_{\mathbf{U}}} \mathbb{1}_{\Omega_{\mathbf{U}}(\mathcal{Y}_*)}(\mathbf{u})\, \mathrm{d}P_{\mathbf{U}} \tag{36}\\
&= \int_{\Omega_{\mathbf{U}}} \mathbb{1}_{\Omega_{\mathbf{U}}(\mathcal{Y}_*)}(\mathbf{u}) \frac{\mathrm{d}P_{\mathbf{U}}/\mathrm{d}\mathbf{u}}{\mathrm{d}Q_{\mathbf{U}}/\mathrm{d}\mathbf{u}}\, \mathrm{d}Q_{\mathbf{U}} \tag{37}
\end{align}
$$

$$= \int_{\Omega_{\mathbf{U}}} \frac{p(\mathbf{u})}{q(\mathbf{u})} \mathbb{1}_{\Omega_{\mathbf{U}}(\mathcal{Y}_*)}(\mathbf{u}) \, \mathrm{d}Q_{\mathbf{U}} \tag{38}$$

$$= \mathbb{E}_{\mathbf{u} \sim Q_{\mathbf{U}}} \left[ \frac{p(\mathbf{u})}{q(\mathbf{u})} \mathbb{1}_{\Omega_{\mathbf{U}}(\mathcal{Y}_*)}(\mathbf{u}) \right]. \tag{39}$$

We assume that the integrand is Lebesgue integrable. It can be observed that the difference between the discrete and continuous cases lies only in the importance weights. Therefore, in subsequent derivations, we will proceed in the continuous case, with the discrete case requiring only the substitution of the importance weights.

The expectation term can be estimated using Monte Carlo methods, resulting in the basic form of the importance sampling estimator:

$$P(\mathcal{Y}_*) = \mathbb{E}_{\mathbf{u} \sim Q_{\mathbf{U}}} [\sigma_{\mathcal{Y}_*}(\mathbf{u})] \approx \frac{1}{n} \sum_{i=1}^{n} \sigma_{\mathcal{Y}_*}(\mathbf{u}^{(i)}), \tag{4}$$

where $\sigma_{\mathcal{Y}_*}(\mathbf{u}) = (p(\mathbf{u})/q(\mathbf{u})) \mathbb{1}_{\Omega_{\mathbf{U}}(\mathcal{Y}_*)}(\mathbf{u})$, and each sample $\mathbf{u}^{(i)} \sim P_{\mathbf{U}}$.

**Variance of the importance sampling estimator**    The variance of the estimator in Eq. 4 is:

$$\mathbb{V} \left[ \frac{1}{n} \sum_{i=1}^{n} \sigma_{\mathcal{Y}_*}(\mathbf{u}^{(i)}) \right] = \frac{1}{n^2} \sum_{i=1}^{n} \mathbb{V}_{\mathbf{u}^{(i)} \sim Q_{\mathbf{U}}} \left[ \sigma_{\mathcal{Y}_*}(\mathbf{u}^{(i)}) \right] \qquad \text{Linearity} \tag{40}$$

$$= \frac{n}{n^2} \mathbb{V}_{\mathbf{u}^{(i)} \sim Q_{\mathbf{U}}} [\sigma_{\mathcal{Y}_*}(\mathbf{u})] \qquad \text{i.i.d} \tag{41}$$

$$= \frac{1}{n} \mathbb{V}_{\mathbf{u} \sim Q_{\mathbf{U}}} [\sigma_{\mathcal{Y}_*}(\mathbf{u})]. \qquad \text{(5, the first equation)}$$

$\mathbb{V}_{\mathbf{u} \sim Q_{\mathbf{U}}} [\sigma_{\mathcal{Y}_*}(\mathbf{u})]$ can be further expanded as:

$$\mathbb{V}_{\mathbf{u} \sim Q_{\mathbf{U}}} [\sigma_{\mathcal{Y}_*}(\mathbf{u})] = \mathbb{V}_{\mathbf{u} \sim Q_{\mathbf{U}}} \left[ \frac{p(\mathbf{u})}{q(\mathbf{u})} \mathbb{1}_{\Omega_{\mathbf{U}}(\mathcal{Y}_*)}(\mathbf{u}) \right] \tag{42}$$

$$= \mathbb{E}_{\mathbf{u} \sim Q_{\mathbf{U}}} \left[ \left( \frac{p(\mathbf{u})}{q(\mathbf{u})} \mathbb{1}_{\Omega_{\mathbf{U}}(\mathcal{Y}_*)}(\mathbf{u}) \right)^2 \right] - \mathbb{E}_{\mathbf{u} \sim Q_{\mathbf{U}}} \left[ \frac{p(\mathbf{u})}{q(\mathbf{u})} \mathbb{1}_{\Omega_{\mathbf{U}}(\mathcal{Y}_*)}(\mathbf{u}) \right] \tag{43}$$

$$= \mathbb{E}_{\mathbf{u} \sim Q_{\mathbf{U}}} \left[ \frac{p^2(\mathbf{u})}{q^2(\mathbf{u})} \mathbb{1}_{\Omega_{\mathbf{U}}(\mathcal{Y}_*)}(\mathbf{u}) \right] - P^2(\mathcal{Y}_*) \tag{44}$$

$$= \int_{\Omega_{\mathbf{U}}} \left( \frac{p(\mathbf{u})}{q(\mathbf{u})} \right) \left( \frac{\mathrm{d}P_{\mathbf{U}}}{\mathrm{d}Q_{\mathbf{U}}} \right) \mathbb{1}_{\Omega_{\mathbf{U}}(\mathcal{Y}_*)}(\mathbf{u}) \, \mathrm{d}Q_{\mathbf{U}} - P^2(\mathcal{Y}_*) \tag{45}$$

$$= \int_{\Omega_{\mathbf{U}}} \frac{p(\mathbf{u})}{q(\mathbf{u})} \mathbb{1}_{\Omega_{\mathbf{U}}(\mathcal{Y}_*)}(\mathbf{u}) \, \mathrm{d}P_{\mathbf{U}} - P^2(\mathcal{Y}_*) \tag{46}$$

$$= \mathbb{E}_{\mathbf{u} \sim P_{\mathbf{U}}} \left[ \frac{p(\mathbf{u})}{q(\mathbf{u})} \mathbb{1}_{\Omega_{\mathbf{U}}(\mathcal{Y}_*)}(\mathbf{u}) \right] - P^2(\mathcal{Y}_*) \tag{47}$$

$$= \mathbb{E}_{\mathbf{u} \sim P_{\mathbf{U}}} [\sigma_{\mathcal{Y}_*}(\mathbf{u})] - P^2(\mathcal{Y}_*). \qquad \text{(5, the second equation)}$$

The proposal distribution $Q_{\mathbf{U}}$ only affects the term $\mathbb{E}_{\mathbf{u} \sim P_{\mathbf{U}}} [\sigma_{\mathcal{Y}_*}(\mathbf{u})]$ in the variance Eq. 5. The only difference between it and the estimator $P(\mathcal{Y}_*) = \mathbb{E}_{\mathbf{u} \sim Q_{\mathbf{U}}} [\sigma_{\mathcal{Y}_*}(\mathbf{u})]$ is that $\mathbb{E}_{\mathbf{u} \sim P_{\mathbf{U}}} [\sigma_{\mathcal{Y}_*}(\mathbf{u})]$ is the expectation over the exogenous distribution $P_{\mathbf{U}}$.

**Optimization of Eq. 5 and the limitations**    Reducing variance implies improving estimation efficiency; in other words, it means achieving the same level of estimation accuracy with fewer samples. One direct method to optimize variance is by sampling from the exogenous distribution $P_{\mathbf{U}}$ using Monte Carlo methods to estimate the term $\mathbb{E}_{\mathbf{u} \sim P_{\mathbf{U}}} [\sigma_{\mathcal{Y}_*}(\mathbf{u})]$ in the variance Eq. 5, and then minimizing the estimated value of this term using stochastic optimization techniques. Alternatively, as suggested by [32, 68], this term can be transformed into an optimization problem involving the cross-entropy or $\chi^2$ divergence between the proposal distribution and the integrand, and then optimizing the values of these mismatches estimated by sampling under the proposal distribution $Q_{\mathbf{U}}$ can also indirectly optimize variance. This optimization process is referred to as proposal learning.

However, a learned individual proposal distribution is typically only applicable to a single estimator, meaning it can only estimate the probability for a single counterfactual event $\mathcal{Y}_*$, as the indicator function $\mathbb{1}_{\Omega_{\mathbf{U}}(\mathcal{Y}_*)}(\mathbf{u})$ in $\sigma_{\mathcal{Y}_*}(\mathbf{u})$ operates solely for $\mathcal{Y}_*$. But $\mathcal{L}_3$ expressions can typically contain multiple counterfactual probability terms, and this limitation implies that, in the worst case, we need to perform a separate optimization for each counterfactual probability, still affecting the efficiency of the counterfactual estimation task.

To address this issue, our solution is to replace the sampling proposal distribution $Q_{\mathbf{U}}$ with a conditional proposal distribution $Q_{\mathbf{U}|\mathbf{Y}_*}$, adding an extra degree of freedom conditioned on $\mathbf{Y}_*$. Our overall idea is to let these different proposal distributions participate in the same optimization (i.e., sharing the same optimization objective) so that a single optimization yields different proposal distributions applicable to different counterfactual events.

Additionally, to limit the scope of responsibility of each proposal distribution $Q_{\mathbf{U}|\mathbf{y}_*}$ — that is, to prevent too many proposal distributions from targeting the same counterfactual event $\mathcal{Y}_*$ and consequently neglecting some counterfactual events — we further impose a one-to-one correspondence between conditions and responsibilities for inference. Intuitively, this would make the proposal distribution $Q_{\mathbf{U}|\mathbf{y}_*}$ corresponding to $\mathbf{y}_* \in \Omega_{\mathbf{Y}_*}$ focus on the support $\Omega_{\mathbf{U}}(\delta(\mathbf{y}_*))$. Given that for any counterfactual event there is an approximation $\mathcal{Y}_* \approx \bigcup_{\mathbf{y}_* \in \mathcal{Y}_*}^{\infty} \delta(\mathbf{y}_*)$, we can use the proposal distributions $Q_{\mathbf{U}|\mathbf{y}_*}$ corresponding to these $\mathbf{y}_* \in \mathcal{Y}_*$ to serve the estimation of $P(\mathcal{Y}_*)$.

**Variance upper bound**  When certain conditions are met in $\mathcal{Y}_*$, for any $\mathbf{y}_* \in \mathcal{Y}_*$, the proposal distribution $Q_{\mathbf{U}|\mathbf{y}_*}$ shares a common upper bound on variance, which is exactly the common optimization objective we seek:

**Theorem 1** (Variance Upper Bound). *Let* $\sigma_{\mathcal{Y}_*}(\mathbf{u}) = (p(\mathbf{u})/q(\mathbf{u}|\mathbf{y}_*)) \, \mathbb{1}_{\Omega_{\mathbf{U}}(\mathcal{Y}_*)}(\mathbf{u})$*, where* $q(\mathbf{u}|\mathbf{y}_*)$ *denote the density of the proposal distribution* $Q_{\mathbf{U}|\mathbf{y}_*}$*, and let* $\mathbf{Y}_*(\mathbf{u})$ *be the potential response w.r.t.* $\mathbf{u}$*. If for any* $\mathbf{y}_* \in \mathcal{Y}_*$*, there exists* $\kappa \geq 1$ *such that* $1/\kappa \leq p(\mathbf{u})/q(\mathbf{u}|\mathbf{y}_*) \leq \kappa$ *holds almost surely on the support* $\Omega_{\mathbf{U}}(\mathcal{Y}_*)$*, then for any* $Q_{\mathbf{U}|\mathbf{y}_*}$ *where* $\mathbf{y}_* \in \mathcal{Y}_*$*:*

$$\mathbb{V}_{\mathbf{u} \sim Q_{\mathbf{U}|\mathbf{y}_*}} [\sigma_{\mathcal{Y}_*}(\mathbf{u})] \leq -\mathbb{E}_{\mathbf{u} \sim P_{\mathbf{U}}} [\log q(\mathbf{u}|\mathbf{Y}_*(\mathbf{u}))] + c, \tag{6}$$

*where the constant* $c$ *is solely dependent on* $\kappa$ *and* $P_{\mathbf{U}}$*.*

*Proof.* For a $\mathcal{Y}_* \in \Sigma_{\mathbf{Y}_*}$ that satisfies bounded importance weights (i.e., there exists $\kappa \geq 1$ such that $1/\kappa \leq p(\mathbf{u})/q(\mathbf{u}|\mathbf{y}_*) \leq \kappa$ holds almost surely on the support $\Omega_{\mathbf{U}}(\mathcal{Y}_*)$), the following inequality holds for any $\mathbf{y}_*, \mathbf{y}_*' \in \mathcal{Y}_*$ holds almost surely:

$$\frac{q(\mathbf{u}|\mathbf{y}_*)}{q(\mathbf{u}|\mathbf{y}_*')} = \frac{q(\mathbf{u}|\mathbf{y}_*)}{p(\mathbf{u})} \cdot \frac{p(\mathbf{u})}{q(\mathbf{u}|\mathbf{y}_*')} \geq \frac{1}{\kappa^2}. \tag{48}$$

In the following derivations, cases where Eq. 48 does not hold will be implicitly omitted. This is because, according to the assumption of bounded importance weights, the exogenous distribution's probability measure corresponding to the cases where Eq. 48 does not hold is zero (in other words, the probability that the inequality Eq. 48 holds is 1). Therefore, omitting the cases where it does not hold does not affect the expectation w.r.t. the exogenous distribution.

For ease of implementation and computation, we focus on log-weights by constructing $\xi_{\mathcal{Y}_*}(\mathbf{u})$ to bridge the inequality conclusions over $\mathbb{E}_{\mathbf{u} \sim P_{\mathbf{U}}} [\xi_{\mathbf{y}_*}(\mathbf{u})]$ and the inequality relationship with the variance, where $\xi_{\mathcal{Y}_*}(\mathbf{u}) = \log(p(\mathbf{u})/q(\mathbf{u}|\mathbf{y}_*)) \mathbb{1}_{\Omega_{\mathbf{U}}(\mathcal{Y}_*)}(\mathbf{u})$.

$$\mathbb{E}_{\mathbf{u} \sim P_{\mathbf{U}}} [\xi_{\mathcal{Y}_*}(\mathbf{u})] \tag{49}$$

$$= \mathbb{E}_{\mathbf{u} \sim P_{\mathbf{U}}} \left[ \log \left( \frac{p(\mathbf{u})}{q(\mathbf{u}|\mathbf{y}_*)} \right) \mathbb{1}_{\Omega_{\mathbf{U}}(\mathcal{Y}_*)}(\mathbf{u}) \right] \tag{50}$$

$$= \mathbb{E}_{\mathbf{u} \sim P_{\mathbf{U}}} \left[ \log \left( \kappa^2 \cdot \frac{1}{\kappa^2} \cdot \frac{p(\mathbf{u})}{q(\mathbf{u}|\mathbf{y}_*)} \right) \mathbb{1}_{\Omega_{\mathbf{U}}(\mathcal{Y}_*)}(\mathbf{u}) \right] \quad \kappa \text{ existence} \tag{51}$$

$$\leq \mathbb{E}_{\mathbf{u} \sim P_{\mathbf{U}}} \left[ \log \left( \kappa^2 \left( \frac{p(\mathbf{u})}{q(\mathbf{u}|\mathbf{Y}_*(\mathbf{u}))} \cdot \frac{q(\mathbf{u}|\mathbf{y}_*)}{p(\mathbf{u})} \right) \cdot \frac{p(\mathbf{u})}{q(\mathbf{u}|\mathbf{y}_*)} \right) \mathbb{1}_{\Omega_{\mathbf{U}}(\mathcal{Y}_*)}(\mathbf{u}) \right] \quad \text{Eq. 48} \tag{52}$$

$$= \mathbb{E}_{\mathbf{u} \sim P_{\mathbf{U}}} \left[ \log \left( \kappa^2 \cdot \frac{p(\mathbf{u})}{q(\mathbf{u}|\mathbf{Y}_*(\mathbf{u}))} \right) \mathbb{1}_{\Omega_{\mathbf{U}}(\mathcal{Y}_*)}(\mathbf{u}) \right] \tag{53}$$

$$\leq \mathbb{E}_{\mathbf{u} \sim P_{\mathbf{U}}} \left[ \log \left( \kappa^2 \cdot \frac{p(\mathbf{u})}{q(\mathbf{u} \mid \mathbf{Y}_*(\mathbf{u}))} \right) \right] \qquad \text{Monotonicity} \qquad (54)$$

$$= - \mathbb{E}_{\mathbf{u} \sim P_{\mathbf{U}}} \left[ \log q(\mathbf{u} \mid \mathbf{Y}_*(\mathbf{u}) \right] + \mathbb{E}_{\mathbf{u} \sim P_{\mathbf{U}}} \left[ \log p(\mathbf{u}) \right] + 2 \log \kappa. \qquad \text{Linearity} \qquad (55)$$

Next, we look for a lower bound for $\mathbb{E}_{\mathbf{u} \sim P_{\mathbf{U}}} \left[ \xi_{\mathcal{Y}_*}(\mathbf{u}) \right]$. Firstly, according to the bounded density ratio condition, for any $\mathbf{u} \in \Omega_{\mathbf{U}}(\mathcal{Y}_*)$ with $p(\mathbf{u}) > 0$ and $\mathbf{y}_* \in \mathcal{Y}_*$, $\kappa^{-1} \leq p(\mathbf{u})/q(\mathbf{u} \mid \mathbf{y}_*) \leq \kappa$ holds. Therefore, based on the concavity of the logarithmic function, the following inequality holds (assuming $\kappa > 1$, as the conclusion is trivially true when $\kappa = 1$):

$$\log \left( \frac{p(\mathbf{u})}{q(\mathbf{u} \mid \mathbf{y}_*)} \right) \geq \frac{2\kappa \log \kappa}{\kappa^2 - 1} \left( \frac{p(\mathbf{u})}{q(\mathbf{u} \mid \mathbf{y}_*)} - \frac{\kappa^2 + 1}{2\kappa} \right). \qquad (56)$$

Due to the boundedness of $\mathbb{E}_{\mathbf{u} \sim P_{\mathbf{U}}} \left[ \xi_{\mathcal{Y}_*}(\mathbf{u}) \right]$ in the logarithmic space (with $\leq \log \kappa$ and $\geq -\log \kappa$), by employing the monotonicity and linearity of the Lebesgue integral, we obtain:

$$\mathbb{E}_{\mathbf{u} \sim P_{\mathbf{U}}} \left[ \xi_{\mathcal{Y}_*}(\mathbf{u}) \right] \qquad (57)$$

$$\geq \frac{2\kappa \log \kappa}{\kappa^2 - 1} \cdot \mathbb{E}_{\mathbf{u} \sim P_{\mathbf{U}}} \left[ \left( \frac{p(\mathbf{u})}{q(\mathbf{u} \mid \mathbf{y}_*)} - \frac{\kappa^2 + 1}{2\kappa} \right) \mathbb{1}_{\Omega_{\mathbf{U}}(\mathcal{Y}_*)}(\mathbf{u}) \right] \qquad (58)$$

$$= \frac{2\kappa \log \kappa}{\kappa^2 - 1} \left( \mathbb{E}_{\mathbf{u} \sim P_{\mathbf{U}}} \left[ \frac{p(\mathbf{u})}{q(\mathbf{u} \mid \mathbf{y}_*)} \mathbb{1}_{\Omega_{\mathbf{U}}(\mathcal{Y}_*)}(\mathbf{u}) \right] - \frac{\kappa^2 + 1}{2\kappa} \cdot \mathbb{E}_{\mathbf{u} \sim P_{\mathbf{U}}} \left[ \mathbb{1}_{\Omega_{\mathbf{U}}(\mathcal{Y}_*)}(\mathbf{u}) \right] \right) \qquad (59)$$

$$\geq \frac{2\kappa \log \kappa}{\kappa^2 - 1} \cdot \mathbb{E}_{\mathbf{u} \sim P_{\mathbf{U}}} \left[ \sigma_{\mathcal{Y}_*}(\mathbf{u}) \right] - \frac{(\kappa^2 + 1) \log \kappa}{\kappa^2 - 1}. \qquad (60)$$

Substitute the above inequalities into Eq. 5:

$$\mathbb{V}_{\mathbf{u} \sim Q_{\mathbf{U} \mid \mathbf{y}_*}} \left[ \sigma_{\mathcal{Y}_*}(\mathbf{u}) \right] \qquad (61)$$

$$= \mathbb{E}_{\mathbf{u} \sim P_{\mathbf{U}}} \left[ \sigma_{\mathcal{Y}_*}(\mathbf{u}) \right] - P^2(\mathcal{Y}_*) \qquad \text{Eq. 5} \qquad (62)$$

$$\leq \mathbb{E}_{\mathbf{u} \sim P_{\mathbf{U}}} \left[ \sigma_{\mathcal{Y}_*}(\mathbf{u}) \right] \qquad \text{Non-negativity} \qquad (63)$$

$$\leq \frac{\kappa^2 - 1}{2\kappa \log \kappa} \mathbb{E}_{\mathbf{u} \sim P_{\mathbf{U}}} \left[ \xi_{\mathcal{Y}_*}(\mathbf{u}) \right] + \frac{\kappa^2 + 1}{2\kappa} \qquad \text{Eq. 60} \qquad (64)$$

$$\leq - \frac{\kappa^2 - 1}{2\kappa \log \kappa} \mathbb{E}_{\mathbf{u} \sim P_{\mathbf{U}}} \left[ \log q(\mathbf{u} \mid \mathbf{Y}_*(\mathbf{u})) \right] \qquad \text{Eq. 55} \qquad (65)$$

$$+ \frac{\kappa^2 - 1}{2\kappa \log \kappa} \mathbb{E}_{\mathbf{u} \sim P_{\mathbf{U}}} \left[ \log p(\mathbf{u}) \right] + \frac{3\kappa^2 - 1}{2\kappa}$$

$$= - \frac{\kappa^2 - 1}{2\kappa \log \kappa} \mathbb{E}_{\mathbf{u} \sim P_{\mathbf{U}}} \left[ \log q(\mathbf{u} \mid \mathbf{Y}_*(\mathbf{u})) \right] + c \qquad \text{Extracting constant terms} \qquad (66)$$

$$\leq - \mathbb{E}_{\mathbf{u} \sim P_{\mathbf{U}}} \left[ \log q(\mathbf{u} \mid \mathbf{Y}_*(\mathbf{u})) \right] + c. \qquad \frac{\kappa^2 - 1}{2\kappa \log \kappa} > 1 \text{ for } \kappa > 1 \qquad (6)$$

The constant $c$ is solely dependent on $\kappa$ and $P_{\mathbf{U}}$, whereas $\mathbb{E}_{\mathbf{u} \sim P_{\mathbf{U}}} \left[ \log p(\mathbf{u}) \right]$ equals the negative entropy of $P_{\mathbf{U}}$. $\qquad \square$

**Relaxation to concentration inequalities via empirical distribution** The bounded importance weights assumption (i.e., there exists $\kappa \geq 1$ such that $1/\kappa \leq p(\mathbf{u})/q(\mathbf{u} \mid \mathbf{y}_*) \leq \kappa$ holds almost surely on the support $\Omega_{\mathbf{U}}(\mathcal{Y}_*)$) may be too strong for general cases, especially when the exogenous distribution has an infinite support set and $\Omega_{\mathbf{U}}(\mathcal{Y}_*)$ is also improper. For instance, even for two Gaussian distributions, their density ratio cannot be guaranteed to be bounded ([18], E.g. 4.1).

One remedy to mitigate this strong assumption is to weaken the statement of the theorem from "almost surely" to "with probability at least $1 - \delta$." In this way, if $\delta$ is small, the conclusion can still hold with high confidence. To reach such a conclusion, we typically need to use concentration inequalities and introduce other weaker assumptions. One of such weaker assumptions we introduce is that for any $\mathbf{y}_* \in \Omega_{\mathbf{Y}_*}$, the expected value of the square of the importance weights under $P_{\mathbf{U}}$ is finite, that is, $\mathbb{E}_{\mathbf{u} \sim P_{\mathbf{U}}} \left[ p^2(\mathbf{u})/q^2(\mathbf{u} \mid \mathbf{y}_*) \right] < +\infty$. Another assumption is about the boundedness of the importance weights in the empirical distribution $\widehat{P}_{\mathbf{U}}$; this is a discrete relaxation of the original boundedness assumption of importance weights in the exogenous distribution, which makes Thm. 1 trivially hold on the empirical distribution.

**Corollary 2** (Thm. 1 with Cantelli's Upper Bound). Assume that for any $\mathbf{y}_* \in \mathcal{Y}_*$, the expectation $\mathbb{E}_{\mathbf{u} \sim P_{\mathbf{U}}} \left[ p^2(\mathbf{u})/q^2(\mathbf{u}|\mathbf{y}_*) \right] < +\infty$. Let $\nu = \sup_{\mathbf{y}_* \in \mathcal{Y}_*} \mathbb{E}_{\mathbf{u} \sim P_{\mathbf{U}}} \left[ p^2(\mathbf{u})/q^2(\mathbf{u}|\mathbf{y}_*) \right]$. Denote by $\widehat{P}_{\mathbf{U}}$ the empirical distribution formed by $n$ i.i.d. samples $\mathbf{u}^{(i)} \sim P_{\mathbf{U}}, i = 1, \ldots, n$ drawn from $P_{\mathbf{U}}$, and assume there exists a constant $\kappa \geq 1$ such that $1/\kappa \leq p(\mathbf{u}^{(i)})/q(\mathbf{u}^{(i)}|\mathbf{y}_*) \leq \kappa$ for all $i = 1, \ldots, n$. Then, with probability at least $1 - \delta$, the following inequality holds:

$$\mathbb{V}_{\mathbf{u} \sim Q_{\mathbf{U}|\mathbf{y}_*}} \left[ \sigma_{\mathcal{Y}_*}(\mathbf{u}) \right] \leq -\mathbb{E}_{\mathbf{u} \sim \widehat{P}_{\mathbf{U}}} \left[ \log q(\mathbf{u} \,|\, \mathbf{Y}_*(\mathbf{u})) \right] + \sqrt{(\frac{1}{\delta} - 1)\frac{\nu}{n}} + c, \tag{67}$$

where the constant $c$ is the same as in Thm. 1.

*Proof.* First, consider the empirical distribution $\widehat{P}_{\mathbf{U}}$ formed by $n$ i.i.d. samples $\mathbf{u}^{(i)} \sim P_{\mathbf{U}}$. Given the existence of the bound $\kappa$ for these samples, meaning that the importance weights exist on the support of $\widehat{P}_{\mathbf{U}}$, it is evident that Thm. 1 holds for $\widehat{P}$:

$$\widehat{\mathbb{V}}_{\mathbf{u} \sim Q_{\mathbf{U}|\mathbf{y}_*}} \left[ \sigma_{\mathcal{Y}_*}(\mathbf{u}) \right] \leq -\mathbb{E}_{\mathbf{u} \sim \widehat{P}_{\mathbf{U}}} \left[ \log q(\mathbf{u} \,|\, \mathbf{Y}_*(\mathbf{u})) \right] + c, \tag{68}$$

where $\widehat{\mathbb{V}}$ denotes the empirical variance, which is calculated as:

$$\widehat{\mathbb{V}}_{\mathbf{u} \sim Q_{\mathbf{U}|\mathbf{y}_*}} \left[ \sigma_{\mathcal{Y}_*}(\mathbf{u}) \right] = \mathbb{E}_{\mathbf{u} \sim \widehat{P}_{\mathbf{U}}} \left[ \sigma_{\mathcal{Y}_*}(\mathbf{u}) \right] - P^2(\mathcal{Y}_*) \qquad \text{Eq. 5} \tag{69}$$

$$= \frac{1}{n} \sum_{i=1}^{n} \sigma_{\mathcal{Y}_*}(\mathbf{u}^{(i)}) - P^2(\mathcal{Y}_*), \tag{70}$$

where we treat $P(\mathcal{Y}_*)$ as a constant.

Next, we need to establish the inequality relationship between the empirical variance and the true variance. According to Cantelli's inequality, we have:

$$P\left(X - \mathbb{E}\left[X\right] \leq -\lambda\right) \leq \frac{\mathbb{V}\left[X\right]}{\mathbb{V}\left[X\right] + \lambda^2}. \tag{71}$$

Reversing the inequality direction, substituting the random variable $X$ with $\widehat{\mathbb{E}}\left[X\right] = \frac{1}{n}\sum_{i=1}^{n} X_i$ (where each $X_i$ is i.i.d.), and letting $\lambda = \sqrt{\left(\frac{1}{\delta} - 1\right)\frac{\mathbb{V}[X]}{n}}$, we obtain:

$$P\left(\mathbb{E}\left[X\right] < \widehat{\mathbb{E}}\left[X\right] + \sqrt{\left(\frac{1}{\delta} - 1\right)\frac{\mathbb{V}\left[X\right]}{n}}\right) > 1 - \delta. \tag{72}$$

Let $X_i = \sigma_{\mathcal{Y}_*}(\mathbf{u}^{(i)})$, and subtract $P(\mathcal{Y}_*)$ from both sides, then we have:

$$P\left(\mathbb{V}_{\mathbf{u} \sim Q_{\mathbf{U}|\mathbf{y}_*}} \left[\sigma_{\mathcal{Y}_*}(\mathbf{u})\right] < \widehat{\mathbb{V}}_{\mathbf{u} \sim Q_{\mathbf{U}|\mathbf{y}_*}} \left[\sigma_{\mathcal{Y}_*}(\mathbf{u})\right] + \sqrt{\left(\frac{1}{\delta} - 1\right)\frac{\mathbb{V}_{\mathbf{u} \sim P_{\mathbf{U}}} \left[\sigma_{\mathcal{Y}_*}(\mathbf{u})\right]}{n}}\right) > 1 - \delta. \tag{73}$$

where

$$\mathbb{V}_{\mathbf{u} \sim P_{\mathbf{U}}} \left[\sigma_{\mathcal{Y}_*}(\mathbf{u})\right] = \mathbb{E}_{\mathbf{u} \sim P_{\mathbf{U}}} \left[\sigma_{\mathcal{Y}_*}^2(\mathbf{u})\right] - \mathbb{E}_{\mathbf{u} \sim P_{\mathbf{U}}}^2 \left[\sigma_{\mathcal{Y}_*}(\mathbf{u})\right] \tag{74}$$

$$\leq \mathbb{E}_{\mathbf{u} \sim P_{\mathbf{U}}} \left[\sigma_{\mathcal{Y}_*}^2(\mathbf{u})\right] \tag{75}$$

$$\leq \sup_{\mathbf{y}_* \in \mathcal{Y}_*} \mathbb{E}_{\mathbf{u} \sim P_{\mathbf{U}}} \left[\frac{p^2(\mathbf{u})}{q^2(\mathbf{u} \,|\, \mathbf{y}_*)}\right] \tag{76}$$

$$= \nu. \tag{77}$$

Substituting Eq. 68 into this equation, we obtain the result:

$$P\left(\mathbb{V}_{\mathbf{u} \sim Q_{\mathbf{U}|\mathbf{y}_*}} \left[\sigma_{\mathcal{Y}_*}(\mathbf{u})\right] \leq -\mathbb{E}_{\mathbf{u} \sim \widehat{P}_{\mathbf{U}}} \left[\log q(\mathbf{u} \,|\, \mathbf{Y}_*(\mathbf{u}))\right] + \sqrt{\left(\frac{1}{\delta} - 1\right)\frac{\nu}{n}} + c\right) \geq 1 - \delta, \tag{78}$$

where the constant $c$ is the same as in Thm. 1. $\qquad \square$

The following obtains a tighter bound after introducing the constraint of sub-Gaussian conditions (specifically, assuming that the two-sided Bernstein's condition is satisfied):

**Corollary 3** (Thm. 1 with Bernstein's Upper Bound). Assume that for any $\mathbf{y}_* \in \mathcal{Y}_*$, the expectation $\mathbb{E}_{\mathbf{u} \sim P_{\mathbf{U}}} \left[ p^2(\mathbf{u})/q^2(\mathbf{u}|\mathbf{y}_*) \right] < +\infty$. Let $\nu = \sup_{\mathbf{y}_* \in \mathcal{Y}_*} \mathbb{E}_{\mathbf{u} \sim P_{\mathbf{U}}} \left[ p^2(\mathbf{u})/q^2(\mathbf{u}|\mathbf{y}_*) \right]$. Additionally, assume that the variable $\sigma_{\mathcal{Y}_*}(\mathbf{u})$ satisfies Bernstein's condition, meaning that there exists a parameter $b > 0$ such that for any $\lambda \in (-1/b, 1/b)$, the following inequality holds:

$$\mathbb{E}_{\mathbf{u} \sim P_{\mathbf{U}}} \left[ \exp \left( \lambda (\sigma_{\mathcal{Y}_*}(\mathbf{u}) - \mathbb{E}_{\mathbf{u} \sim P_{\mathbf{U}}} \left[ \sigma_{\mathcal{Y}_*}(\mathbf{u}) \right] ) \right) \right] \leq \exp \left( \frac{(\mathbb{V}_{\mathbf{u} \sim P_{\mathbf{U}}} \left[ \sigma_{\mathcal{Y}_*}(\mathbf{u}) \right]) \lambda^2/2}{1 - b|\lambda|} \right). \quad (79)$$

Denote by $\widehat{P}_{\mathbf{U}}$ the empirical distribution formed by $n$ i.i.d. samples $\mathbf{u}^{(i)} \sim P_{\mathbf{U}}, i = 1, \ldots, n$ drawn from $P_{\mathbf{U}}$, and assume there exists a constant $\kappa \geq 1$ such that $1/\kappa \leq p(\mathbf{u}^{(i)})/q(\mathbf{u}^{(i)}|\mathbf{y}_*) \leq \kappa$ for all $i = 1, \ldots, n$. Then, with probability at least $1 - \delta$, the following inequality holds:

$$\mathbb{V}_{\mathbf{u} \sim Q_{\mathbf{U}|\mathbf{y}_*}} \left[ \sigma_{\mathcal{Y}_*}(\mathbf{u}) \right] \leq -\mathbb{E}_{\mathbf{u} \sim \widehat{P}_{\mathbf{U}}} \left[ \log q(\mathbf{u}|\mathbf{Y}_*(\mathbf{u})) \right] + \frac{b}{n} \log \left( \frac{1}{\delta} \right) + \sqrt{2 \log \left( \frac{1}{\delta} \right) \frac{\nu}{n}} + c, \quad (80)$$

where the constant $c$ is the same as in Thm. 1.

*Proof.* When Bernstein's condition is satisfied, the following inequality holds:

$$P \left( \left| \widehat{\mathbb{E}} \left[ X \right] - \mathbb{E} \left[ X \right] \right| < \frac{b}{n} \log \left( \frac{1}{\delta} \right) + \sqrt{2 \log \left( \frac{1}{\delta} \right) \frac{\mathbb{E} \left[ V \right]}{n}} \right) \geq 1 - \delta. \quad (81)$$

The remaining derivation is similar to that in Cor. 2. $\square$

It is trivially provable that $\sqrt{\frac{1}{\delta} - 1} \gg \sqrt{2 \log \left( \frac{1}{\delta} \right)}$ when $\delta$ approaches zero. This implies that, for the same $\delta$ close to zero, the remainder term in Cor. 3 is typically smaller than that in Cor. 2, meaning that the former provides a much tighter upper bound with a high probability.

**A guard proposal distribution to ensuring boundedness**  For a proposal distribution $Q_{\mathbf{U}|\mathbf{y}_*}$, define the bounded part as $\Omega_{\mathbf{U}}^{[\kappa]} = \{ \mathbf{u} \mid 1/\kappa \leq p(\mathbf{u})/q(\mathbf{u}|\mathbf{y}_*) \leq \kappa \}$. If the probabilities of the unbounded part under the exogenous distribution and the proposal distribution, $P_{\mathbf{U}} \left( \Omega_{\mathbf{U}} \setminus \Omega_{\mathbf{U}}^{[\kappa]} \right) = \mathbb{E}_{\mathbf{u} \sim P_{\mathbf{U}}} \left[ \mathbb{1}_{\Omega_{\mathbf{U}} \setminus \Omega_{\mathbf{U}}^{[\kappa]}} \right]$ and $Q_{\mathbf{U}|\mathbf{y}_*} \left( \Omega_{\mathbf{U}} \setminus \Omega_{\mathbf{U}}^{[\kappa]} \right) = \mathbb{E}_{\mathbf{u} \sim Q_{\mathbf{U}|\mathbf{y}_*}} \left[ \mathbb{1}_{\Omega_{\mathbf{U}} \setminus \Omega_{\mathbf{U}}^{[\kappa]}} \right]$, are known, we can splice the exogenous distribution over the unbounded part to form a new proposal distribution that ensures the bounded importance weight condition:

**Proposition 2** (Importance Weight Bound Guard). For a proposal distribution $Q_{\mathbf{U}|\mathbf{y}_*}$, construct a new proposal distribution as follows:

$$q^{[\kappa]}(\mathbf{u}|\mathbf{y}_*) = \begin{cases} \frac{1 - P_{\mathbf{U}} \left( \Omega_{\mathbf{U}} \setminus \Omega_{\mathbf{U}}^{[\kappa]} \right)}{1 - Q_{\mathbf{U}|\mathbf{y}_*} \left( \Omega_{\mathbf{U}} \setminus \Omega_{\mathbf{U}}^{[\kappa]} \right)} q(\mathbf{u}|\mathbf{y}_*), & \text{if } \mathbf{u} \in \Omega_{\mathbf{U}}^{[\kappa]} \\ p(\mathbf{u}), & \text{otherwise} \end{cases}, \quad (82)$$

which we refer to as the guard of the proposal distribution $Q_{\mathbf{U}|\mathbf{y}_*}$. Then, for any $\mathbf{u} \in \Omega_{\mathbf{U}}$, there exists a constant

$$\kappa' = \max \left( \frac{1 - P_{\mathbf{U}} \left( \Omega_{\mathbf{U}} \setminus \Omega_{\mathbf{U}}^{[\kappa]} \right)}{1 - Q_{\mathbf{U}|\mathbf{y}_*} \left( \Omega_{\mathbf{U}} \setminus \Omega_{\mathbf{U}}^{[\kappa]} \right)}, \frac{1 - Q_{\mathbf{U}|\mathbf{y}_*} \left( \Omega_{\mathbf{U}} \setminus \Omega_{\mathbf{U}}^{[\kappa]} \right)}{1 - P_{\mathbf{U}} \left( \Omega_{\mathbf{U}} \setminus \Omega_{\mathbf{U}}^{[\kappa]} \right)} \right) \cdot \kappa, \quad (83)$$

such that $1/\kappa' \leq q^{[\kappa]}(\mathbf{u}|\mathbf{y}_*)/p(\mathbf{u}) \leq \kappa'$.

*Proof.* First, we prove that the constructed proposal distribution is a valid distribution, mainly by showing that it satisfies the normalization condition:

$$\int_{\Omega_{\mathbf{U}}} q^{[\kappa]}(\mathbf{u}\,|\,\mathbf{y}_*)\,\mathrm{d}\mathbf{u} \tag{84}$$

$$= \int_{\Omega_{\mathbf{U}}^{[\kappa]}} \frac{1 - P_{\mathbf{U}}\left(\Omega_{\mathbf{U}} \setminus \Omega_{\mathbf{U}}^{[\kappa]}\right)}{1 - Q_{\mathbf{U}|\mathbf{y}_*}\left(\Omega_{\mathbf{U}} \setminus \Omega_{\mathbf{U}}^{[\kappa]}\right)} q(\mathbf{u}\,|\,\mathbf{y}_*)\,\mathrm{d}\mathbf{u} + \int_{\Omega_{\mathbf{U}} \setminus \Omega_{\mathbf{U}}^{[\kappa]}} p(\mathbf{u})\,\mathrm{d}\mathbf{u} \tag{85}$$

$$= \frac{1 - P_{\mathbf{U}}\left(\Omega_{\mathbf{U}} \setminus \Omega_{\mathbf{U}}^{[\kappa]}\right)}{1 - Q_{\mathbf{U}|\mathbf{y}_*}\left(\Omega_{\mathbf{U}} \setminus \Omega_{\mathbf{U}}^{[\kappa]}\right)} \left(1 - Q_{\mathbf{U}|\mathbf{y}_*}\left(\Omega_{\mathbf{U}} \setminus \Omega_{\mathbf{U}}^{[\kappa]}\right)\right) + P_{\mathbf{U}}\left(\Omega_{\mathbf{U}} \setminus \Omega_{\mathbf{U}}^{[\kappa]}\right) \tag{86}$$

$$= 1. \tag{87}$$

Next, we prove the boundedness: when $\mathbf{u} \in \Omega_{\mathbf{U}}^{[\kappa]}$, according to the definition of $\Omega_{\mathbf{U}}^{[\kappa]}$, we have $1/\kappa \le p(\mathbf{u})/q(\mathbf{u}\,|\,\mathbf{y}_*) \le \kappa$. Since $\kappa' \ge \kappa$, it follows that $1/\kappa' \le p(\mathbf{u})/q^{[\kappa]}(\mathbf{u}\,|\,\mathbf{y}_*) \le \kappa'$. When $\mathbf{u} \notin \Omega_{\mathbf{U}}^{[\kappa]}$, it trivially holds that $p(\mathbf{u})/q^{[\kappa]}(\mathbf{u}\,|\,\mathbf{y}_*) = 1$. $\qquad\square$

In the subsequent experiments, we assume that the probability of the unbounded part could be non-zero but approximates zero. Formally, $P_{\mathbf{U}}\left(\Omega_{\mathbf{U}}^{[\kappa]}\right) \approx 1$ and $Q_{\mathbf{U}|\mathbf{y}_*}\left(\Omega_{\mathbf{U}}^{[\kappa]}\right) \approx 1$. This implies that the error terms in Cor. 2 and Cor. 3 would be small enough, or that when using the guard proposal distribution Prop. 2, we could obtain a relatively unbiased estimate without the need for re-weighting (where $\left(1 - P_{\mathbf{U}}\left(\Omega_{\mathbf{U}} \setminus \Omega_{\mathbf{U}}^{[\kappa]}\right)\right) / \left(1 - Q_{\mathbf{U}|\mathbf{y}_*}\left(\Omega_{\mathbf{U}} \setminus \Omega_{\mathbf{U}}^{[\kappa]}\right)\right) \approx 1$).

**The generalization of Thm. 1** Cor. 1 is generalized from Thm. 1 in a straightforward manner through monotonicity. In particular, the relaxation and guarantees discussed earlier also apply to Cor. 1 when the scope of assumptions extends from a single counterfactual variable $\mathcal{Y}_*$ to multiple counterfactual variables of different forms, $\mathcal{Y}_*^{(s)}, s \in \mathcal{S}$.

**Corollary 1** (Expected Variance Upper Bound). Let $Q_{\mathcal{S}}$ denote an arbitrary distribution defined over the state space $\mathcal{S}$ of a stochastic counterfactual process, and let $P_{\mathcal{Y}_*}^{(s)}$ denote an arbitrary distribution defined over the $\sigma$-algebra $\Sigma_{\mathbf{Y}_*}^{(s)}$ corresponding to a set of counterfactual variables $\mathbf{Y}_*^{(s)}$ given a state $s \in \mathcal{S}$. $P_{\mathcal{Y}_*}$ is the joint distribution induced by $P_{\mathcal{Y}_*}^{(s)}$ and $Q_{\mathcal{S}}$. If the conditions in Thm. 1 are met for any $\mathcal{Y}_*^{(s)}$ and any $s \in \mathcal{S}$, then for any $Q_{\mathbf{U}|\mathbf{y}_*}$ where $\mathbf{y}_* \in \mathcal{Y}_*^{(s)}$ and any $s \in \mathcal{S}$:

$$\mathbb{E}_{\mathcal{Y}_*^{(s)} \sim P_{\mathcal{Y}_*}} \left[\mathbb{V}_{\mathbf{u} \sim Q_{\mathbf{U}|\mathbf{y}_*}}\left[\sigma_{\mathcal{Y}_*}(\mathbf{u})\right]\right] \le -\mathbb{E}_{s \sim Q_{\mathcal{S}}}\left[\mathbb{E}_{\mathbf{u} \sim P_{\mathbf{U}}}\left[\log q(\mathbf{u}\,|\,\mathbf{Y}_*^{(s)}(\mathbf{u}))\right]\right] + c, \tag{7}$$

where the constant $c$ is the same as in Thm. 1.

*Proof.* For any state $s \in \mathcal{S}$, and any $\mathcal{Y}_*^{(s)} \in \Sigma_{\mathbf{Y}_*}^{(s)}$, according to Thm. 1, if $\mathcal{Y}_*^{(s)}$ satisfies the bounded density ratio condition, we have:

$$\mathbb{V}_{\mathbf{u} \sim Q_{\mathbf{U}|\mathbf{y}_*}}\left[\sigma_{\mathcal{Y}_*}(\mathbf{u}\,|\,\mathbf{y}_*)\right] \le -\mathbb{E}_{\mathbf{u} \sim P_{\mathbf{U}}}\left[\log q(\mathbf{u}\,|\,\mathbf{Y}_*(\mathbf{u}))\right] + c. \tag{6}$$

According to the monotonicity of expectation, we take the expectation w.r.t. $P_{\mathcal{Y}_*}^{(s)}$ on both sides, then:

$$\mathbb{E}_{\mathcal{Y}_*^{(s)} \sim P_{\mathcal{Y}_*}^{(s)}} \left[\mathbb{V}_{\mathbf{u} \sim Q_{\mathbf{U}|\mathbf{y}_*}}\left[\sigma_{\mathcal{Y}_*}(\mathbf{u}\,|\,\mathbf{y}_*^{(s)})\right]\right] \tag{88}$$

$$\le \mathbb{E}_{\mathcal{Y}_*^{(s)} \sim P_{\mathcal{Y}_*}^{(s)}} \left[-\mathbb{E}_{\mathbf{u} \sim P_{\mathbf{U}}}\left[\log q(\mathbf{u}\,|\,\mathbf{Y}_*(\mathbf{u}))\right] + c\right] \tag{89}$$

$$= -\mathbb{E}_{\mathcal{Y}_*^{(s)} \sim P_{\mathcal{Y}_*}^{(s)}} \left[\mathbb{E}_{\mathbf{u} \sim P_{\mathbf{U}}}\left[\log q(\mathbf{u}\,|\,\mathbf{Y}_*(\mathbf{u}))\right]\right] + \mathbb{E}_{\mathcal{Y}_*^{(s)} \sim P_{\mathcal{Y}_*}^{(s)}} [c] \tag{90}$$

$$= -\mathbb{E}_{\mathbf{u} \sim P_{\mathbf{U}}} \left[\log q(\mathbf{u}\,|\,\mathbf{Y}_*^{(s)}(\mathbf{u}))\right] + c, \tag{91}$$

where $\mathbb{E}_{\mathbf{u} \sim P_{\mathbf{U}}}\left[\log q(\mathbf{u}\,|\,\mathbf{Y}_*^{(s)}(\mathbf{u}))\right]$ is independent of $\mathcal{Y}_*^{(s)}$, and the constant $c$ is irrelevant to $\mathcal{Y}_*^{(s)}$.

We then take the expectation w.r.t. $P_{\mathbf{U}}$ on both sides of the inequality, producing:

$$\mathbb{E}_{\mathcal{Y}_* \sim P_{\mathcal{Y}_*}} \left[ \mathbb{V}_{\mathbf{u} \sim Q_{\mathbf{U}|\mathbf{y}_*}} \left[ \sigma_{\mathcal{Y}_*}(\mathbf{u}\,|\,\mathbf{y}_*) \right] \right] \tag{92}$$

$$= \mathbb{E}_{s \sim Q_{\mathcal{S}}} \left[ \mathbb{E}_{\mathcal{Y}_*^{(s)} \sim P_{\mathcal{Y}_*}^{(s)}} \left[ \mathbb{V} \left[ \sigma_{\mathcal{Y}_*}(\mathbf{u}\,|\,\mathbf{y}_*) \right] \right] \right] \tag{93}$$

$$\leq \mathbb{E}_{s \sim Q_{\mathcal{S}}} \left[ -\mathbb{E}_{\mathbf{u} \sim P_{\mathbf{U}}} \left[ \log q(\mathbf{u}\,|\,\mathbf{Y}_*^{(s)}(\mathbf{u})) \right] + c \right] \tag{94}$$

$$= -\mathbb{E}_{s \sim Q_{\mathcal{S}}} \left[ \mathbb{E}_{\mathbf{u} \sim P_{\mathbf{U}}} \left[ \log q(\mathbf{u}\,|\,\mathbf{Y}_*^{(s)}(\mathbf{u})) \right] \right] + \mathbb{E}_{s \sim Q_{\mathcal{S}}} \left[ c \right] \tag{95}$$

$$= -\mathbb{E}_{s \sim Q_{\mathcal{S}}} \left[ \mathbb{E}_{\mathbf{u} \sim P_{\mathbf{U}}} \left[ \log q(\mathbf{u}\,|\,\mathbf{Y}_*^{(s)}(\mathbf{u})) \right] \right] + c, \tag{7}$$

where $c$ is a constant independent of any state $s \in \mathcal{S}$. $\qquad\square$

### A.3 Learning and Inference

**Learning**  Cor. 1 provides an upper bound that directly gives an optimization objective:

$$\arg\min_q -\mathbb{E}_{s \sim Q_{\mathcal{S}}} \left[ \mathbb{E}_{\mathbf{u} \sim P_{\mathbf{U}}} \left[ \log q(\mathbf{u}\,|\,\mathbf{Y}_*^{(s)}(\mathbf{u})) \right] \right]. \tag{8}$$

This is equivalent to learning the conditional distribution $Q_{\mathbf{U}|\mathbf{Y}_*}$ using maximum likelihood. To obtain an estimate of this log-likelihood, we can directly use Monte Carlo methods, which simply involve two sampling processes and one computation: i) sampling state $s$ from the prior state distribution $Q_{\mathcal{S}}$; ii) sampling $\mathbf{u}$ from the exogenous distribution $P_{\mathbf{U}}$; iii) computing the log density (or probability for discrete exogenous variables) $\log q(\mathbf{u}\,|\,\mathbf{Y}_*^{(s)}(\mathbf{u}))$.

---

**Algorithm 1** Exogenous Matching Learning

**Input:** Exogenous distribution $P_{\mathbf{U}}$, prior distribution for states $Q_{\mathcal{S}}$, conditional distribution model $Q_{\mathbf{U}|\mathbf{Y}_*}$ with parameters $\theta_{\mathbf{y}_*}$, minibatch size $n$ and learning rate $\eta$.

1: **while** Eq. 8 not converged **do**
2:     **for** $i \leftarrow 1$ and $i \leq n$ **do**
3:         $s \sim Q_{\mathcal{S}}, \mathbf{u} \sim P_{\mathbf{U}}$
4:         $\theta_{\mathbf{y}_*} \leftarrow \theta_{\mathbf{y}_*} - \eta \cdot \nabla_{\theta_{\mathbf{y}_*}} \log q(\mathbf{u}\,|\,\mathbf{Y}_*^{(s)}(\mathbf{u}))$
5:     **end for**
6: **end while**

Figure 5: The learning algorithm for Exogenous Matching.

---

Assuming that we can compute the gradient of $q(\mathbf{u} \,|\, \mathbf{y}_*)$ w.r.t. the parameters $\theta_{\mathbf{y}_*}$ of $Q_{\mathbf{U}|\mathbf{y}_*}$, as modeled in the paper, then combining with the minibatch gradient descent algorithm, we can write the complete algorithm for the training process as in Fig. 5.

**Inference**  If for any $\mathbf{y}_* \in \mathcal{Y}_*$, the support of the proposal distribution $Q_{\mathbf{U}|\mathbf{y}_*}$ covers the support of $P_{\mathbf{U}}$, then it constitutes a valid proposal, and an unbiased estimate can be obtained using Eq. 4. More can be achieved through multiple importance sampling, as any estimate obtained from the proposal $Q_{\mathbf{U}|\mathbf{y}_*}$ is unbiased for such $\mathbf{y}_* \in \mathcal{Y}_*$. Consequently, the expectation over these estimates remains unbiased, i.e.,

$$P(\mathcal{Y}_*) = \mathbb{E}_{\mathbf{u} \sim Q_{\mathbf{U}|\mathbf{y}_*}} \left[ \sigma_{\mathcal{Y}_*[\mathbf{y}_*]}(\mathbf{u}) \right] \tag{96}$$

$$= \mathbb{E}_{\mathbf{y}_* \sim Q_{\mathbf{Y}_*}} \left[ \mathbb{E}_{\mathbf{u} \sim Q_{\mathbf{U}|\mathbf{y}_*}} \left[ \sigma_{\mathcal{Y}_*[\mathbf{y}_*]}(\mathbf{u}) \right] \right], \tag{9}$$

---

**Algorithm 2** Multiple Proposals Inference

**Input:** Prior distribution $Q_{\mathcal{Y}_*}$, conditional proposal distribution $Q_{\mathbf{U}|\mathbf{Y}_*}$, sample size $n$.

1: $S \leftarrow \emptyset$
2: **for** $i \leftarrow 1$ and $i \leq n$ **do**
3:     $\mathbf{y}_*^{(i)} \sim Q_{\mathbf{Y}_*}, \mathbf{u}^{(i)} \sim Q_{\mathbf{U}|\mathbf{y}_*^{(i)}}$
4:     $\sigma^{(i)} \leftarrow \left( p(\mathbf{u}^{(i)})/q(\mathbf{u}^{(i)}\,|\,\mathbf{y}_*) \right) \mathbb{1}_{\Omega_{\mathbf{U}}(\mathcal{Y}_*)}(\mathbf{u}^{(i)})$
5:     $S \leftarrow S \cup \{\sigma^{(i)}\}$
6: **end for**
7: **Output:** $\widehat{P}(\mathcal{Y}_*) \leftarrow \left( \sum_{i=1}^{n} \sigma^{(i)} \right)/n$

Figure 6: The inference algorithm for multiple importance sampling.

---

which corresponds to the estimator used in Eq. 9 and provides more robust results when an appropriate prior $Q_{\mathbf{Y}_*}$ is chosen. Our inference algorithm involves two sampling processes and one computation: i) sampling $\mathbf{y}_*$ from the prior distribution $Q_{\mathbf{Y}_*}$; ii) sampling $\mathbf{u}$ from the proposal distribution $Q_{\mathbf{U}|\mathbf{y}_*}$; iii) computing the importance weights and checking if $\mathbf{Y}_*(\mathbf{u}) \in \mathcal{Y}_*$ to obtain $\sigma_{\mathcal{Y}_*[\mathbf{y}_*]}(\mathbf{u})$. This process is detailed in Fig. 6. In our experiments, because the tasks are density estimation (for continuous) and point probability queries (for discrete), we actually still use a single proposal distribution at $\mathbf{y}_*$, which is a special case where $Q_{\mathbf{Y}_*}$ is a Dirac distribution.

## A.4 Markov Boundary

In order to express counterfactuals that span multiple submodels using a single model, the twin SCM concept is often used to articulate counterfactuals for a particular setting within a unified framework.

**Definition 3** (Twin SCM). For an SCM $\mathcal{M} = \langle \mathbf{U}, \mathbf{V}, \mathcal{F}, P_{\mathbf{U}} \rangle$ and a set of its submodels $\{\mathcal{M}_{1[\mathbf{x}_1]}, \mathcal{M}_{2[\mathbf{x}_2]}, \ldots, \mathcal{M}_{k[\mathbf{x}_k]}\}$, we refer to $\mathcal{M}_* = \langle \mathbf{U}, \mathbf{V}_*, \mathcal{F}_*, P_{\mathbf{U}} \rangle$ as a Twin Structural Causal Model (Twin SCM), where $\mathbf{V}_* = \bigcup_{i=1}^k \mathbf{V}_{i[\mathbf{x}_i]}$ and $\mathcal{F}_* = \bigcup_{i=1}^k \mathcal{F}_{i[\mathbf{x}_i]}$. $\mathbf{V}_{i[\mathbf{x}_i]}$ represents the endogenous variables in the $i$-th submodel, and $\mathcal{F}_{i[\mathbf{x}_i]}$ denotes the corresponding causal mechanisms.

In general, the twin SCM amalgamates multiple distinct submodels into one SCM, thereby rendering conclusions drawn on a single SCM applicable to the twin SCM. It is trivial to demonstrate that if the original SCM is recursive, then the twin SCM remains recursive. Consequently, the causal graph derived from the twin SCM is also an ADMG, and the augmented graph is a DAG.

Our proof is based on d-separation [75, 76], which is a criterion for quickly determining conditional independence on a Directed Acyclic Graph (DAG). Suppose a distribution is denoted by $P$ and its corresponding DAG is $\mathcal{G}$. We say that a path $p$ is blocked by a set of nodes $\mathbf{Z}$ if and only if: i) $p$ contains a chain $A \to B \to C$ or a fork $A \leftarrow B \to C$ and the intermediate node $B$ is in $\mathbf{Z}$; ii) $p$ contains a collider $A \to B \leftarrow C$, and neither the intermediate node $B$ nor its descendants are in $\mathbf{Z}$. If all paths between two sets of nodes $\mathbf{X}$ and $\mathbf{Y}$ are blocked, then $\mathbf{X}$ and $\mathbf{Y}$ are said to be d-separated given $\mathbf{Z}$, denoted as $\mathbf{X} \perp\!\!\!\perp_{\mathcal{G}} \mathbf{Y} \mid \mathbf{Z}$, implying $\mathbf{X} \perp\!\!\!\perp_P \mathbf{Y} \mid \mathbf{Z}$, i.e., $\mathbf{X}$ and $\mathbf{Y}$ are conditionally independent given $\mathbf{Z}$. In this work, we assume faithfulness, which means that if the conditions for conditional independence are met, then d-separation holds. Combining acyclicity allows for mutual inference between the d-separation and independence.

First, let us restate the definition of the Markov boundary.

**Definition 2** (Counterfactual Markov Boundary). For an exogenous variable $U_j \in \mathbf{U}$ and a set of counterfactual variables $\mathbf{Y}$, along with their joint distribution $P$. If $U_j$ is independent of $\mathbf{Y} \setminus \mathbf{X}$ given $\mathbf{X}$ under $P$, i.e. $U_j \perp\!\!\!\perp_P (\mathbf{Y} \setminus \mathbf{X}) \mid \mathbf{X}$, then $\mathbf{X}$ is termed a Markov blanket of $U_j$ on $\mathbf{Y}$. The collection of all Markov blankets of $U_j$ on $\mathbf{Y}$ is denoted as $\mathfrak{B}_j(\mathbf{Y})$. If $\mathbf{X} \in \mathfrak{B}_j(\mathbf{Y})$ is a Markov blanket of $U_j$ on $\mathbf{Y}$, and there exists no $\mathbf{X}' \subsetneq \mathbf{X}$ such that $\mathbf{X}' \in \mathfrak{B}_j(\mathbf{Y})$, then $\mathbf{X}$ is termed a (counterfactual) Markov boundary of $U_j$ on $\mathbf{Y}$, denoted as $\mathbf{B}_j(\mathbf{Y})$.

This definition relies on conditional independence. With the aid of the augmented graph $\mathcal{G}^a$ and the d-separation based on the twin SCM, it is possible to derive Thm. 2 and Thm. 3 from the properties on the graph.

**Theorem 2** (Counterfactual Markov Boundary Independence). If $\mathbf{Y}_* = \bigcup_{i=1}^k \mathbf{Y}_{i[\mathbf{x}_i]}$ and each $\mathbf{Y}_{i[\mathbf{x}_i]}$ corresponds to a different submodel $\mathcal{M}_{\mathbf{x}_i}$, then for each $U_j \in \mathbf{U}$, there exists a Markov boundary $\mathbf{B}_j(\mathbf{Y}_*) = \bigcup_{i=1}^k \mathbf{B}_j(\mathbf{Y}_{i[\mathbf{x}_i]})$ on $\mathbf{Y}_*$, where $\mathbf{B}_j(\mathbf{Y}_{i[\mathbf{x}_i]})$ is a Markov boundary on $\mathcal{M}_{\mathbf{x}_i}$.

*Proof.* We aim to demonstrate that $\bigcup_{i=1}^k \mathbf{B}_j(\mathbf{Y}_{i[\mathbf{x}_i]})$ forms a Markov boundary for $\mathbf{Y}_*$, where $\mathbf{B}_j(\mathbf{Y}_{i[\mathbf{x}_i]})$ denotes the Markov boundary over $\mathbf{Y}_{i[\mathbf{x}_i]}$.

(i) $\bigcup_{i=1}^k \mathbf{B}_j(\mathbf{Y}_{i[\mathbf{x}_i]})$ is a Markov blanket of $U_j$: For any submodel $\mathcal{M}_j$, according to the faithfulness assumption, since $\mathbf{B}_j(\mathbf{Y}_{i[\mathbf{x}_i]})$ is a Markov blanket over $\mathbf{Y}_{i[\mathbf{x}_i]}$, it blocks all paths between $U_j$ and $\mathbf{Y}_{i[\mathbf{x}_i]} \setminus \mathbf{B}_j(\mathbf{Y}_{i[\mathbf{x}_i]})$ in the augmented graph of the twin SCM. We proceed to prove that, given $\mathbf{B}_j(\mathbf{Y}_{i[\mathbf{x}_i]})$, it does not alter the blocked paths of $\mathbf{B}_j(\mathbf{Y}_{i[\mathbf{x}_i]})$. For a blocked path $p \in \mathcal{P}$, unblocking can occur only at the intermediate node and its descendants in the colliders. However, the nodes in $\mathbf{B}_j(\mathbf{Y}_{i[\mathbf{x}_i]})$ are not in $\mathbf{Y}_{i[\mathbf{x}_i]}$, thus demonstrating that its blocking status remains unchanged. Therefore, $\bigcup_{i=1}^k \mathbf{B}_j(\mathbf{Y}_{i[\mathbf{x}_i]})$ is a Markov blanket of $U_j$.

(ii) Any proper subset of $\bigcup_{i=1}^k \mathbf{B}_j(\mathbf{Y}_{i[\mathbf{x}_i]})$ is not a Markov blanket of $U_j$: Suppose that a proper subset $\mathbf{D} \subset \bigcup_{i=1}^k \mathbf{B}_j(\mathbf{Y}_{i[\mathbf{x}_i]})$ is a Markov blanket. Let $\mathbf{D}_j = \mathbf{D} \cap \mathbf{Y}_{i[\mathbf{x}_i]}$ be a subset in some submodel of this Markov blanket. Obviously, since any proper subset of $\mathbf{B}_j(\mathbf{Y}_{i[\mathbf{x}_i]})$ is not a Markov blanket in $\mathbf{Y}_{i[\mathbf{x}_i]}$, $\mathbf{D}_j$ is not a Markov blanket on $\mathbf{Y}_{i[\mathbf{x}_i]}$. This implies that there exists a subset $\mathbf{S}_j$ of $\mathbf{Y}_{i[\mathbf{x}_i]}$ which is not independent of $U_j$ given $\mathbf{D}_j$. We now show that $\mathbf{S}_j$ remains dependent on $U_j$ given $\mathbf{D}$. By the faithfulness assumption, as $\mathbf{S}_j$ is not independent of $U_j$ given $\mathbf{D}_j$, there still exists an unblocked path from $\mathbf{S}_j$ to $U_j$ in the augmented graph of the twin SCM. Continuing with the condition

---
**Algorithm 3** Finding Counterfactual Markov Boundary
---

**Input:** An exogenous variable $U_j \in \mathbf{U}$, a set of counterfactual variables $\mathbf{Y}_*$, an augmented graph $\mathcal{G}^a$ of SCM $\mathcal{M} = \{\mathbf{V}, \mathbf{U}, \mathcal{F}, P_{\mathbf{U}}\}$.

1: $\mathbf{B}_j(\mathbf{Y}_*) \leftarrow \emptyset$
2: **for** $\mathbf{Y}_{i[\mathbf{x}_i]} \subseteq \mathbf{Y}_*$ **do**
3:     $\mathbf{B}_j(\mathbf{Y}_{i[\mathbf{x}_i]}) \leftarrow \emptyset$
4:     **for** $Y \in \mathbf{Y}_{i[\mathbf{x}_i]}$ **do**
5:         **if** $\neg$ D-SEPARATION$(\{Y_{\mathbf{x}_i}\}, \{U_j\}, \mathbf{Y}_{i[\mathbf{x}_i]} \setminus \{Y_{\mathbf{x}_i}\}, \mathcal{G}^a_{\mathbf{x}_i})$ **then**
6:             $\mathbf{B}_j(\mathbf{Y}_{i[\mathbf{x}_i]}) \leftarrow \mathbf{B}_j(\mathbf{Y}_{i[\mathbf{x}_i]}) \cup \{Y\}$           ▷ Thm. 3
7:         **end if**
8:     **end for**
9:     $\mathbf{B}_j(\mathbf{Y}_*) \leftarrow \mathbf{B}_j(\mathbf{Y}_*) \cup \mathbf{B}_j(\mathbf{Y}_{i[\mathbf{x}_i]})$           ▷ Thm. 2
10: **end for**
**Output:** counterfactual Markov boundary $\mathbf{B}_j(\mathbf{Y}_*)$.

---

Figure 7: Inferring counterfactual Markov boundaries through d-separation, Thms. 2 and 3, with faithfulness is assumed. D-SEPARATION$(\mathbf{X}, \mathbf{Y}, \mathbf{Z}, \mathcal{G})$ returns true if and only if $\mathbf{X} \perp\!\!\!\perp_{\mathcal{G}} \mathbf{Y} \mid \mathbf{Z}$.

on $\mathbf{D} \setminus \mathbf{D}_j$, we discuss all the paths $\mathcal{P}$ from $\mathbf{S}_j$ to $U_j$. For an intermediate node of a collider on a path $p \in \mathcal{P}$, since it is unblocked, its descendants must be conditioned, thus continuing to condition on $\mathbf{D} \setminus \mathbf{D}_j$ does not affect the colliders. For a non-collider on a path $p \in \mathcal{P}$, for $p$ to be blocked, there must exist an intermediate node of a non-collider in $\mathbf{D} \setminus \mathbf{D}_j$, however, $(\mathbf{D} \setminus \mathbf{D}_j) \cap \mathbf{Y}_{i[\mathbf{x}_i]} = \emptyset$, so this condition cannot be met. Hence, continuing to condition on $\mathbf{D} \setminus \mathbf{D}_j$ does not affect any unblocked paths. According to d-separation, given $\mathbf{D}$, $\mathbf{S}_j$ and $U_j$ remain dependent. Therefore, $\mathbf{D}$ is not a Markov blanket, which contradicts the assumption. This means that no proper subset of $\bigcup_{i=1}^k \mathbf{B}_j(\mathbf{Y}_{i[\mathbf{x}_i]})$ is a Markov blanket of $U_j$. $\qquad\square$

**Theorem 3** (Counterfactual Markov Boundary on Graph). For an exogenous variable $U_j \in \mathbf{U}$ and a counterfactual variable set $\mathbf{Y}_{\mathbf{x}}$ from the submodel $\mathcal{M}_{\mathbf{x}}$, the counterfactual variable $Y_{\mathbf{x}} \in \mathbf{B}_j(\mathbf{Y}_{\mathbf{x}})$ if and only if $Y_{\mathbf{x}} \not\perp\!\!\!\perp_{\mathcal{G}^a_{\mathbf{x}}} U_j \mid \mathbf{Y}_{\mathbf{x}} \setminus \{Y_{\mathbf{x}}\}$, i.e., when given $\mathbf{Y}_{\mathbf{x}} \setminus \{Y_{\mathbf{x}}\}$, $Y_{\mathbf{x}}$ and $U_j$ are not d-separated on $\mathcal{G}^a_{\mathbf{x}}$, where $\mathcal{G}^a_{\mathbf{x}}$ is the augmented graph induced from the submodel $\mathcal{M}_{\mathbf{x}}$.

*Proof.* (i) $Y_{\mathbf{x}} \in \mathbf{B}_j(\mathbf{Y}_{\mathbf{x}})$ implies $Y_{\mathbf{x}} \not\perp\!\!\!\perp_{\mathcal{G}^a_{\mathbf{x}}} U_j \mid \mathbf{Y}_{\mathbf{x}} \setminus \{Y_{\mathbf{x}}\}$: Assuming $Y_{\mathbf{x}} \perp\!\!\!\perp_{\mathcal{G}^a_{\mathbf{x}}} U_j \mid \mathbf{Y}_{\mathbf{x}} \setminus \{Y_{\mathbf{x}}\}$, according to the faithfulness assumption, all paths between $U_j$ and $Y_{\mathbf{x}}$ are blocked given $\mathbf{Y}_{\mathbf{x}} \setminus \{Y_{\mathbf{x}}\}$. If a path consists solely of colliders, then, since the intermediate node are given, the path remains unblocked, contradicting the assumption. This implies that on every path, there exists at least one non-collider intermediate node, denoted this set by $\mathbf{R}$. Since each path has at least one non-collider intermediate node, given $\mathbf{R}$, all paths between $Y_{\mathbf{x}}$ and $U_j$ are blocked. Let $\mathbf{R}^b = \mathbf{R} \cap \mathbf{B}_j(\mathbf{Y}_{\mathbf{x}})$ and $\mathbf{R}^c = \mathbf{R} \setminus \mathbf{R}^b$. For $\mathbf{R}^c$, the Markov boundary property holds, blocking all paths between $U_j$ and $Y_{\mathbf{x}}$ given $\mathbf{B}_j(\mathbf{Y}_{\mathbf{x}})$. Thus, for every path between $Y_{\mathbf{x}}$ and $U_j$, it either passes through nodes in $\mathbf{R}^b$ and is blocked given $\mathbf{B}_j(\mathbf{Y}_{\mathbf{x}}) \setminus Y_{\mathbf{x}}$, or it passes through nodes in $\mathbf{R}^c$. We need to prove that the paths that pass through $\mathbf{R}^c$ are also blocked given $\mathbf{B}_j(\mathbf{Y}_{\mathbf{x}}) \setminus Y_{\mathbf{x}}$. Taking any $R \in \mathbf{R}^c$, for any path $p$ between $R$ and $U_j$, if $p$ does not pass through $Y_{\mathbf{x}}$, then $p$ is blocked given $\mathbf{B}_j(\mathbf{Y}_{\mathbf{x}}) \setminus Y_{\mathbf{x}}$, which means that any path from $\mathbf{Y}_*$ passing through $R$ and then through $p$ to reach $U_j$ is also blocked by $\mathbf{B}_j(\mathbf{Y}_{\mathbf{x}}) \setminus Y_{\mathbf{x}}$. If $p$ passes through $Y_{\mathbf{x}}$, it must pass through another $R' \in \mathbf{R}$ and eventually reach $U'$ through a path that does not include $Y_{\mathbf{x}}$, which is also blocked by $\mathbf{B}_j(\mathbf{Y}_{\mathbf{x}}) \setminus Y_{\mathbf{x}}$. Therefore, any path involving $Y_{\mathbf{x}}$ is blocked by $\mathbf{B}_j(\mathbf{Y}_{\mathbf{x}}) \setminus Y_{\mathbf{x}}$, which means that any path from $U_j$ to $Y_{\mathbf{x}}$ is also blocked. For colliders on ancestors of $Y_{\mathbf{x}}$, not conditioning on $Y_{\mathbf{x}}$ adds blocking to the paths, not removing any existing blocks; for $Y_{\mathbf{x}}$ being a non-collider, any path involving $Y_{\mathbf{x}}$ is blocked by $\mathbf{B}_j(\mathbf{Y}_{\mathbf{x}}) \setminus Y_{\mathbf{x}}$, thus remaining blocked even if $Y_{\mathbf{x}}$ is not conditioned on. Therefore, $Y_{\mathbf{x}}$ does not influence the blocking status of any relevant paths. Consequently, given $\mathbf{B}_j(\mathbf{Y}_{\mathbf{x}}) \setminus Y_{\mathbf{x}}$, all the paths between any other nodes in $\mathbf{Y}_{\mathbf{x}}$ (including $Y_{\mathbf{x}}$) and $U_j$ are blocked. This implies that $\mathbf{B}_j(\mathbf{Y}_{\mathbf{x}}) \setminus Y_{\mathbf{x}}$ is also a Markov blanket. However, it is a proper subset of $\mathbf{B}_j(\mathbf{Y}_{\mathbf{x}})$, which contradicts $\mathbf{B}_j(\mathbf{Y}_{\mathbf{x}})$ being a Markov boundary. Therefore, $Y_{\mathbf{x}} \in \mathbf{B}_j(\mathbf{Y}_{\mathbf{x}})$ implies $Y_{\mathbf{x}} \not\perp\!\!\!\perp_{\mathcal{G}^a_{\mathbf{x}}} U_j \mid \mathbf{Y}_{\mathbf{x}} \setminus \{Y_{\mathbf{x}}\}$.

(ii) $Y_{\mathbf{x}} \not\perp_{\mathcal{G}_{\mathbf{x}}^a} U_j \mid \mathbf{Y}_{\mathbf{x}} \setminus \{Y_{\mathbf{x}}\}$ implies $Y_{\mathbf{x}} \in \mathbf{B}_j(\mathbf{Y}_{\mathbf{x}})$: It can be proven that the contrapositive holds. If $Y_{\mathbf{x}}$ is not in the Markov boundary, then it is independent of $U_j$ given the Markov boundary $Y_{\mathbf{x}} \in \mathbf{B}_j(\mathbf{Y}_{\mathbf{x}})$. Clearly, this conclusion follows directly from the definition of the Markov blanket. Therefore, $Y_{\mathbf{x}} \not\perp_{\mathcal{G}_{\mathbf{x}}^a} U_j \mid \mathbf{Y}_{\mathbf{x}} \setminus \{Y_{\mathbf{x}}\}$ holds trivially. $\qquad \square$

Our final algorithm for efficiently computing Markov boundaries is obtained directly from d-separation, Thms. 2 and 3 as shown in Fig. 7. However, a limitation of these theorems is the reliance on the faithfulness assumption, which may not always hold, even in the case of a fully specified SCM, where numerical violations can potentially occur.

# B  Related Works

## B.1  Importance Sampling

To estimate Eq. 2, a straightforward approach is to start from the definition and employ rejection sampling, as implemented in [108]:

$$P(\mathcal{Y}_*) = \mathbb{E}_{\mathbf{u} \sim P_{\mathbf{u}}} \left[ \mathbb{1}_{\Omega_{\mathbf{U}}(\mathcal{Y}_*)}(\mathbf{u}) \right]. \tag{97}$$

This method operates effectively within a countable, finite exogenous space. However, as discussed in the main text, it exhibits low efficiency in general settings due to the infrequency with which the indicator function in the counterfactual probability expression is effective (i.e., non-zero). One solution to this problem is importance sampling, where selection of the proposal distribution is a critical issue. In this section, we discuss related works in detail.

**Cross-entropy based importance sampling**  It is straightforward to prove that when the density of the proposal distribution $q(\mathbf{u}) \propto p(\mathbf{u})|f(\mathbf{u})|$ (where, in the context of counterfactual estimation, $f(\mathbf{u})$ is $\mathbb{1}_{\Omega_{\mathbf{U}}(\mathcal{Y}_*)}(\mathbf{u})$), the corresponding importance sampling estimator possesses the minimum variance. This proposal distribution is known as the optimal proposal distribution, denoted as $Q_{\mathbf{U}}^*$. The idea behind a class of methods is to minimize the cross-entropy (or equivalently, the KL divergence) between the conditional proposal distribution $Q_{\mathbf{U}}$ and the optimal proposal distribution $Q_{\mathbf{U}}^*$. The expression for this in the context of counterfactual estimation is:

$$H(q^*, q) = -\mathbb{E}_{\mathbf{u} \sim Q_{\mathbf{U}}^*} \left[ \log q(\mathbf{u}) \right] \propto -\mathbb{E}_{\mathbf{u} \sim P_{\mathbf{U}}} \left[ \mathbb{1}_{\Omega_{\mathbf{U}}(\mathcal{Y}_*)}(\mathbf{u}) \log q(\mathbf{u}) \right]. \tag{98}$$

In the task of sampling rare events, due to the difficulty of directly sampling from $P_{\mathbf{u}}$ to obtain effective samples for which $f(\mathbf{u})$ is valid, it is common to estimate the cross-entropy by sampling from a proposal distribution $Q_{\mathbf{u}}$. This method, known as importance sampling, is considered adaptive because it continuously adapts based on the samples already drawn from the proposal distribution. For instance, [32] modeled $Q_{\mathbf{U}}$ as a Gaussian Mixture Model (GMM) and optimized an equivalent form of cross-entropy to improve sampling efficiency. When applied to counterfactual estimation, the corresponding optimization objective is:

$$\arg\min_q -\mathbb{E}_{\mathbf{u} \sim Q_{\mathbf{U}}} \left[ \frac{p(\mathbf{u}) \mathbb{1}_{\Omega_{\mathbf{U}}(\mathcal{Y}_*)}(\mathbf{u})}{q(\mathbf{u})} \log q(\mathbf{u}) \right]. \tag{99}$$

**Neural importance sampling**  In addition to GMM, normalizing flows provide a more expressive capability as density estimators. Therefore, as enumerated in the main text, numerous recent methods have leveraged normalizing flows as proposal distributions for importance sampling. Many of these methods still rely on optimizing cross-entropy (and KL divergence), as introduced earlier. Another optimization method discussed in [68] is based on the $\chi^2$ divergence between the optimal proposal distribution $Q_{\mathbf{U}}^*$ and the proposal distribution $Q_{\mathbf{U}}$:

$$D_{\chi^2}(q^*, q) = \int_{\Omega_{\mathbf{u}}} \frac{(q^*(\mathbf{u}) - q(\mathbf{u}))^2}{q(\mathbf{u})} d\mathbf{u} \tag{100}$$

$$= \int_{\Omega_{\mathbf{u}}} \frac{(q^*)^2(\mathbf{u})}{q(\mathbf{u})} d\mathbf{u} - \left( 2 \int_{\Omega_{\mathbf{u}}} q^*(\mathbf{u}) d\mathbf{u} - \int_{\Omega_{\mathbf{u}}} q(\mathbf{u}) d\mathbf{u} \right) \tag{101}$$

$$= \int_{\Omega_{\mathbf{u}}} \frac{(q^*)^2(\mathbf{u})}{q(\mathbf{u})} d\mathbf{u} - 1 \tag{102}$$

$$\propto \int_{\Omega_{\mathbf{u}}} \frac{p^2(\mathbf{u}) \mathbb{1}_{\Omega_{\mathbf{U}}(\mathcal{Y}_*)}(\mathbf{u})}{q(\mathbf{u})} d\mathbf{u} - 1 \tag{103}$$

$$= \mathbb{E}_{\mathbf{u} \sim P_{\mathbf{U}}} \left[ \frac{p(\mathbf{u})}{q(\mathbf{u})} \mathbb{1}_{\Omega_{\mathbf{U}}(\mathcal{Y}_*)}(\mathbf{u}) \right] - 1. \tag{104}$$

It can be seen that the only difference between Eq. 104 and Eq. 5 lies in a constant term. Thus, directly optimizing this equation is equivalent to optimizing the variance itself. Naturally, considering the difficulty of sampling from $P_{\mathbf{U}}$ as discussed in the paper, an adaptive approach is employed. This involves sampling from the proposal distribution $Q_{\mathbf{U}}$, with the corresponding optimization objective being:

$$\arg\min_q \mathbb{E}_{\mathbf{u} \sim Q_{\mathbf{U}}} \left[ \frac{p^2(\mathbf{u})}{q^2(\mathbf{u})} \mathbb{1}_{\Omega_{\mathbf{U}}(\mathcal{Y}_*)}(\mathbf{u}) \right]. \tag{105}$$

In [68], the proposal distribution $Q_{\mathbf{U}}$ is modeled as NICE [24].

**Extension and limitation**  The above method, without extensions, is only applicable to estimate a single counterfactual probability $P(\mathcal{Y}_*)$. We extend these methods similarly to our proposed approach by introducing a conditional proposal distribution $Q_{\mathbf{U}|\mathbf{Y}_*}$, and employing the same estimator as in Eq. 9 during inference as baselines. During the variance optimization phase, by introducing prior distributions $Q_{\mathcal{S}}$ and $P_{\mathbf{U}}$ and letting $\mathbf{y}_* = \mathbf{Y}_*^{(s)}(\mathbf{u}')$ be sampled from stochastic counterfactual processes ($\mathbf{u}' \sim P_{\mathbf{U}}$ and $s \sim Q_{\mathcal{S}}$), we reformulate the optimization expressions for cross-entropy based importance sampling (CEIS) as:

$$\arg\min_q -\mathbb{E}_{\mathbf{y}_*} \left[ \mathbb{E}_{\mathbf{u} \sim Q_{\mathbf{U}|\mathbf{y}_*}} \left[ \frac{p(\mathbf{u}) \mathbb{1}_{\Omega_{\mathbf{U}}(\delta(\mathbf{y}_*))}(\mathbf{u})}{q(\mathbf{u}|\mathbf{y}_*)} \log q(\mathbf{u}|\mathbf{y}_*) \right] \right], \tag{106}$$

and neural importance sampling (NIS) as:

$$\arg\min_q \mathbb{E}_{\mathbf{y}_*} \left[ \mathbb{E}_{\mathbf{u} \sim Q_{\mathbf{U}|\mathbf{y}_*}} \left[ \frac{p^2(\mathbf{u})}{q^2(\mathbf{u}|\mathbf{y}_*)} \mathbb{1}_{\Omega_{\mathbf{U}}(\delta(\mathbf{y}_*))}(\mathbf{u}) \right] \right]. \tag{107}$$

where $\delta(\mathbf{y}_*)$ denotes a small region around $\mathbf{y}_*$. The above optimization objective tasks $Q_{\mathbf{U}|\mathbf{y}_*}$ with sampling from $\Omega_{\mathbf{U}}(\delta(\mathbf{y}_*))$, where any $\mathcal{Y}_*$ can be decomposed into a set of $\mathbf{y}_*^{(j)} \in \mathcal{Y}_*$, such that $\mathcal{Y}_* = \bigcup_{j=1}^{\infty} \delta(\mathbf{y}_*^{(j)})$, and $\Omega_{\mathbf{U}}(\mathcal{Y}_*) = \bigcup_{j=1}^{\infty} \Omega_{\mathbf{U}}(\delta(\mathbf{y}_*^{(j)}))$.

An unavoidable drawback of the extended method is the presence of indicator functions in the optimization term. If the initial proposal distribution $Q_{\mathbf{u}}$ is similar to $P_{\mathbf{u}}$ in performance, the entire optimization process requires a prolonged warm-up period before effective samples start to emerge from the proposal distribution $Q_{\mathbf{u}}$. For example, if the entire space is the unit plane $[0,1]^2$ and $\Omega_{\mathbf{U}}(\delta(\mathbf{y}_*))$ is a small region $\{(x,y) \mid x \leq 0.001, y \leq 0.001\}$, then if the chosen initial proposal distribution is uniform, it has only a probability $10^{-6}$ of sampling an effective sample, resulting in a low efficiency for optimizing the loss functions.

In contrast, the optimization term in Exogenous Matching according to Thm. 1 does not include the indicator function, ensuring that every sample is effective during the optimization process, thus significantly enhancing learning efficiency.

### B.2  Proxy SCMs

In this section, we provide a detailed description of the construction of the proxy SCMs used in the experiments and their identifiability.

**Nerual Causal Model**  The Neural Causal Model (NCM, [107, 108]) is a recently proposed general approach to mimicking the original SCMs to construct proxy SCMs. For identifiability, its primary assumption is the causal graph of the original SCM. By replacing the causal mechanisms in the original SCM with neural networks that have a strong expressive power, it can effectively proxy the original SCM.

Specifically, an NCM is a 4-tuple $\langle \mathbf{U}, \mathbf{V}, \widehat{\mathcal{F}}, \widehat{P}_{\mathbf{U}} \rangle$, which is almost identical to the SCM definition, except that each $f_{V_i} \in \widehat{\mathcal{F}}$ is a neural network with sufficient expressive power, and $\widehat{P}_{\mathbf{U}}$ is any well-defined distribution over $\mathbf{U}$.

Their method for inducing submodels is the same as for a regular SCM, i.e., replacing causal mechanisms under perfect interventions, which they term neural mutilation. For counterfactual estimation, they use the rejection sampling introduced earlier because the endogenous variables in their experimental scenarios are discrete and finite, making rejection sampling efficient enough to sample effective samples.

Regarding identifiability, this work assumes causal graphs and has been shown to operate effectively when endogenous variables are discrete and finite. Specifically, they construct a class of models called $\mathcal{G}$-constrained NCMs, which have the same causal graph $\mathcal{G}$ as the original SCM. Then, they prove that any NCM in the $\mathcal{G}$-constrained NCM family has dual identifiability with the causal graph $\mathcal{G}$, i.e., the NCMs in the $\mathcal{G}$-constrained NCM family (given some observational or interventional distributions) output the same result when answering a $\mathcal{Q} \in \mathcal{L}_3$, if and only if the graph algorithms for counterfactual identification output identifiable results given input $\mathcal{Q}$ and observational or interventional distributions.

Furthermore, based on this duality, they develop a sound and complete identification algorithm, and since this identification algorithm is based on the learning process of NCMs, it only requires gradient optimization. Certainly, although this work guarantees theoretical results for identifiability in discrete and finite settings [108] as well as partial identifiability in Lipschitz continuous settings [93], further improvements are necessary to apply its estimation methods in broader scenarios.

**Causal Normalizing Flow**   Causal Normalizing Flow (CausalNF, [42]) is a recently summarized method to construct proxies of SCMs based on flow models. For identifiability, it assumes the causal graph or causal ordering of the true SCM, and the true SCM is Markovian and diffeomorphic.

Specifically, a CausalNF is a 4-tuple $\langle \mathbf{U}, \mathbf{V}, \mathcal{T}^{-1}, \widehat{P}_{\mathbf{U}} \rangle$, where $\mathcal{T} = \{f_{V_i}(u_i; \mathbf{pa}_{V_i}) \mid V_i \in \mathbf{V}\}$ is a sequence-preserving Autoregressive Normalizing Flow (ANF), and $\widehat{P}_{\mathbf{U}}$ is the base distribution of this flow model.

Their approach to derive submodels differs from conventional SCMs because the modeled CausalNFs have multiple architectures, and substituting causal mechanisms is only applicable in the recursive case. Their solution is to alter exogenous distributions and leverage the Markov assumption to ensure that the density of the intervened exogenous distribution is:

$$\hat{p}_{\mathbf{u}[\mathbf{x}]}(\mathbf{u}) = \left( \prod_{V_i \in \mathbf{X}} \delta(\{v_i = f_{V_i}(u_i; \mathbf{pa}_{V_i})\}) \right) \cdot \prod_{V_j \notin \mathbf{X}} p(u_j). \tag{108}$$

$\delta$ denotes the Dirac distribution, where a density only exists when the condition inside its parentheses is satisfied. In practice, the distribution of the intervention is determined by the following steps: i) Sample $\mathbf{u}$ from $\widehat{P}_{\mathbf{U}}$; ii) Compute $\mathbf{v}_1 = \mathcal{T}^{-1}(\mathbf{u})$; iii) Replace $\mathbf{v}_1$, resulting in $\mathbf{v}_2 = \{v_i \mid v_i \in \mathbf{v}_1, v_i \notin \mathbf{x}\} \cup \mathbf{x}$; iv) Compute $\mathbf{u}_2 = \mathcal{T}(\mathbf{v}_2)$; v) Replace $\mathbf{u}_2$, yielding $\mathbf{u}_3 = \{u_i \mid u_i \in \mathbf{u}_2, v_i \notin \mathbf{x}\} \cup \{u_i \mid u_i \in \mathbf{u}_1, v_i \notin \mathbf{x}\}$; vi) Compute $\mathbf{v}_{\mathbf{x}} = \mathcal{T}^{-1}(\mathbf{u}_3)$. Here, step (v) corresponds to replacing relevant parts of the density of the intervened exogenous distribution with the Dirac distribution, as shown in Eq. 108. The last four steps among these six steps are employed for counterfactual reasoning, but this form of counterfactual reasoning is not for generalized cases. Specifically, the counterfactual query corresponding to these four steps is $P(\mathbf{V}_{\mathbf{x}} \mid \mathbf{V}_1)$. Although the submodel cannot be directly obtained, the $\mathbf{v}_{\mathbf{x}}$ obtained from the above steps is indeed the potential response $\mathbf{V}_{\mathbf{x}}(\mathbf{u})$, so our method is still applicable to this model.

As for identifiability, this work requires numerous assumptions, including Markovianity, diffeomorphism, causal order, and uses results from [106] to assist in proving the identifiability of the learned model up to invertible transformations.

## B.3  Arbitrary Conditioning

There is a challenge of modeling an exponentially large number of conditional distributions with a single model. Several existing works have focused on this problem, which is termed arbitrary conditioning. Arbitrary conditioning is an unsupervised learning task that, given a dataset over $\mathbf{V}$, aims to answer any query of the form $P(\mathbf{X} \mid \mathbf{Y} = \mathbf{y})$, where $\mathbf{Y} \subseteq \mathbf{V}$ and $\mathbf{y} \in \Omega_{\mathbf{Y}}$ are arbitrary.

The initial step to address the problem of arbitrary conditioning is to encode the condition $\mathbf{y}$ into a fixed-length vector. Subsequently, this vector is learned through various modeling methods, and the

training process incorporates various random masks to simulate different forms of $\mathbf{Y}$. Specifically, related works introduce a binary mask $\mathbf{b}$ to indicate whether a variable has been observed. In this context, the conditional probability can be expressed as $P(\mathbf{V}_{1-\mathbf{b}} \,|\, \mathbf{v_b}, \mathbf{b})$. Consequently, the learning objective can equivalently be viewed as maximizing the following log-likelihood term:

$$\arg \max_q \mathbb{E}_{\mathbf{b} \sim P_{\mathbf{B}}} \left[ \mathbb{E}_{\mathbf{v} \sim P_{\mathbf{V}}} \left[ \log q(\mathbf{v}_{1-\mathbf{b}} \,|\, \mathbf{v_b}, \mathbf{b}) \right] \right], \tag{109}$$

where $P_{\mathbf{b}}$ represents the prior distribution of the random mask, while $P_{\mathbf{V}}$ denotes the observational distribution. The model $q$ corresponds to the probabilistic model employed for this task. Specific methods employed in related works are enumerated in the main text, involving a variety of generative models that are suited to different task scenarios. For instance, some methods permit direct training with partially observed data, some can only output samples without density estimation, and others are incapable of performing efficient marginal probability computation.

The similarity between this work and these methods lies in the encoding scheme and the training objectives. For example, the conditional encoding method in this paper also employs binary masks, introducing both the observation mask $\omega(\mathbf{X})$ and the intervention mask $\omega(\mathbf{Y})$. Furthermore, the training objective, as defined in Eq. 8, closely resembles Eq. 109.

The distinction between our work and these methods lies in the available data and the target tasks. The problem of arbitrary conditioning is based on learning from the observed distribution and is typically applied to imputation tasks. In contrast, our conditional distribution modeling can be seen as an extension of arbitrary conditioning, generalizing the observed distribution to counterfactual distributions. This allows the conditions to originate from multiple hypothetical worlds and is applied in counterfactual reasoning.

The work most similar to ours in terms of overall structure and optimization objectives is Posterior Matching [91]. It addresses the problem of arbitrary conditioning by matching the encoder of a given VAE. The optimization formulation is as follows:

$$\arg \min_q \mathbb{E}_{\mathbf{b} \sim P_{\mathbf{B}}} \left[ \mathbb{E}_{\mathbf{v} \sim P_{\mathbf{V}}} \left[ \mathbb{E}_{\mathbf{u} \sim P_{\mathbf{U}|\mathbf{v}}} \left[ \log q(\mathbf{u} \,|\, \mathbf{v_b}, \mathbf{b}) \right] \right] \right], \tag{110}$$

where $P_{\mathbf{U}|\mathbf{v}}$ represents the posterior distribution in a VAE, also referred to as the encoder. By comparing the optimization objective of our method, Eq. 8, with Eq. 110, we observe a notable similarity between the two. We can establish a one-to-one correspondence between the terms in the optimization formulations: the mask $\mathbf{b}$ can be analogized to the state $s$ in the stochastic counterfactual process; the decoder of the VAE (mapping from latent variables to observable variables) corresponds to the SCM's potential responses (mapping from exogenous to endogenous variables); the encoder of the VAE (mapping from observable variables to latent variables) corresponds to the abduction process in the SCM (mapping from endogenous to exogenous variables). Consequently, the posterior distribution aligns with the abduction process in the SCM, represented by the product of the indicator function and the exogenous distribution.

An intuitive explanation is that posterior matching attempts to align the conditional distribution with the posterior distribution, whereas our work can be interpreted as attempting to align the conditional distribution with the optimal proposal distribution. The interpretation of the optimal proposal distribution is detailed in App. B.1, where in the counterfactual estimation task, the optimal proposal distribution is proportional to the product of the indicator function and the exogenous distribution.

## B.4 Injecting Prior Knowledge

This work involves injecting prior structure into the learning process. Specifically, such methods of injecting prior structure are achieved by directly influencing the internal connectivity of the model.

Related work includes MADE [31], Zuko [81], and StrNN [13]. MADE interprets from the perspective of conditional probability, altering the connectivity between inputs and outputs in autoregressive models by masking, thereby enabling the model to output specific conditional distributions $p(x_i \,|\, \mathbf{y})$ only when there exists connectivity from inputs in $\mathbf{y}$ to output $x_i$; however, this method only respects autoregressive properties. The method in Zuko is primarily used for implementing masked autoregressive normalizing flows, but is not limited to autoregressive dependencies. Based on the interpretation of Jacobian matrices, it alters the connectivity between inputs and outputs in MLPs through masking, ensuring that the Jacobian matrix of MLP outputs is non-zero only when the

mask is one, thus representing the dependency between input and output variables. However, the masking algorithm in Zuko only approximately preserves the maximum number of connections under dependency constraints in variable order. StrNN extensively investigates the issue of variable dependency and connectivity, formalizing it as an integer programming problem, and providing exact and greedy algorithms that outperform the previous two methods in terms of performance.

The tasks typically addressed by the above methods include representing autoregression (autoregressive models), constraining the dependency of Bayesian networks (Bayesian network inference), and constraining the dependency of causal graphs (causal inference). Different from these tasks, this paper applies them to representing the dependency of Markov boundary (Markov boundary feature selection). The masking algorithm adopted in this paper is Zuko, as it can be directly combined with the normalizing flows it implements and used directly in subsequent experiments.

## C  Experiments

### C.1  Metrics

We formally introduce all the metrics used in the subsequent experiments, which aim to reflect the quality of counterfactual estimation:

**Effective Sample Proportion (ESP)**     As mentioned earlier, effective samples here refer to samples that make the indicator function equal to one (i.e., the constraints of potential responses are satisfied). For a specific $\mathcal{Y}_*$, we define

$$\eta(\mathcal{Y}_*) = \frac{1}{n} \left( \sum_{i=1}^{n} \mathbb{1}_{\Omega_{\mathbf{U}}(\mathcal{Y}_*)}(\mathbf{u}^{(i)}) \right), \tag{111}$$

which reflects the proportion of effective samples sampled in a counterfactual estimation for $\mathcal{Y}_*$. In rare event sampling, this metric essentially represents the success rate, indicating the proportion of samples in which the rare event occurs. ESP is about the expectation of this function over a prior distribution $P_{\mathcal{Y}_*}$ (related to the stochastic counterfactual process):

$$\text{ESP} = \mathbb{E}_{\mathcal{Y}_* \sim P_{\mathcal{Y}_*}} [\eta(\mathcal{Y}_*)]. \tag{112}$$

Specifically, ESP reflects the sampling efficiency when the importance weights are ignored. It focuses on the probability that the sampled $\mathbf{u}$ falls within the support $\Omega_{\mathbf{U}}(\mathcal{Y}_*)$, for all $\mathcal{Y}_*$.

**Failure Rate (FR)**     This work claims that the proposed method holds the potential for counterfactual estimation in any form following a single training session. To verify the reliability of this claim, we introduce the FR metric, which reflects the efficacy of the proposal distribution post a single training across the entire state space of the relevant stochastic counterfactual processes, defined as follows:

$$\text{FR} = \mathbb{E}_{\mathcal{Y}_* \sim P_{\mathcal{Y}_*}} \left[ \begin{cases} 1, & \eta(\mathcal{Y}_*) \leq m \\ 0, & \text{otherwise} \end{cases} \right], \tag{113}$$

where $0 < m \leq 1$ is a constant representing the acceptance threshold. An $\eta(\mathcal{Y}_*)$ below this threshold is considered a failure. The closer the FR is to zero, the more it indicates that for the vast majority of $\mathcal{Y}_*$, there are sufficiently many samples of $\mathbf{u}$ falling within the support $\Omega_{\mathbf{U}}(\mathcal{Y}_*)$, making the counterfactual estimation considered tractable.

### C.2  Design of Stochastic Counterfactual Processes

This paper introduces a stochastic counterfactual process (Def. 1), allowing the proposal distribution after a single training to potentially estimate counterfactual probabilities in multiple or even infinite forms in a tractable manner. This facilitates efficient estimation of a $\mathcal{L}_3$ expression in real-world scenarios (assuming that the $\mathcal{L}_3$ expression can be approximated by a finite set of counterfactual probabilities). Therefore, if the proposal distribution after a single training can tractably estimate these counterfactual probabilities, it implies that estimation of the $\mathcal{L}_3$ expression only requires one training. Specifically, employing stochastic counterfactual processes, this paper conducts stochastic optimization on various forms of counterfactuals and demonstrates its feasibility (Cor. 1). In this section, we elaborate on its specific design in experiments.

Table 3: The state space of $\mathfrak{Y}_*^{\mathcal{Q}}$ (i.e., all involved counterfactual variables), as well as the expression of $\mathcal{Q}$. $\mathcal{Q} \in \{\text{ATE}, \text{ETT}, \text{NDE}, \text{CtfDE}\}$.

| $\mathcal{Q}$ | Counterfactual Variables ($\mathbf{Y}_*$) | Expression |
|---|---|---|
| ATE | $Y_{X=1}$ 
 $Y_{X=0}$ | $P(Y_{X=1} = 1) - P(Y_{X=0} = 1)$ |
| ETT | $\{Y_{X=1}, X\}$ 
 $\{Y_{X=0}, X\}$ 
 $X$ | $\frac{P(Y_{X=1}=1, X=1) - P(Y_{X=0}=1, X=1)}{P(X=1)}$ |
| NDE | $\{Y_{X=1,W=w}, W_{X=0}\}, w \in \Omega_W$ 
 $Y_{X=0}$ | $\sum_w P(Y_{X=1,W=w} = 1, W_{X=0} = w) - P(Y_{X=0} = 1)$ |
| CtfDE | $\{Y_{X=1,W=w}, W_{X=0}\}, w \in \Omega_W$ 
 $Y_{X=0}$ 
 $X$ | $\frac{\sum_w P(Y_{X=1,W=w} = 1, W_{X=0} = w) - P(Y_{X=0} = 1)}{P(X = 1)}$ |

**Construction of $\mathfrak{Y}_*^{\mathcal{B}}$**  If $\mathcal{Q} \in \mathcal{L}_3$ is an arbitrary $P(\mathcal{Y}_*)$ such that $\mathcal{Y}_* \subseteq \Omega_{\mathbf{Y}_*}$ is ensured, and precisely involves $k$ distinct submodels, then we can design $\mathfrak{Y}_*^{\mathcal{B}}$ to answer any such $\mathcal{Q}$. Specifically, the state space of $\mathfrak{Y}_*^{\mathcal{B}}$ is $\mathcal{S} = \{s \mid |s| = k\}$.

Here, we design the prior distribution $Q_s^{\mathcal{B}}$ to cover $\mathcal{S}$:

$$\mathbf{b}_{1j}^{(i)} \sim \text{Bernoulli}(\rho_1), \qquad\qquad 0 < \rho_1 < 1, \qquad\qquad (114)$$

$$\mathbf{b}_1^{(i)} = \{\mathbf{b}_{1j}^{(i)} \mid V_j \in \mathbf{V}\}, \qquad\qquad\qquad\qquad (115)$$

$$\mathbf{X}^{(i)} = \{V_j \mid b_{1j}^{(i)} = 1\}, \qquad\qquad\qquad\qquad (116)$$

$$\mathbf{x}^{(i)} \sim P_{\mathbf{X}^{(i)}}, \qquad\qquad p(\mathbf{x}^{(i)}) > 0, \qquad\qquad (117)$$

$$\mathbf{b}_{2j}^{(i)} \sim \text{Bernoulli}(\rho_2), \qquad\qquad 0 < \rho_2 < 1, \qquad\qquad (118)$$

$$\mathbf{b}_2^{(i)} = \{\mathbf{b}_{2j}^{(i)} \mid V_j \in \mathbf{V} \setminus \mathbf{X}^{(i)}\}, \qquad\qquad\qquad\qquad (119)$$

$$\mathbf{Y}^{(i)} = \{V_j \mid b_{2j}^{(i)} = 1\}, \qquad\qquad\qquad\qquad (120)$$

$$\mathbf{Y}_* = \{\mathbf{Y}_{\mathbf{x}^{(i)}}^{(i)} \mid i \leq k\}, \qquad\qquad\qquad\qquad (121)$$

where $\text{Bernoulli}(\rho_1)$ is a Bernoulli distribution with parameter $\rho_1$, meaning it has a probability of $\rho_1$ to yield 1; $P_{\mathbf{X}}$ is the endogenous distribution corresponding to $\mathbf{X}$ and for any $\mathbf{x} \in \Omega_{\mathbf{X}}$, $p(\mathbf{x}) > 0$.

This distribution, due to the constrained Bernoulli distribution parameters ($0 < \rho_1, \rho_2 < 1$), implies that any $\mathbf{X}, \mathbf{Y} \subseteq V$ may potentially be sampled; and for intervention values, assuming positivity ($p(\mathbf{x}^{(i)}) > 0$), any possible intervention value can be sampled. Therefore, for any of these submodels, any counterfactual variable is covered, and hence any variable set is also covered.

In subsequent experiments, we set $\rho_1 = 0.2$ and $\rho_2 = 0.75$, which are utilized during both the training and testing phases. The rationale behind employing $\mathfrak{Y}_*^{\mathcal{B}}$ for experiments lies in the large size of $\mathcal{S}$, which provides more credible evidence to verify whether the conclusion of Cor. 1 satisfies the claim and whether the designs bridge the gap between theory and practice.

**Construction of $\mathfrak{Y}_*^{\mathcal{Q}}$**  To apply our method to real-world tasks, we construct stochastic counterfactual processes $\mathfrak{Y}_*^{\mathcal{Q}}$ for several common causal query tasks, where $\mathcal{Q} \in \{\text{ATE}, \text{ETT}, \text{NDE}, \text{CtfDE}\}$.

As shown in Tab. 3, the state spaces for these queries are finite or easily sampled, allowing the design of the prior distribution $Q_{\mathbf{S}}^{\mathcal{Q}}$ to be a simple uniform distribution over these available states. We assume that the endogenous variable space is discrete and finite, which means that we only need to estimate the counterfactual probability for each of them. By computing according to their expressions, we can obtain the estimate for $\mathcal{Q}$.

## C.3  Fully-specified SCMs

In this section, we briefly introduce all fully specified SCMs used in our experiments.

**Markovian diffeomorphic SCMs**  These SCMs are sourced from [42] and are primarily used in the experiments for CausalNF, including: CHAIN-LIN-3, CHAIN-NLIN-3, CHAIN-LIN-4, CHAIN-LIN-5, COLLIDER-LIN, FORK-LIN, FORK-NLIN, LARGEBD-NLIN, SIMPSON-NLIN, SIMPSON-SYMPROD, TRIANGLE-LIN, and TRIANGLE-NLIN. Markovianity denotes the absence of hidden confounders in these SCMs, meaning that each exogenous variable is involved in the causal mechanisms of only one endogenous variable. The diffeomorphism is a parametric assumption, positing that the transformations between exogenous and endogenous variables satisfy the properties of a diffeomorphism. For detailed construction of these SCMs, see the Appendix of [42].

**Semi-Markovian continuous SCMs**  These SCMs are described in [107]. We have reformulated their functional forms to adapt them to continuous scenarios. Specifically, this involves replacing their causal mechanisms with combinations of linear and nonlinear functions, all of which originate from the Markovian diffeomorphic SCMs already present above. These SCMs include: BACK-DOOR, FRONT-DOOR, M, and NAPKIN. Here, Semi-Markovianity indicates that there may be latent confounders within these SCMs, yet they still satisfy recursiveness. Continuity refers to the assumptions regarding their parameters.

**Regional canonical SCMs**  These SCMs are originated from [108], where all endogenous variables are discrete and finite (binary), with complexity arises from the causal mechanisms being a type of stochastic mapping. These SCMs include: FAIRNESS, FAIRNESS-XW, FAIRNESS-XY, FAIRNESS-YW, where subscripts indicate the presence of hidden confounders between variables. Please refer to the Appendix of [108] for the specific constructions of these SCMs.

## C.4  Reproducibility

In this section, we present details pertinent to reproducibility, encompassing the complete architecture of the models utilized, the computational resources employed, and the execution and random seeds of the reported experiments.

**Architecture details**  The entire model architecture involves three components: the sampler, the conditioning model, and the density estimation model.

The sampler is used to sample $s$ from the prior distribution $\mathcal{Q}_{\mathcal{S}}$ of the stochastic counterfactual process and $\mathbf{u}$ from the exogenous distribution $P_{\mathbf{U}}$, then compute the potential outcome $\mathbf{Y}_*^{(s)}(\mathbf{u})$. In our experiments, we selected a batch size of 256 and generated a training set of size 16,384 through the sampler. The validation set is also generated using this sampler. Specifically, we utilize the same sampler to construct $P_{\mathcal{Y}_*}$, from which random $\mathbf{y}_*$ is sampled. For discrete distributions, counterfactual events are assigned as $\mathcal{Y}_* \in \{\mathbf{y}_*\}$, and for continuous distributions, counterfactual events are assigned as $\mathcal{Y}_* \in \delta_l(\mathbf{y}_*)$, where $\delta_l(\mathbf{y}_*)$ is a cube centered at $\mathbf{y}_*$ with side length $l$.

A brief overview of the conditioning model is shown in Fig. 2, where both $g$ and $h$ are neural networks. When masking with Markov boundaries, $h$ is a neural network that allows dynamic influence on internal connectivity, as depicted in App. B.4. On the other hand, $g$ is a permutation-invariant neural network used to aggregate hidden encodings from multiple different submodels. We conduct ablation studies related to the construction of these two neural networks in App. C.8.

For the density estimation model, we chose normalizing flows and GMM. Specific models for normalizing flows include MAF [73], NSF [26], NCSF [80], NICE [24], NAF [39], and SOSPF [41]. The implementations of these models are based on the Zuko library [81], with some internal implementation details modified to fit our work. Regarding the hyperparameters of these models, the number of components in GMM is fixed at $10$, the number of transformations for all flow models is set to $5$, and all the neural networks involved contain 2 hidden layers with 64 neurons each (in some experiments, 256 neurons were used, which will be specifically indicated and discussed in subsequent sections).

**Hardware**    Each experiment was conducted on GPUs, and the primary computational overhead arising from the sampling process. All models, including the pre-trained SCM agents, were trained and tested on an NVIDIA RTX 4090 and Intel(R) Xeon(R) Gold 6430 platform. The training time for EXOM is primarily influenced by the number of variables in the SCM, the number of submodels, and the chosen density estimation model. For example, in the case of SIMPSON-NLIN with $|s| = 5$, one epoch (16,384 samples, batch size of 256) including the time for the sampler to prepare data takes approximately 8 seconds. In all EXOM experimental setups, we fixed the training to 200 epochs. All experiments, including the unreported ones, consumed approximately 1000 GPU hours in total.

**Execution**    To reflect the error, each type of experiment was conducted five times with different random seeds, and the range of errors exhibited is reflected in the experimental results. During training, EXOM does not require sampling and is trained for a fixed 200 epochs. The sampling size for CEIS is set to $10^5$ and trained for 200 epochs, whereas NIS has a sampling size of $10^4$ and is trained for 50 epochs (we found that its training speed is excessively slow due to the time-consuming nature of sampling from the flow model). During validation and testing, the validation set queries 1024 counterfactual probabilities from the sampler, with sampling sizes for EXOM, CEIS, and NIS set to $10^3$, and RS set to $10^6$. For all EXOM, CEIS, and NIS training, we used AdamW [62] with an initial learning rate of 0.001 as the optimizer and ReduceOnPlateau with a patience of 5 and a factor of 0.5 as the learning rate scheduler. To improve training robustness, the conditional inputs $\mathbf{y}_*$ and both the input and the output exogenous samples $\mathbf{u}$ were standardized. During inference, the probability densities and importance weights were processed in logarithmic form, with the terms where the indicator function is zero replaced by $-\inf$, and the log-sum-exp technique was used to estimate the probabilities.

### C.5    Convergence

Here, we present additional results on convergence experiments. Our experiments span across combinations of 4 different types of SCMs (SIMPSON-NLIN, LARGEBD-NLIN, FAIRNESS-XW, NAPKIN) and 4 different density estimation models (GMM, MAF, NICE, SOSPF). All models were trained and validated based on the stochastic counterfactual process $\mathfrak{Y}_*^{\mathcal{B}}$, which provides a sufficiently large state space to increase difficulty, where $|s| = 3$.

As shown in Fig. 8, for all combinations, as LL (log-likelihood, which is the negative form of the training objective Eq. 8, minimizing Eq. 8 is equivalent to maximizing LL) increases, ESP improves. In particular, there is a clear inflection point in the experiments with three continuous SCMs, after which the ESP rapidly improves until it stops increasing because of the convergence of LL. This series of experiments indicates that we can indeed optimize the upper bound (indirectly equivalent to LL) to optimize the variance (as discussed in App. C.1, ESP and variance are strongly related). Thus, empirically, these experiments support the correctness of Thm. 1, and also empirically indicate that our design potentially satisfies the bounded importance weight condition discussed in App. A.2.

Fig. 9 reflects the relationship between LL and FR. It is observed that for all combinations, as LL increases, FR decreases (with one exception in the SOSPF and LARGEBD-NLIN combination) until it stops increasing due to LL convergence. This phenomenon empirically supports the conclusion in

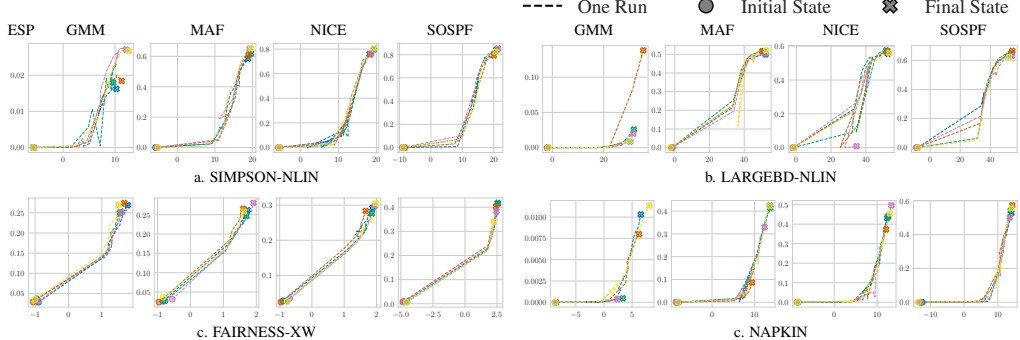

Figure 8: The relationship between ESP and LL on 4 SCMs with distinct types and 4 different density estimation models. Within these combinations, ESP always increases with the improvement of LL.

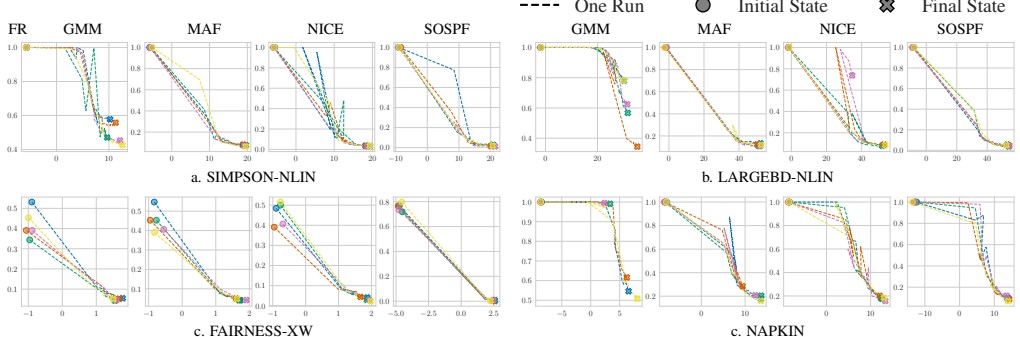

Figure 9: The relationship between FR and LL on 4 SCMs with distinct types and 4 different density estimation models. Within these combinations, FR always decreases with the improvement of LL.

Cor. 1, and manifests the claim of simultaneously optimizing the variance of counterfactual estimators across the entire state space.

## C.6    Comparison

We provide more comprehensive and detailed information regarding the comparison experiment in the main text. These comparisons are conducted on three different types of SCMs (SIMPSON-NLIN, NAPKIN, FAIRNESS-XW). EXOM employs two density estimation models (MAF, GMM), while NIS utilizes NICE, and CEIS uses GMM. All models are trained and validated based on a stochastic counterfactual process $\mathfrak{Y}_*^{\mathcal{B}}$, where $|s| = 1, 3, 5$.

As illustrated in Tab. 1, overall, our proposed method, EXOM, outperforms other importance sampling methods, while CEIS and NIS show comparable performance in the aforementioned experiments. The advantage of EXOM is particularly pronounced when dealing with multiple distinct submodels, significantly surpassing the other methods. All three importance sampling methods outperform RS, demonstrating that variance optimization indeed enhances estimation efficiency.

## C.7    Combinaitons of SCMs and Density Estimation Models

In this section, we detail the performance of EXOM when using various density estimation models (GMM, MAF, NSF, NCSF, NICE, NAF and SOSPF) in different SCMs enumerated in App. C.3. These experiments are based on $\mathfrak{Y}_*^{\mathcal{B}}$, with $|s| = 3$.

In Figs. 10a and 11a, deeper colors indicate a higher number of effective samples, which implies greater sampling efficiency. For density estimation models, it is evident that MAF and NSF generally perform well, while other density estimation models are suitable for specific scenarios. Regarding SCM, we observe that FORK-LIN and TRIANGLE-NLIN perform worse than other SCMs, whereas some SCMs such as CHAIN-LIN-5, LARGEBD-NLIN, and NAPKIN show compatibility only with certain density estimation models. The performance improved significantly after increasing the hidden layer width from 64 to 256.

In Figs. 10b and 11b, lighter colors indicate fewer failure cases, meaning that the method works effectively within the state space $\mathfrak{Y}_*^{\mathcal{Q}}$. It can be noted that for any of the aforementioned SCMs, there is always a density estimation model that matches and keeps the FR at a low level. Furthermore, compared with Figs. 10a and 11a, there is a strong negative correlation between FR and ESP, consistent with the theoretical results. In regions where FR is particularly high, in addition to the poor performance of GMM due to its limited expressive capacity, there are some notable aspects. For example, FORK-NLIN and TRIANGLE-NLIN are the most challenging settings to learn. Furthermore, as the diameter of the SCM increases, the performance of EXOM gradually decreases, which is reflected in the performance under the CHAIN-LIN-$N$ series settings. Finally, comparing Figs. 10b and 11b, some models with high parameter number requirements (such as NSF, NCSF, NAP, SOSPF) show significant improvements when the width of the hidden layers is increased.

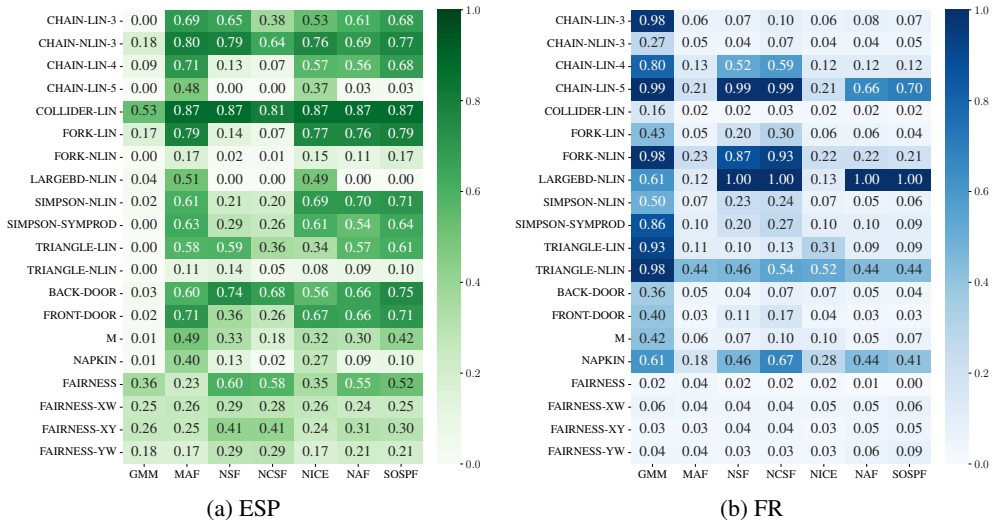

Figure 10: Performance on 20 SCMs and 7 density estimation models using EXOM. The experiment is based on the stochastic counterfactual process $\mathfrak{Y}_*^{\mathcal{B}}$ where $|s| = 3$. The width of the hidden layer is set to 64.

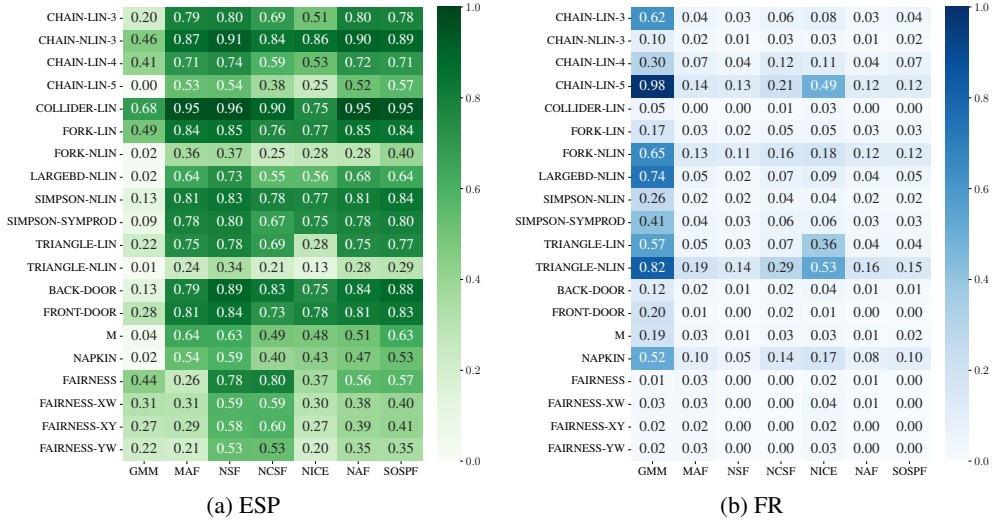

Figure 11: (Fig. 10 continued). Performance on 20 SCMs and 7 density estimation models using EXOM. The experiment is based on the stochastic counterfactual process $\mathfrak{Y}_*^{\mathcal{B}}$ where $|s| = 3$. The width of the hidden layer is set to 256.

**The impact of hidden layer width on performance** Experiments show that increasing the width of hidden layers significantly enhances performance. This improvement is not only due to the increased expressive power of the neural network itself but also because the experiments mentioned above utilize the Markov boundary masking trick. According to the description in App. B.4, we directly inject the Markov boundary masking into the weight matrix, masking certain weights (i.e., specific neurons are always inactive for specific tasks). This implies that when the graph structure of SCM is sparse, the Markov boundary is typically smaller, which leads to a substantial increase in the number of masked weights, thereby significantly reducing the neural network's expressive power. Therefore, increasing the width of hidden layers helps mitigate the degradation in expressive power caused by injecting the Markov boundary.

In practice, the recommended width is closely related to the number of parameters required by the conditional distribution models. For instance, models such as MAF, NICE, and GMM require fewer

parameters, so setting the hidden layer width to 64 is sufficient. In contrast, models like NSF, NCSF, NAF, and SOSPF, which require a larger number of parameters, will need a wider hidden layer — 256 is recommended — since, with a smaller Markov boundary, each parameter connection's hidden neurons are insufficient. Optimal width configuration should be fine-tuned according to the settings.

### C.8 Ablation

Our ablation experiments were conducted on the architecture of the two neural networks $g$ and $h$ in the conditioning model.

**Markov boundary masking** Let us first introduce how the mechanism of Markov boundaries operates on specific density estimation models. Firstly, we calculate the Markov boundary corresponding to each $U_i$ according to Theorem 3, and generate masks $\mathbf{m}_i$ using the method described later. The subsequent processing is then model-specific.

1. For **GMM**, $h$ primarily outputs each component's mean, diagonal matrix, and anti-diagonal matrix. Here, we apply the mask $\mathbf{m}_i$ only to the mean corresponding to the $i$-th exogenous variable in each component, ensuring that if the counterfactual variable is not on the Markov boundary, the inference of the mean is independent of it.
2. For **Autoregressive models** (including MAF, NSF, NCSF, UNF, and SOSPF), each layer transformation follows an autoregressive sequence, where all previous layers' hidden encodings may influence the subsequent layer. Consequently, each $U_i$'s hidden encoding appears in each layer. Therefore, we only need to directly apply the mask to the inference of hidden encodings, ensuring that the inferred next layer hidden encodings are only relevant to the counterfactual variables when they are in the Markov boundary.
3. For **Coupling models** (including NICE), each layer transformation involves only a subset of hidden encodings from the previous layer; hence, each layer's output only corresponds to some $U_i$. Thus, we create masks only for the $U_i$ that serve as the output for each layer transformation, ensuring that the inferred hidden encodings are only relevant to the counterfactual variables in the Markov boundary.

Of course, the above initial design will raise some issues in practice:

1. There are some **inevitable discrepancies between theory and practice**, which we have summarized in App. D.3.
2. Another issue worth discussing is how to handle the **counterfactual Markov boundaries in submodels**. In perfect interventions, all connections between intervened variables and their parent variables are cut (All Cut), which is also a way to represent causal graphs in submodels. However, theoretically, directly cutting connections would result in intervened variables never participating in the conditional reasoning of their parent exogenous variables. Hence, we have also devised two relaxed and compromised methods: one is to retain all connections (No Cut), pretending that the submodel has not been intervened upon, thus not reflecting the intervention through the mask of the Markov boundary but instead relying on the intervention information included in the conditional encoding; the second is to only retain connections with parent exogenous variables (Endo. Cut), while relationships between endogenous variables are still cut.
3. As discussed in App. C.7, the introduction of the Markov boundary theoretically increases the model's focus on specific variables. However, this comes **at the cost of expressiveness**, especially in models with a larger number of parameters. Therefore, in the ablation experiments, we use a hidden layer width of 256 for NICE and SOSPF, while for GMM and MAF, the hidden layer width is set to 64. This does not affect the fairness of the ablation, as the goal is to compare the performance difference with and without the Markov boundary. Under the same hidden layer width, the presence of a Markov boundary actually means fewer neurons are involved in reasoning.

Fig. 12 and Fig. 13 respectively illustrate the effects of Markov boundary masks on ESP and FR under different SCMs and density estimation models, where error bars and outliers are computed based on the quartile algorithm. The first phenomenon that can be observed is that introducing Markov boundary masks in Markovian SCMs (SIMPSON-NLIN, LARGEBD-NLIN, TRIANGLE-NLIN) results in significant performance improvement compared to not introducing them. There is one exception, namely LARGEBD-NLIN and SIMPSON-NLIN, which will be discussed in App. D.3. When applied to semi-Markovian SCMs (M, NAPKIN), the scenario changes, with the improvements becoming less pronounced and the effects post-application not being very stable. Across all intervention designs (All Cut, No Cut, Endo. Cut), these methods perform similarly

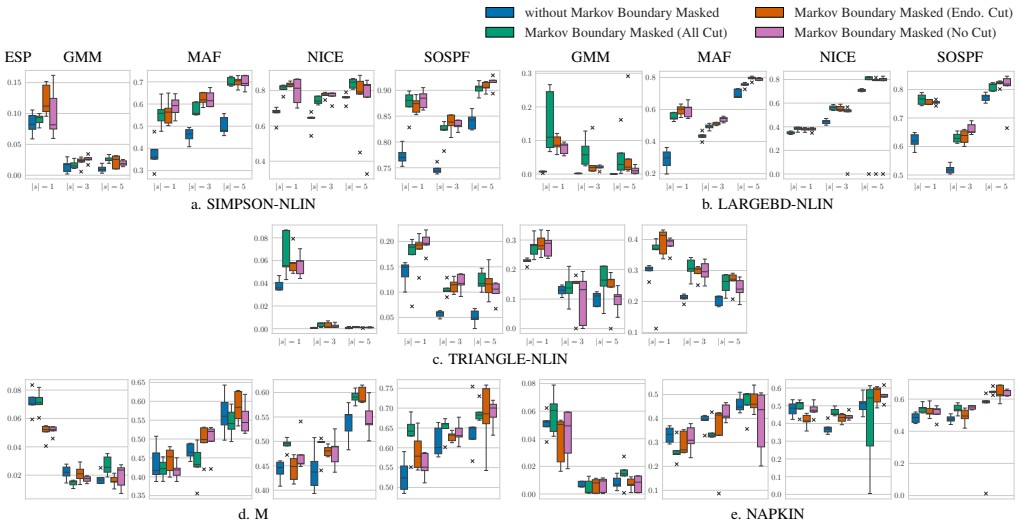

Figure 12: Ablation study for Markov boundaries on 4 different settings of SCMs: (a) SIMPSON-NLIN, (b) LARGEBD-NLIN, (c) M, (d) NAPKIN. Higher ESP indicates higher sampling efficiency.

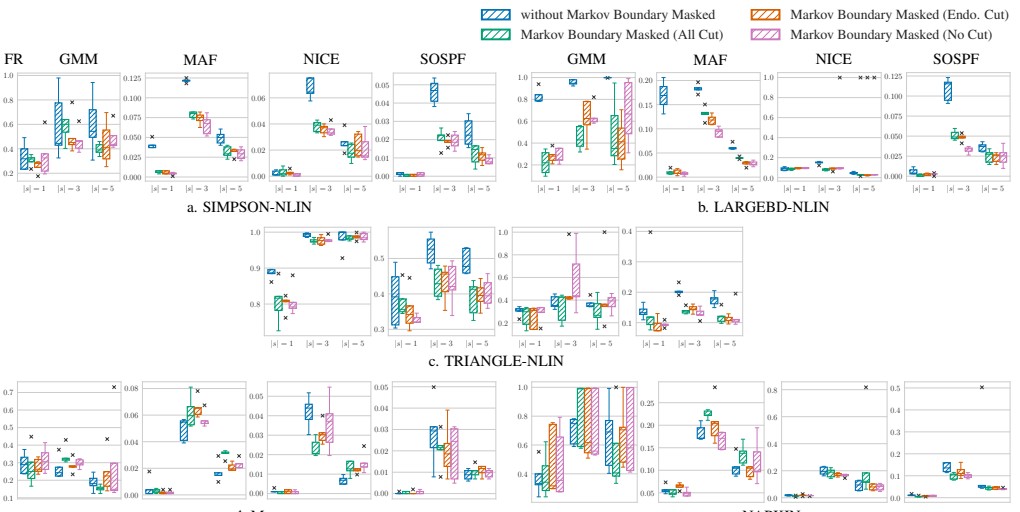

Figure 13: Ablation study for Markov boundaries on 4 different settings of SCMs: (a) SIMPSON-NLIN, (b) LARGEBD-NLIN, (c) M, (d) NAPKIN. Lower FR indicates broader coverage.

without a clear outperforming approach. However, we find that when the All Cut method performs weaker than that without Markov boundary masks, the latter two serve as compromises and indeed provide better results than All Cut. Given these observations, Endo. Cut was chosen in our previous experiments.

**Aggregation function** The next problem to address is the choice of $g$. We have posited the use of permutation-invariant functions in this paper because of the permutation invariance of sets. Empirical evidence in our experiments supports this assertion.

We compare 4 different choices of $g$: i) Concatenation: directly concatenate different encodings into a fixed-length vector, with zero-padding in empty positions. This is not a permutation-invariant function and is therefore included as a limiting case for comparison. ii) Summation: the simplest permutation-invariant method, which directly sums all encodings. iii) Weighted Summation: an extension of Summation that provides additional expressive power. Specifically, encodings are first mapped through a network $w$ to obtain element-wise weights (via softmax), which are then multiplied by the encodings themselves to yield the result. iv) Attention: offering stronger expressive

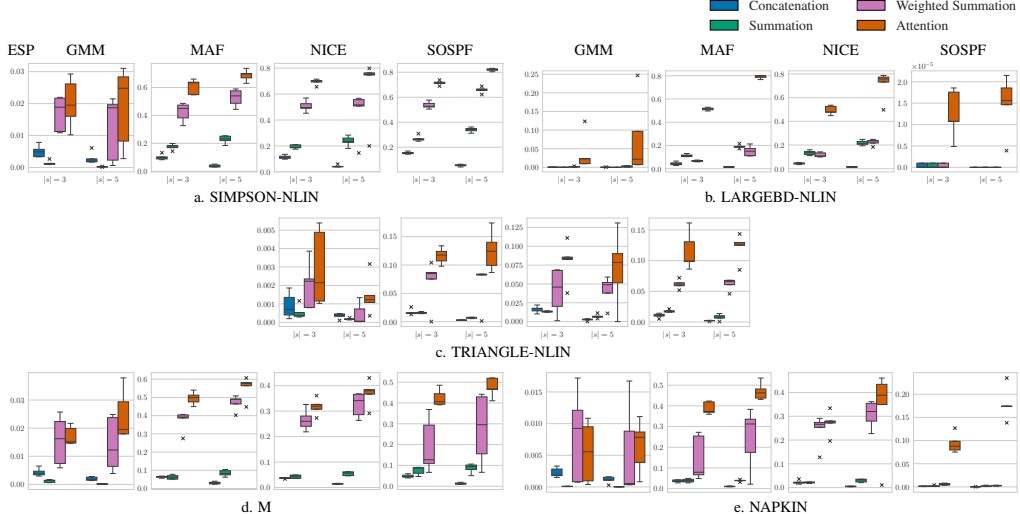

Figure 14: Ablation study for Markov boundaries on 4 different settings of SCMs: (a) SIMPSON-NLIN, (b) LARGEBD-NLIN, (c) M, (d) NAPKIN. Higher ESP indicates higher sampling efficiency.

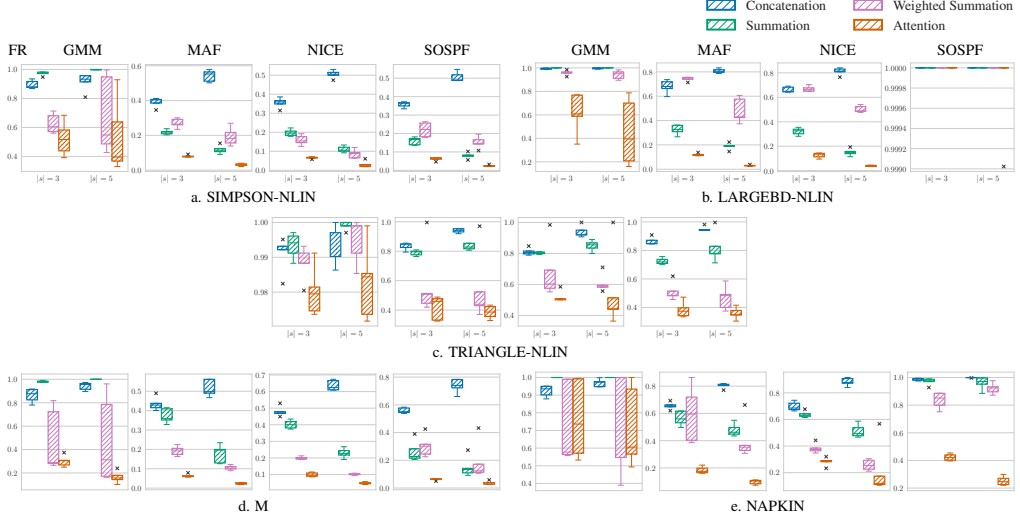

Figure 15: Ablation study for Markov boundaries on 4 different settings of SCMs: (a) SIMPSON-NLIN, (b) LARGEBD-NLIN, (c) M, (d) NAPKIN. Lower FR indicates broader coverage.

power, Attention differs from Weighted Summation in that it uses encodings before input $h$ rather than after output $h$, and a new network $a$ maps them to attention weights. Attention encoding is calculated through queries and keys $\mathrm{softmax}(h(x)a^\top(g(x)))$, where we set the values in the attention mechanism to be constantly 1.

We no longer explore aggregation methods with stronger expressive power, as we found in our experiments that they are likely to converge prematurely to local optima. This causes the variance in the state space to increase as the optimization process progresses, which in turn prevents Eq. 8 from being uniformly applied across the entire state space.

Fig. 14 and Fig. 15 respectively illustrate the impact of $g$ selection on ESP and FR under different SCM and density estimation models. Overall, permutation-invariant methods (Summation, Weighted Summation, Attention) consistently outperform permutation-variant methods (Concatenation), aligning with theoretical expectations and intuition. Among all methods, Attention consistently outperforms others, thus serving as the choice for all preceding experiments.

Table 4: Counterfactual density kernel ($\delta(\mathbf{y}_*)$) estimation using different sampling methods on CausalNF, extended for Tab. 2. Here, "O" represents the original SCM, and "P" represents the proxy SCM. Regularization are applied to the dimensions. The main metric used is FR ("-" indicates FR equals 1), with subscripts representing the regularized error bound.

| Method | SCM | SIMPSON-NLIN | | | TRIANGLE-NLIN | | LARGEBD-LIN |
|---|---|---|---|---|---|---|---|
| | | $\|s\|=1$ | $\|s\|=3$ | $\|s\|=5$ | $\|s\|=1$ | $\|s\|=3$ | $\|s\|=1$ |
| RS | O | $0.89_{0.014}$ | - | - | $0.84_{0.015}$ | $1.00_{0.000}$ | $0.99_{0.019}$ |
| | P | $0.90_{0.012}$ | - | - | $0.85_{0.013}$ | - | - |
| EXOM[MAF] | O | $0.00_{0.005}$ | $0.02_{0.015}$ | $0.01_{0.021}$ | $0.11_{0.007}$ | $0.12_{0.013}$ | $0.00_{0.007}$ |
| | P | $0.01_{0.005}$ | $0.40_{0.012}$ | $0.61_{0.016}$ | $0.11_{0.008}$ | $0.50_{0.014}$ | $0.04_{0.007}$ |
| EXOM[NICE] | O | $0.00_{0.006}$ | $0.02_{0.016}$ | $0.01_{0.046}$ | $0.08_{0.005}$ | $0.17_{0.016}$ | $0.01_{0.007}$ |
| | P | $0.01_{0.005}$ | $0.40_{0.013}$ | $0.61_{0.018}$ | $0.15_{0.007}$ | $0.55_{0.018}$ | $0.08_{0.007}$ |

## C.9 Counterfactual Estimation on Proxy SCMs

We combine identifiable proxy SCM with Exogenous Matching for counterfactual estimation. Specifically, we employ two distinct proxy SCM methods tailored for these scenarios to conduct counterfactual estimation on two types of SCMs.

**Counterfactual estimation with CausalNF** We conduct experiments on 3 different Markovian diffeomorphic SCMs. According to the previous arguments, CausalNF belongs to $\mathfrak{M}^{\mathrm{IM}(P\mathbf{v},\mathcal{O})}$, indicating that they are $\mathcal{L}_3$-identifiable. In our experiments, we mainly measure the error in counterfactual probability queries $P(\delta_l(\mathbf{y}_*))$, comparing the SCM represented by CausalNF to the original SCM in 1024 different queries. These results are reflected in Tab. 4, with averages presented. For the original SCM, we train EXOM five times and calculate the error bound for the same query. For CausalNF, we first train 5 proxy SCMs using observational data, then train EXOM 5 times for each model, calculate the error bound for the same queries, and finally take the average error bound across proxy SCMs.

Since in the continuous case it is challenging for RS to sample instances that satisfy the counterfactual event $\delta(\mathbf{y}_*)$, obtaining ground truth in experiments becomes difficult. Therefore, we primarily use FR as the main evaluation metric, while the error bound is employed to reflect the bias. When measuring error bounds, it is imperative to regularize the dimensionality of the counterfactual variables. Failure to do so may result in significant discrepancies in the final outcomes (estimates in high-dimensional scenarios tend to be considerably lower than those in low-dimensional ones, leading to a severe underestimation of errors in high-dimensional cases). Therefore, we calculate the error bounds as follows:

$$\bar{\epsilon}(\hat{P}) = 2 \cdot \mathbb{E}_{\mathcal{Y}_*} \left[ \mathrm{std}(\{\hat{P}_i^{\frac{1}{|\mathcal{Y}_*|}}(\mathcal{Y}_*) \mid i \in 1 \dots t\}) \right], \tag{122}$$

where $\hat{P}_i^{\frac{1}{|\mathcal{Y}_*|}}(\mathcal{Y}_*)$ represents the estimate of $P(\mathcal{Y}_*)$ in the $i$-th experiment, regularized by the exponent $\frac{1}{|\mathcal{Y}_*|}$, which denotes the expectation in a single dimension assuming independence across dimensions. The term $\mathrm{std}$ refers to the standard deviation. $t = 5$ is the number of trials.

The results in Tab. 4 indicate that EXOM exhibits a relatively small error, comparable to that of RS. Furthermore, when EXOM is applied to the proxy SCM, it yields an even smaller error (in part because all failure cases are excluded from consideration).

**Counterfactual estimation with NCM** We conducted experiments on 4 Regional canonical SCMs. According to their proof, NCM exhibits duality with respect to the identifiability of the counterfactual query $\mathcal{Q}$ and the graphical causal identifiability. In our experiments, we focus primarily on the unbiasedness of EXOM combined with NCM. The results are shown in Tabs 5 and 6.

As RS is straightforward to sample in the discrete case, the ground truth can be computed. We used RS with $10^6$ samples as ground truth (in the experiment, we compared against RS with $10^3$ samples) and then computed the average bias. Additionally, since these queries only involve a single dimension (regarding solely to $Y$), there is no need for dimensional regularization of the error bound.

In estimating these causal queries, EXOM yielded relatively small errors and showed even smaller errors when applied to proxy SCMs. This indicates that integration of EXOM with proxy SCMs

Table 5: Counterfactual effect estimation using different sampling methods on NCM, extended for Tab. 2. Here, "O" represents the original SCM, and "P" represents the proxy SCM. The main metric used is the average bias w.r.t. the ground truth, with the subscript denotes the 95% CI error bound over 5 trials

| Method | SCM | FAIRNESS | | | | FAIRNESS-XW | | | |
|---|---|---|---|---|---|---|---|---|---|
| | | ATE | ETT | NDE | CtfDE | ATE | ETT | NDE | CtfDE |
| RS | O | $0.01_{0.013}$ | $0.01_{0.018}$ | $0.01_{0.015}$ | $0.01_{0.020}$ | $0.01_{0.013}$ | $0.01_{0.027}$ | $0.01_{0.013}$ | $0.01_{0.023}$ |
| | P | $0.01_{0.013}$ | $0.01_{0.021}$ | $0.01_{0.014}$ | $0.01_{0.023}$ | $0.01_{0.012}$ | $0.02_{0.031}$ | $0.01_{0.013}$ | $0.02_{0.031}$ |
| EXOM[MAF] | O | $0.04_{0.095}$ | $0.06_{0.168}$ | $0.04_{0.131}$ | $0.06_{0.161}$ | $0.05_{0.140}$ | $0.07_{0.213}$ | $0.05_{0.149}$ | $0.07_{0.244}$ |
| | P | $0.01_{0.013}$ | $0.01_{0.024}$ | $0.01_{0.013}$ | $0.01_{0.044}$ | $0.00_{0.011}$ | $0.02_{0.028}$ | $0.00_{0.011}$ | $0.01_{0.031}$ |
| EXOM[NICE] | O | $0.01_{0.018}$ | $0.01_{0.030}$ | $0.01_{0.020}$ | $0.01_{0.039}$ | $0.01_{0.060}$ | $0.02_{0.036}$ | $0.02_{0.056}$ | $0.04_{0.069}$ |
| | P | $0.01_{0.017}$ | $0.01_{0.021}$ | $0.01_{0.012}$ | $0.01_{0.022}$ | $0.01_{0.012}$ | $0.02_{0.029}$ | $0.01_{0.014}$ | $0.01_{0.031}$ |

Table 6: (Tab. 5 continued). Counterfactual effect estimation using different sampling methods on NCM, extended for Tab. 2. Here, "O" represents the original SCM, and "P" represents the proxy SCM. The main metric used is the average bias w.r.t. the ground truth, with the subscript denotes the 95% CI error bound over 5 trials

| Method | SCM | FAIRNESS-XY | | | | FAIRNESS-YW | | | |
|---|---|---|---|---|---|---|---|---|---|
| | | ATE | ETT | NDE | CtfDE | ATE | ETT | NDE | CtfDE |
| RS | O | $0.01_{0.015}$ | $0.01_{0.022}$ | $0.01_{0.015}$ | $0.01_{0.023}$ | $0.00_{0.012}$ | $0.01_{0.025}$ | $0.01_{0.015}$ | $0.01_{0.027}$ |
| | P | $0.01_{0.012}$ | $0.01_{0.021}$ | $0.01_{0.014}$ | $0.01_{0.019}$ | $0.01_{0.022}$ | $0.02_{0.035}$ | $0.01_{0.017}$ | $0.01_{0.029}$ |
| EXOM[MAF] | O | $0.05_{0.189}$ | $0.09_{0.257}$ | $0.04_{0.100}$ | $0.13_{0.338}$ | $0.04_{0.106}$ | $0.10_{0.363}$ | $0.05_{0.181}$ | $0.13_{0.396}$ |
| | P | $0.00_{0.009}$ | $0.01_{0.020}$ | $0.00_{0.013}$ | $0.01_{0.029}$ | $0.01_{0.026}$ | $0.01_{0.028}$ | $0.01_{0.024}$ | $0.01_{0.031}$ |
| EXOM[NICE] | O | $0.01_{0.019}$ | $0.03_{0.111}$ | $0.05_{0.066}$ | $0.05_{0.144}$ | $0.03_{0.129}$ | $0.05_{0.124}$ | $0.06_{0.150}$ | $0.11_{0.344}$ |
| | P | $0.01_{0.013}$ | $0.01_{0.022}$ | $0.00_{0.011}$ | $0.01_{0.054}$ | $0.01_{0.022}$ | $0.01_{0.033}$ | $0.01_{0.018}$ | $0.01_{0.031}$ |

is practical to address real-world problems. In particular, the errors in ETT and CtfDE within the original SCM were substantially higher than those in RS, primarily because their expressions include the denominator $P(X)$, and inaccuracies in $P(X)$ markedly influence their results. This matter will be summarized in App. D.3.

## D    Discussions

### D.1    Broader Impacts

Our work aims to enhance the efficiency of counterfactual reasoning and make it applicable to theoretically broader contexts. On the positive side, current counterfactual reasoning has a positive impact on society in terms of counterfactual fairness and counterfactual decision making. However, on the flip side, if assumptions and premises are not applicable to real-world situations, counterfactual reasoning can still lead to potential biases, inequalities, and misuse. Therefore, careful consideration of the correctness of the assumptions and the appropriateness of usage scenarios is required before applying counterfactual reasoning.

### D.2    Counterfactual Density

For continuous distributions, it is the density function rather than the probability measure that garners widespread attention and study within the relevant field. According to the definition of probability density, we can find its connection with the Lebesgue differentiation theorem, which directly provides a method for density estimation:

**Theorem 4** (Counterfactual Density Estimation). Assuming the counterfactual variables $\mathbf{Y}_*$ are absolutely continuous on $\langle \Omega_{\mathbf{Y}_*}, \Sigma_{\mathbf{Y}_*}, P_{\mathbf{Y}_*} \rangle$, with a probability density existing at $\mathbf{y}_* \in \Omega_{\mathbf{Y}_*}$. Let $\delta(\mathbf{y}_*)$ be a function that maps $\mathbf{y}_*$ to a cube (or ball) centered at $\mathbf{y}_*$, and $\delta(\mathbf{y}_*) \to \mathbf{y}_*$ denotes the cube's side length (or the ball's diameter) tending to 0, then the following equation holds:

$$p(\mathbf{y}_*) = \lim_{\delta(\mathbf{y}_*) \to \mathbf{y}_*} \frac{P(\mathbf{Y}_* \in \delta(\mathbf{y}_*))}{|\delta(\mathbf{y}_*)|} \approx \frac{P(\mathbf{Y}_* \in \delta(\mathbf{y}_*))}{|\delta(\mathbf{y}_*)|} \tag{123}$$

where $|\delta(\mathbf{y}_*)|$ is the volume (i.e., the Lebesgue measure) of $\delta(\mathbf{y}_*)$.

*Proof.* Directly inferred from the definition of the probability density function (Radon-Nikodym derivative) and the Lebesgue differentiation theorem. □

This method is analogous to kernel density estimation (with the Parzen window as a cube). However, the precision of the estimates provided by this method is highly dependent on $|\delta(\mathbf{y}_*)|$, further highlighting the difficulty of satisfying the indicator function in the counterfactual probability measure. We indirectly investigated the performance of EXOM on this issue in our related experiments on continuous counterfactual estimation, where we chose $\mathbf{Y}_* \in \delta_l(\mathbf{y}_*)$ as the counterfactual event for the estimation. The results of the related experiments confirmed the potential of EXOM in addressing this problem.

Of course, kernel-based density estimation has many limitations. We hope that Thm. 4 will serve as a lemma for future work, leading to the derivation of more suitable density estimation methods for continuous counterfactual distributions.

### D.3 Limitations

1. **Dependency on counterfactual generation**. In practical tasks, we typically only have access to observational or experimental data. In some settings, we also assume the acquisition of causal graphs or causal orderings through causal discovery and prior knowledge. However, the proposed method is not directly applicable to such real-world settings, as indicated by the assumption in Sec. 3, which requires the ability for counterfactual generation. In practical scenarios, one approach is to pre-train a neural proxy SCM to perform the counterfactual generation task.
2. **Boundedness condition on importance weights**. In the context of learning and optimization, we introduce additional designs: the bounded importance weight condition. We assumed conditional distribution model to satisfy the boundedness of importance weight as much as possible. However, in practical applications, this depends on the specific forms of the exogenous distribution and the conditional distribution model, necessitating case-by-case analysis.
3. **Faithfulness**. For injection of the Markov boundary, we also introduce the faithfulness assumption, which assists us in proving theorems and providing fast algorithms. However, this assumption might be violated, leading to more variables being included in the actual Markov blanket than our calculations predict. Additionally, modifications to autoregressive and coupling models do not fully align with the ideal effect. In fact, due to potential interactions within the hidden encodings of autoregressive and coupling models, even if each layer adheres to the Markov boundary dependencies, the aggregation of multiple layers may violate these dependencies.
4. **Recursiveness**. Another potential assumption in this work is the recursive SCM, which might be violated in more general settings, such as systems with cycles. All the theoretical foundations of this work are based on recursive SCMs, thus requiring these theories to be extended to more general cases.
5. **Variance of $\mathcal{L}_3$ expressions**. In estimating counterfactual expressions $\mathcal{Q} \in \mathcal{L}_3$, although the final result is unbiased, computing each term $P(\mathcal{Y}_*)$ individually introduces significant variance, as demonstrated in the experiments on counterfactual query estimation with real SCMs (see Tabs 5 and 6). Therefore, future work should focus on reducing the estimation variance of specific queries.
6. **Potential sensitivity to the specific forms of SCM in practice**. With respect to the parameter form of the SCM, although we impose no theoretical constraints on the form of SCM, experiments indicate that in some settings the form of SCM still significantly influences outcomes. For example, combinatorial experiments and ablation studies reveal unexpected scenarios, particularly with the combinations of long-chain SCMs (CHAIN-LIN-4, CHAIN-LIN-5, LARGEBD-NLIN). As the causal chain lengthens, the performance of these density estimation models gradually weakens, suggesting that long causal chains complicate counterfactual inference. This phenomenon indicates that the form of SCM in practice affects the specific effectiveness of our methods, highlighting the necessity of selecting appropriate density estimation models and making corresponding adjustments.

