# OpenReview forum: "Exogenous Matching: Learning Good Proposals for Tractable Counterfactual Estimation"
_NeurIPS.cc/2024/Conference — NeurIPS 2024 poster_

### Official Review · Reviewer_rY6N · 2024-07-10

**Soundness:** 3
**Presentation:** 2
**Contribution:** 3
**Rating:** 6
**Confidence:** 4

**Summary:**

The manuscript introduces exogenous matching, an importance sampling method for efficient estimation of counterfactual expressions in various settings. This method transforms the variance minimization problem into a conditional distribution learning problem, allowing integration with existing modeling approaches. The authors validate their theoretical findings through experiments with different Structural Causal Models (SCMs), showing competetive performance in a range of counterfactual estimation tasks. They also examine the impact of structural prior knowledge and demonstrate the method's unbiased estimates and practical applicability in identifiable proxy SCMs.

Update: revising score upward following author rebuttal.

**Strengths:**

The topic is timely and important, as counterfactual estimation has become an increasingly popular subject in statistics and machine learning. The proposal builds on recent results in neural causal models, specifically with normalizing flows. The ability to incorporate prior knowledge in the form of Markov boundaries is especially welcome, since computing counterfactuals is often intractable without such constraints. The theoretical results appear sound (though I confess I did not go closely through the proofs) and the empirical results are compelling.

**Weaknesses:**

The manuscript is not always clear, probably because a great deal of material has been moved to the appendix to accommodate page count. The result is a somewhat disjointed text that would likely be better served by a journal publication than a conference paper. That said, I am generally supportive of this submission and would be willing to revise my score upward if my questions are adequately addressed (see below).

**Questions:**

When we say the causal model is “not fully specified”, does that just refer to the structural equations or to the graphical structure as well? In general, I was not always certain just how much causal information is used as input to this method.

I’m a bit confused by Eq. 5. If this is meant to be a variance estimator, then presumably the RHS should be something like $\mathbb{E}[X^2] – (\mathbb{E}[X])^2$, where $X$ denotes the likelihood ratio $p(u) / q(u)$, correct? This is almost but not quite what we find here. Looking at the appendix, I don’t see why Eq. 48 follows from Eq. 47. Why do we get to drop the square from the first term?

What does the constant $c$ denote in Eq. 6? Is it just the entropy of $P$? A brief word on this would help with intuition.

Should there definitely be a negative both in front of the expectation *and* within it? I assume the first summand should just be $-\mathbb{E} [ \log Q(u \mid Y_* (u)) ] $? This would look more like the classic cross entropy formula, as indicated in the following line. Eq. 7 also suggests this.

On “augmented graphs” – does this “reverse projection” of the ADMG always work? Different DAGs can have the same ADMG, for instance if two latent variables have all the same endogenous children. Perhaps there’s an unstated minimality assumption at work here?

What is $m$ in Eq. 13?

It is not clear to me from Sect. 5 what the sample size and data dimensionality are for these tasks? In general, this section appears rushed. The performance metrics are also somewhat surprising. If data is simulated, then we presumably have ground truth with respect to counterfactual probabilities. If so, then why not just compute the mean square error of the proposed estimator, perhaps as a function of sample size?

**Limitations:**

Limitations are adequately addressed.

---

> ### Author Rebuttal · Authors · 2024-08-06
>
> We thank the reviewer for the patience and valuable feedback. We acknowledge that, due to page limitations, we had to move some material to the appendix, which may have affected the clarity of the manuscript. We understand that this may have made the paper appear somewhat disorganized. To address this issue, we will reorganize the content to ensure a clearer and more coherent presentation. We will introduce more transitional text to improve the flow of the manuscript and make the key points more understandable.
>
> Below are our responses to the questions:
>
> > **Question 1**: When we say the causal model is “not fully specified”, does that just refer to the structural equations or to the graphical structure as well? How much causal information is used as input to this method?
>
> **Response**: “Fully specified” means that $\mathcal{F}$ is completely known, which means that we can write out the explicit expression of any $f \in \mathcal{F}$ (i.e., the structural equation). This also implies that the graphical structure is known, as the graph structure can be derived from $\mathcal{F}$. On the other hand, “not fully specified but $\mathcal{F}$ is evaluable” means that given an input $\mathbf{u} \in \Omega_\mathbf{U}$, we can obtain the output $\mathcal{F}(\mathbf{u})$. In other words, $\mathcal{F}$ is allowed to be regarded as a black box, such as a neural network.
>
> > **Question 2**: Confusion by Eq. 5 and inadequate derivations in the appendix.
>
> **Response**: We apologize for the confusion caused by the omission of the relevant steps in the appendix. Specifically, the subscript of the expectation operator has changed:
>
> $\mathbb{E}_Q\left[\left(\frac{P(\mathbf{u})}{Q(\mathbf{u})}\right)^2 f^2(\mathbf{u})\right]=\sum\_{\mathbf{u}\in\Omega\_{\mathbf{U}}} Q(\mathbf{u}) \left(\frac{P(\mathbf{u})}{Q(\mathbf{u})}\right)^2 f^2(\mathbf{u})$ $= \sum\_{\mathbf{u} \in \Omega\_{\mathbf{U}}} \frac{P^2(\mathbf{u})}{Q(\mathbf{u})} f^2(\mathbf{u})$ $= \sum\_{\mathbf{u} \in \Omega\_{\mathbf{U}}} P^2(\mathbf{u}) \left(\frac{f^2(\mathbf{u})}{Q(\mathbf{u})}\right)$ $= \sum\_{\mathbf{u} \in \Omega\_{\mathbf{U}}} P(\mathbf{u}) \left(\frac{P(\mathbf{u})}{Q(\mathbf{u})}\right) f^2(\mathbf{u})$ $= \mathbb{E}_P\left[\frac{P(\mathbf{u})}{Q(\mathbf{u})} f^2(\mathbf{u})\right].$
>
> Similar steps and the same conclusion also apply to continuous distributions.
>
> > **Question 3**: What does the constant $c$ denote in Eq. 6? Is it just the entropy of $P$? A brief word on this would help with intuition.
>
> As described in the appendix (line 1018), $c$ is composed of the entropy of $P$ and $\log\kappa$. We will include an brief explanation of this in the main text.
>
> > **Question 4**: Typo error of $-\mathbb{E}\left[\log q(\mathbf{u}\mid\mathbf{Y}_*(\mathbf{u}))\right]$.
>
> We thank the reviewer for the careful review and correction. This was indeed a typo, and we will fixed it in the updated version.
>
> > **Question 5**: On “augmented graphs” – does this “reverse projection” of the ADMG always work? Perhaps there’s an unstated minimality assumption at work here?
>
> In recursive SCMs, there is not always a "reverse projection" between ADMGs and their corresponding augmented DAGs, as the reviewer mentioned, different DAGs may correspond to the same ADMG. We have tried to understand why the reviewer might have this question. We believe it may be due to some ambiguity in our presentation in the "Markov boundary" section, which might have led readers to think that "ADMG -> DAG -> MB" is implied. However, we only emphasize "DAG -> MB". We thank the reviewer for this question and will clarify this information.
>
> > **Question 6**: What is $m$ in Eq. 13?
>
> In Appendix C.1, we explained $m$, but we forgot to explain it in the main text. We will add the missing explanation of \$m$ in the main text, and the correct notation is $\le m$ instead of $\ge m$. Here, $m$ is a threshold value; if the proportion of valid samples in a test is less than $m$, it is considered a failure.
>
> > **Question 7**: It is not clear to me from Sect. 5 what the sample size and data dimensionality are for these tasks? In general, this section appears rushed. The performance metrics are also somewhat surprising. If data is simulated, then we presumably have ground truth with respect to counterfactual probabilities. If so, then why not just compute the mean square error of the proposed estimator, perhaps as a function of sample size?
>
> The experimental settings in Section 5 are detailed in App. C.4. For sample sizes, RS was set to $10^6$, while the other methods used $10^3$. For variable dimensions, this is related to $|s|$ and the SCM settings; for example, LARGEBD-NLIN with 9 variables results in 45 dimensions when $|s|=5$. As for the choice of metrics, our use of ESP and LL is motivated by the intuition that indicator functions in counterfactual probability estimators are more challenging to satisfy (that is, equals $1$). Therefore, we refer to metrics from rare event sampling, which are computationally convenient, dimension-independent, and naturally range from 0 to 1. Another reason for using this metric is that as dimensions increase, counterfactual probabilities decrease exponentially, making it nearly impossible for RS to sample effective samples in high-dimensional cases. RS is a tractable method for estimating ground truth in general cases, so the ground truths are almost unattainable in such situations.\
> Considering the reviewer's feedback, we note that metrics based on the deviation from ground truth are suitable for counterfactual effect estimation, given the low dimensionality in the experimental setting. Therefore, we have updated the section "Counterfactual Estimation on Proxy SCMs" to use the bias with respect to the true values ($|\hat{\mathbb{E}} - \mathbb{E}|$) as a metric of unbiasedness. The updated results can be found in the Author Rebuttal PDF (Tab. 1).

---

> > ### Comment · Reviewer_rY6N · 2024-08-12
> > **Re: Author rebuttal**
> >
> > Many thanks to the authors for their detailed replies to my comments. I will be revising my score upward in light of these clarifications.

---

> > > ### Author Response · Authors · 2024-08-13
> > >
> > > We thank the reviewer for reading our rebuttal and updating the score accordingly. If there are any further questions, we would be happy to provide additional information.

---

### Official Review · Reviewer_6eu3 · 2024-07-12

**Soundness:** 3
**Presentation:** 2
**Contribution:** 3
**Rating:** 5
**Confidence:** 3

**Summary:**

This paper introduces an importance sampling method for efficient estimation of counterfactual expressions within general settings. It transforms the variance minimization problem into a conditional distribution learning issue, allowing integration with existing modeling approaches. The paper also explores the impact of incorporating structural prior knowledge, i.e. Markov boundaries, and applies the method to identifiable proxy SCMs, proving the unbiasedness of estimates and illustrating the method's practical applicability.

**Strengths:**

1. The paper is well-structured, with many subsections and bullet points summarizing paragraphs.
2. The approach proposed in this paper has clear intuition and is easy to implement.

**Weaknesses:**

1. Contributions are not disentangled well. All three points involve experimental or empirical findings.
2. Some results of the ablation study are abnormal. First, the results show that the approach proposed is not robust. Under setting SIMPSON-NLIN and M, the inclusion of Markov Boundary Mask significantly improves ESP. However, under setting LARGEBD-NLIN and NAPKIN, the Markov Boundary Mask harms the performance, especially with backbone SOSPF. Second, the ESPs under setting LARGEBD-NLIN with backbone SOSPF are almost 0, even when $|s|=1$, which is hard to explain if including Markov Boundary Mask is effective. Third, the variance under setting LARGEBD-NLIN with backbone NICE is extremely large.
3. Insufficient explanation or legend for figures, making it difficult for readers to understand. For example, in Figure 1, $\mathcal{M}$ with a subscripted hammer is not explained. In Figure 3, the legend does not indicate what different colors mean.
4. Hard to follow. Some terminologies need explanation or reference. For example, in line 234, *faithfulness* is not defined.

**Questions:**

I would like to know whether Theorem 2 and 3 are novel. Have the papers cited (like 81, 3, 94, 111)  and other papers proposed methods to obtain Markov boundaries? If so, what is the improvement of the method proposed in this paper?

**Limitations:**

The authors adequately discussed the limitations in the paper.

---

> ### Author Rebuttal · Authors · 2024-08-06
>
> We thank the reviewer for highlighting the issues and providing valuable feedback to help us improve the manuscript. Below are our responses to the identified weaknesses and questions:
>
> > **Weakness 1**: Contributions are not disentangled well. All three points involve experimental or empirical findings.
>
> **Response**: We thank the reviewer for the suggestion. We will reorganize the presentation of our contributions by placing the experimental and empirical conclusions in the third point, while the first two points will separately address the two main contributions of our work: i) the introduction of EXOM and its theoretical guarantees, and ii) the injection of Markov boundaries into neural networks for conditioning.
>
> > **Weakness 2**: Some results of the ablation study are abnormal. First, the results show that the approach proposed is not robust. Second, it is hard to explain if including Markov Boundary Mask is effective under LARGEBD-NLIN with backbone SOSPF.Third, the variance under setting LARGEBD-NLIN with backbone NICE is extremely large.
>
> **Response**: We have found that the abnormality observed is not due to the lack of robustness in the method but rather is caused by the width of the hidden layers in the neural network. Specifically, the injection of Markov boundaries relies on masking connections in the neural network, which means that with the same width, the injection of Markov boundaries results in fewer weights being used during inference. In experiments where the performance was normal, the injection of Markov boundaries still exhibited superiority even with fewer weights. Moreover, as the Markov boundaries become smaller (as seen with NAPKIN and LARGEBD-NLIN), the masks become sparser, resulting in even fewer weights being involved in inference. When the number of hyperparameters to be inferred for the conditional model is very high (e.g., SOSPF), the neural network's representational capacity is significantly insufficient, leading to markedly poor performance.
>
> In the previous ablation studies, to ensure fairness, we used a hidden layer width of 64 across all settings, but this was inadequate for SOSPF. We have now increased the width to 256, and the result has reversed, with the experimental results on SOSPF showing consistency with results from other settings, as detailed in the Author Rebuttal PDF (Fig. 1).
>
> In the follow-up, we will also update the relevant results in the appendix. We will include results in App. C.7 and App. C.8 with a hidden layer width of 256, then add a discussion about the recommended hyperparameter settings and their theoretical rationale in App. C.4.
>
> > **Weakness 3**: Insufficient explanation or legend for figures, making it difficult for readers to understand. For example, in Figure 1, $\mathcal{M}$ with a subscripted hammer is not explained. In Figure 3, the legend does not indicate what different colors mean.
>
> **Response**: In Fig. 1, $\mathcal{M}$ with a subscripted hammer represents the SCM under intervention, i.e., the submodel. In Fig. 3, the different colors indicate different submodels. We will address fidelity after d-separation, where fidelity refers to the independence implied in the probability distribution corresponding to d-separation in the DAG.
>
> > **Weakness 4**: Hard to follow. Some terminologies need explanation or reference. For example, in line 234, faithfulness is not defined.
>
> **Response**: We apologize for the confusion caused by the omission of the conceptual explanation in the main text. We have explained the concept of faithfulness in the appendix (line 1115, 1116). We will revise the manuscript to include the discussion of faithfulness in the main text, following the section on d-separation. This is because faithfulness and d-separation are highly related; it signifies that every conditional independence present in the distribution is entailed by the d-separation in the DAG.
>
> Regarding the issue of lacking explanations for legends and terminology, we will thoroughly review the entire manuscript to identify any figures, concepts, or terms that may be missing explanations. We will ensure that these are properly addressed and clarified in the updated version.
>
> > **Question 1**: Whether Theorem 2 and 3 are novel? Have the papers cited (like 81, 3, 94, 111) and other papers proposed methods to obtain Markov boundaries? If so, what is the improvement of the method proposed in this paper?
>
> **Response**: In fact, the cited works and the proposed method are used for completely different tasks. The cited works are used to learn Markov boundaries among observable variables from data, while the proposed method is specifically designed to derive the counterfactual Markov boundaries (Def. 2) of exogenous variables from an augmented graph. The cited works cannot be directly applied to learn the counterfactual Markov boundary unless the exogenous variables are also observed in the data.

---

> > ### Comment · Reviewer_6eu3 · 2024-08-13
> >
> > Thanks for the rebuttal by the authors addresses my concerns. I will update my rating accordingly.

---

> > > ### Author Response · Authors · 2024-08-14
> > >
> > > We are glad that the rebuttal addressed the reviewer's concerns, and we appreciate the reviewer for updating the rating accordingly.

---

### Official Review · Reviewer_aexG · 2024-07-13

**Soundness:** 2
**Presentation:** 2
**Contribution:** 2
**Rating:** 5
**Confidence:** 3

**Summary:**

This paper presents Exogenous Matching (EXOM), a new importance sampling method for estimating counterfactual probabilities in Structural Causal Models (SCMs). EXOM transforms variance minimization into a conditional distribution learning problem, providing an upper bound on counterfactual estimator variance as per Theorem 1. It outperforms existing methods across various SCM settings and integrates well with identifiable neural proxy SCMs for practical applications. By incorporating prior knowledge through Markov boundaries, EXOM further enhances performance, demonstrating its potential as an efficient tool for counterfactual estimation in diverse scenarios.

**Strengths:**

1. EXOM provides a tractable and efficient approach for counterfactual estimation in general settings, including scenarios with discrete or continuous exogenous variables and various observations and interventions. This flexibility makes it applicable to a wide range of causal inference problems.

2. The method is built on solid theoretical grounds, with the authors deriving an optimizable variance upper bound for counterfactual estimators.

3. The authors incorporate structural prior knowledge, specifically Markov boundaries, into the neural networks used for parameter optimization. They empirically validate the effectiveness of this approach across various scenarios.

4. EXOM consistently outperforms other importance sampling methods in various SCM settings, as demonstrated by the experimental results. Its compatibility with identifiable neural proxy SCMs further enhances its practical applicability.

**Weaknesses:**

1. Theorem 1 relies on the assumption that the density ratio $q(\mathbf{u}|\mathbf{y}_ {\ast})/q(\mathbf{u}|\mathbf{y}_ {\ast}^\prime)\leq \kappa$ holds for all $\mathbf{u} \in \Omega_{\mathbf{u}}$ and $y_{\ast}$, $y_ {\ast} ^\prime \in \Omega_{\mathbf{Y}_{\ast}}$. This assumption may be overly stringent, as probability measures with infinite support sets might easily violate it. Could the authors elaborate on this assumption and provide examples of distributions that satisfy it?

2. In the Sampling and Optimization section, the distribution of the exogenous variable $\mathbf{U}$ is assumed to be known. However, in practical scenarios, $\mathbf{U}$ is often unknown, necessitating additional efforts to estimate $\mathbf{P_U}$ [A]​. Could the authors provide further clarification on this assumption and discuss potential methods for estimating $\mathbf{P_U}$​?

3. I understand the authors only consider models that provide identifiability results. However, it is encouraged to include neural proxy SCM methods based on VAE and DDPM as experimental baselines. While these may lack identifiability guarantees, comparing against them would further illustrate the superiority of the proposed method in relation to current state-of-the-art techniques.

4. While the method shows good performance on the tested SCMs, it's unclear how well it scales to larger, more complex causal models. The experiments are conducted on relatively small SCMs, and scalability to high-dimensional or densely connected causal graphs isn't thoroughly addressed.


[A] Ren, Shaogang, and Xiaoning Qian. "Causal Bayesian Optimization via Exogenous Distribution Learning." *arXiv preprint arXiv:2402.02277* (2024).

**Questions:**

1. In Table 1, the EXOM method with MAP shows significantly better performance on the SIMPSON-NLIN and NAPKIN datasets compared to EXOM with GMM, whereas the performance on the FAIRNESS-XW dataset is similar for both methods. Could the authors provide further explanation for this discrepancy? Why does the GMM approach underperform in these specific cases, and what factors contribute to the similar performance on FAIRNESS-XW?

2. In the ablation study investigating the impact of injecting Markov boundaries, could the authors please include the performance results of EXOM without Markov boundaries? This comparison would further illustrate the benefit of incorporating Markov boundaries.

**Limitations:**

The authors discuss the limitations of their work in Section 6 and Section D.3. Notably, this work does not have a negative societal impact.

---

> ### Author Rebuttal · Authors · 2024-08-06
>
> We thank the reviewer's patience in reading our manuscript and the attention to issues concerning assumptions, generalization, scalability, and experimental aspects. Below are our responses to these concerns:
>
> > **Weakness 1**: Could the authors elaborate on the assumption about density ratio in Theorem 1 and provide examples of distributions that satisfy it?
>
> **Response**: We agree with this concern; this is indeed not a weak assumption, especially for probability measures with infinite support sets. This is because the proof of Thm. 1 relies on the linearity of Lebesgue integration, which requires Lebesgue integrable functions to be bounded. Some alternative formulations with more relaxed assumptions are as follows:
>
> 1. **By Sufficient Condition**: If the importance weights satisfy $\kappa^{-\frac{1}{2}} \le p(\mathbf{u}) / q(\mathbf{u} \mid \mathbf{y}_\*) \le \kappa^\frac{1}{2}$ with probability 1, then, almost surely, $q(\mathbf{u} \mid \mathbf{y}'\_\*) / q(\mathbf{u} \mid \mathbf{y}\_\*) \le \kappa$, and hence Thm. 1 holds.
>
> 2. **By Concentration Inequality**: If the importance weights satisfy $\kappa^{-\frac{1}{2}} \le p(\mathbf{u}) / q(\mathbf{u} \mid \mathbf{y}\_\*) \le \kappa^\frac{1}{2}$ with probability $\zeta < 1$, then by introducing the assumption that the second moment of the weights $\mathbb{E}\_w^2$ is bounded, and using Thm. 1 from [1], we can derive an inequality of the form $\mathbb{V} \le \widehat{\mathbb{V}} + f(n, \zeta, \kappa, \delta, \mathbb{E}\_w^2)$ with probability $(1-\delta) \cdot \zeta^n$, where $\widehat{\mathbb{V}}$ is computed from $n$ i.i.d. samples $\mathbf{u}^{(1)}, \dots, \mathbf{u}^{(n)}$ with bounded weights. In this case, for $\widehat{\mathbb{V}}$, almost surely $q(\mathbf{u}^{(i)} \mid \mathbf{y}'\_\*) / q(\mathbf{u}^{(i)} \mid \mathbf{y}\_\*) \le \kappa$, so Thm. 1 holds for $\widehat{\mathbb{V}}$, and consequently, it also holds for $\mathbb{V}$ by the inequality relationship.
>
> 3. **By Approximation**: Similar to [2], consider only the approximation in bounded regions. If the importance weights satisfy $\kappa^{-\frac{1}{2}} \le p(\mathbf{u}) / q(\mathbf{u} \mid \mathbf{y}\_\*) \le \kappa^\frac{1}{2}$ in $\Omega\_{\mathbf{U}}^+$, let $\mathcal{Y}^\*\_+ = \\{\mathbf{Y}^*(\mathbf{u}) \mid \mathbf{u} \in \Omega\_{\mathbf{U}}^+ \\} \cap \mathcal{Y}\_\*$, and if $P(\mathcal{Y}\_\*) \approx P(\mathcal{Y}^\*\_+)$, then the theorem holds for $\mathcal{Y}^\*\_+$ and approximately for $\mathcal{Y}\_\*$.
>
> > **Weakness 2**: Could the authors provide further clarification on the assumption that $P_\mathbf{U}$ is known and discuss potential methods for estimating $\mathbf{U}$?
>
> **Response**: We agree with the viewpoint that obtaining the exogenous distribution $P^*_\mathbf{U}$ of the true SCM $\mathcal{M}^*$ is challenging. Literature [A] addresses this by introducing additional parameter assumptions for the causal mechanism $\mathcal{F}$ to recover the exogenous distribution. However, for counterfactual tasks, recovering the true exogenous distribution may not be necessary, since identifiable counterfactual results can still be provided by neural proxy SCMs. As deep generative models, the exogenous (or, latent) distribution $P_\mathbf{U}$ is typically either pre-specified or trainable, so the assumption that $P_\mathbf{U}$ is known is weak in this context.
>
> > **Weakness 3**: It is encouraged to include neural proxy SCM methods based on VAE and DDPM to further illustrate the superiority.
>
> **Response**: We thank the reviewer for the suggestion. Although in our experiments we only selected models that provide identifiability results as baselines, theoretically, the proposed method is not limited to the type of SCMs. We will consider applying it to neural proxy SCMs with VAE or DDPM as the backbone in future work for larger, more complex tasks.
>
> > **Weakness 4**: While the method shows good performance on the tested SCMs, it's unclear how well it scales to larger, more complex causal models.
>
> **Response**: In principle, we do not impose any restrictions on the size, complexity, or dimensionality of variables for SCM. However, in practice, it is well-known that computational complexity is a challenge when using normalizing flows for high-dimensional data. Therefore, scalability is an aspect worth investigating in future research.
>
> > **Question 1**: Could the authors provide further explanation for the discrepancy of performance on the SIMPSON-NLIN and NAPKIN datasets? Why does the GMM approach underperform in these specific cases, and what factors contribute to the similar performance on FAIRNESS-XW?
>
> **Response**: The FAIRNESS dataset is discrete and finite, whereas the SIMPSON-NLIN dataset is continuous and diffeomorphic. Thus, the sampling space of the former is smaller, which is why it did not show a significant difference in performance. The performance of GMM is as expected since its expressive capacity is weaker than that of MAF. We will include a breif discussion of these results in the main text to facilitate reader understanding.
>
> > **Question 2**: Could the authors please include the performance results of EXOM without Markov boundaries in the ablation study?
>
> **Response**: We have included results for "EXOM without Markov boundaries" (i.e., the blue bars) in both Fig. 4 and App. C.8 (Fig. 9), and compared them with "EXOM with Markov boundaries" (i.e., the orange bars). These results demonstrate the advantages of incorporating Markov boundaries. We will further improve the display of the legends in the figures to enhance readability.
>
> ----
> [1] [Cortes, C., Mansour, Y., & Mohri, M. (2010). Learning Bounds for Importance Weighting.](https://papers.nips.cc/paper_files/paper/2010/hash/59c33016884a62116be975a9bb8257e3-Abstract.html)\
> [2] [Tengchao Yu, Linjun Lu, & Jinglai Li. (2019). A weight-bounded importance sampling method for variance reduction.](https://arxiv.org/abs/1811.09436)

---

> > ### Comment · Reviewer_aexG · 2024-08-13
> >
> > Thank you to the authors for the detailed responses. Regarding Weakness 1, in addition to the relaxed assumptions, could the authors kindly provide some concrete examples of distributions with infinite support sets that satisfy these assumptions?

---

> > > ### Author Response · Authors · 2024-08-13
> > >
> > > Thank you for the reviewer’s attention to this weakness. In fact, without concentration inequalities or approximations,  it is challenging to find concrete examples that fully satisfy this assumption on infinite support sets. Even the density ratios between Gaussian distributions often violate boundedness asymptotically, as illustrated in [1] (Eg. 4.1). The best example we could find is from [3] (Eq. 11), which provides upper and lower bounds between the binomial distribution and the Poisson distribution (with infinite support) under certain conditions.
> > >
> > > Of course, once concentration inequalities or approximations are employed (as in relaxed formulations 2 and 3), the weakness regarding infinite support becomes less acute, provided we prove or assume that the probability measure of $P_\mathbf{U}$ on the unbounded portion is zero or very close to zero.
> > >
> > > ----
> > > [3] [Dümbgen, L., Samworth, R., & Wellner, J. (2021). Bounding distributional errors via density ratios.](https://arxiv.org/abs/1905.03009)

---

### Official Review · Reviewer_8MDB · 2024-07-17

**Soundness:** 4
**Presentation:** 4
**Contribution:** 4
**Rating:** 7
**Confidence:** 3

**Summary:**

Based on the importance sampling methods, the authors propose an exogenous matching approach to estimate counterfactual probability in general settings. They derive the variance upper bound of counterfactual estimators and transform it into the conditional learning problem. They also employ the Markov boundaries information in the inference to improve the learning performances further. Extensive experiments validate the superiority and practicality of their method.

**Strengths:**

- This paper is clearly and well written.

- The authors give a theoretical analysis of their estimator, its log-variance upper bound in the general settings, and the counterfactual Markov boundary, and they also perform extensive experiments to demonstrate effectiveness in several cases: with two types of stochastic counterfactual processes, with three categories of fully specified SCMs, etc.

**Weaknesses:**

- This paper makes it clear in lines 140-144 about the assumptions needed for the proposed method. Regarding assumption ii), I think it is not mild, and I am wondering if the proposed method would be sensitive to the specified distribution $P_{\textbf{U}}$ of $\textbf{U}$. The authors might have performed such experiments, but it is not quite clear.

- Are the Markov boundaries learned from the observational data via the d-separation, or they are given prior? The authors claimed that such Markov boundaries are structural prior knowledge in line 223, whereas they gave Theorem 3 to demonstrate how to obtain them.

- It is suggested to offer the whole procedure or pseudo code of their proposed algorithm somewhere.

- In line 239, “augmentied” might be a typo error.

**Questions:**

Please see my questions in Weaknesses.

**Limitations:**

Not applicable.

---

> ### Author Rebuttal · Authors · 2024-08-06
>
> We thank the reviewer’s positive evaluation and valuable feedback. Below are our responses to the questions raised in the Weaknesses section:
>
> > **Weakness 1**: Regarding assumption ii), if the proposed method would be sensitive to the specified distribution $P_\mathbf{U}$ of $\mathbf{U}$?
>
> **Response**: The proposed method is theoretically insensitive to the exogenous distribution $P_\mathbf{U}$, as we do not impose any restrictions on the parameters or specific form of $P_\mathbf{U}$. However, we acknowledge that in our experiments, we only considered specific exogenous distributions. This is because, in practice, neural proxy SCMs, as deep generative models, typically use a form-specific and relatively simple distribution for latent variables to facilitate inference.
>
> > **Weakness 2**: Are the Markov boundaries learned from the observational data via the d-separation, or they are given prior?
>
> **Response**: We would like to emphasize that the counterfactual Markov boundary (Def. 2) for counterfactual estimation tasks serves as prior knowledge, which is not necessary and is known before estimation. The Markov boundary here refers to the counterfactual Markov boundary of exogenous variable, which can be directly derived from other prior knowledge (i.e., the augmented graph) as described in Thm. 2 and Thm. 3. Specifically, if the augmented graph is provided, we can use graph algorithms (Thm. 3) to effectively derive the Markov boundary of any exogenous variable. For learning the counterfactual Markov boundary from data, some methods cited in the manuscript (e.g., 3, 94, 111) cannot be directly applied unless the data includes observations of the exogenous variables. We thank the reviewer for the question and will refine the text to clarify this information.
>
> > **Weakness 3, 4**: It is suggested to offer the whole procedure or pseudo code of their proposed algorithm somewhere; In line 239, “augmentied” might be a typo error.
>
> We will carefully correct the typo errors in the manuscript and, in response to the reviewer's feedback, we will include pseudo code describing the proposed algorithm in the updated version.

---

### Author Rebuttal · Authors · 2024-08-06

We thank all the reviewers for their valuable feedbacks, which will help us improve the manuscript. Here, we summarize and address some common concerns, then list the changes to be made in the next updated manuscript.

### **Discussion on Assumptions**

- **Assumptions Required for Exogenous Matching**

    This refers to the necessary inputs for the Exogenous Matching algorithm. As indicated in lines 140-144 of the manuscript and as mentioned in 8MDB, these assumptions consist of two parts:

    1. An exogenous distribution $P_\mathbf{U}$ that is both sampleable and has computable density or probability.

        We do not impose any restrictions on its parameters or form. Please refer to our response to reviewer 8MDB for more details.

        Moreover, in practice, assuming the exogenous distribution is known is a weak assumption for using neural proxy SCMs for counterfactual estimation. See our response to reviewer aexG for further clarification.

    2. An evaluable causal mechanism $\mathcal{F}$.

        Here, "evaluable" means that given an input $\mathbf{u} \in \Omega_\mathbf{U}$, we can obtain the output $\mathcal{F}(\mathbf{u})$ without needing to know the specific expressions of all structural equations $f \in \mathcal{F}$. This leads to the concept of "fully specified," which is elaborated in our response to reviewer rY6N.

    These assumptions are weak for proxy SCMs; therefore, we have emphasized in the limitations section that our method does not directly estimate from data in practice but requires first training a proxy SCM, as demonstrated in our experiment "Counterfactual Estimation on Proxy SCMs."

- **Assumptions in Theorem 1**

    We acknowledge that the original description in the main text is not weak. To address this, we have introduced 3 more relaxed formulations. Please refer to our response to reviewer aexG for details. We will briefly incorporate these relaxations into the main text and provide their proofs in the appendix.


### **Clarification for Injecting Markov Boundaries**

In response to questions from reviewers 8MDB, 6eu3, and rY6N, we need to clarify some fundamental facts about "Injecting Markov Boundaries". Firstly, the "Markov Boundaries" here actually refer to counterfactual Markov Boundaries (Def. 2). Additionally, methods for learning Markov Boundaries from observational data cannot be directly applied to learning counterfactual Markov Boundaries. Therefore, we need Thm. 2 and Thm. 3 to derive counterfactual Markov Boundaries from the augmented graph. We will supplement this explanation in the updated version.

### **Discussion on Experiments**

- **Ablation**

    According to the feedback from reviewer 6eu3, the ablation experiments related to the Markov boundary exhibited anomalies, which could lead readers to perceive our method as not robust. We identified that these anomalies stem from the width of the hidden layers, which is a practical engineering issue and can be reasonably explained. For a detailed explanation, please refer to our response to reviewer 6eu3.

    We have updated the experiments accordingly, and the corresponding part of Fig. 4 in the main text will be revised to Fig. 1 in the Author Rebuttal PDF. We will also continue to refine the relevant experiments in the appendix, adding results with increased hidden layer widths, and include a discussion on hyperparameter selection and its impact.

- **Counterfactual Estimation on Proxy SCMs**

    Based on reviewer rY6N's suggestion, a more intuitive metric should be the deviation from the ground truth. We have elaborated on the intuition and rationale for choosing ESP and LL as metrics, which is due to the difficulty of obtaining the ground truth in high-dimensional settings in our experimental setup. For more details, please refer to our response to reviewer rY6N.

    Of course, for low-dimensional cases, such as in counterfactual effect estimation experiments, the ground truth is easily accessible. Therefore, we have updated the metrics for these experiments, and Tab. 2 in the main text will be revised to Tab. 1 in the Author Rebuttal PDF.

### **Discussion on Presentation**

Due to page limitations, we did indeed move some content to the appendix and inadvertently omitted some key information from the main text. We will carefully review and correct any typos, complete the missing explanations and results in the main text, and reorganize key sections with additional transitions to improve readability.

### **List of Changes**

Based on the reviewer’s suggestions, we will carry out the following changes:

- **Regarding the presentation in the main text:**
   1. Correct typo errors, reorganize the description of contributions, and ensure that legends clearly describe the elements in the figures. For example, eliminate the ambiguity caused by the missing legends in Fig. 1 and 2.
   2. Assess the inclusion of concepts, methods, and experimental results in the main text, and consider transferring relevant content from the appendix to the main text where appropriate, such as the explanation of the concept of faithfulness.
   3. Reorganize the presentation of some key sections, using smoother transitions or more intuitive ways to connect different parts to improve readability.

- **Regarding further clarification of assumptions:**
   1. Add additional discussion about the assumptions required for Exogenous Matching.
   2. Clarify the motivation and significance of (counterfactual) Markov boundaries, distinguish the proposed method from data-driven learning approaches.
   3. Add some relaxations to the assumptions in Thm. 1.

- **Regarding modifications to experimental figures:**
   1. Replace the corresponding part of Fig. 4 in the main text with Fig. 1 from the PDF.
   2. Replace Tab. 2 in the main text with Tab. 1 from the PDF.

---

### Decision · Program_Chairs · 2024-09-25

**Decision:**

Accept (poster)

**Comment:**

This paper aims to estimate counterfactual probabilities in Structural Causal Models, and the authors proposed EXOM, an importance sampling approach to solve this task. All reviewers are inclined towards a consensus of accepting the paper. During the rebuttal phase, the authors have added supporting arguments and results to justify the proposed method.

This is a particularly long paper for a conference: the paper ends on page 54. I encourage the authors to submit a cleaner, more concise version of the camera-ready version. In fact, the reviewers all raise concerns about the paper's clarity and comment that the paper is not easy to follow. The authors should also include the discussion about the assumptions of Theorem 1 in the camera-ready version.